# Online Bayesian Persuasion Without a Clue

**Francesco Bacchiocchi**
Politecnico di Milano
francesco.bacchiocchi@polimi.it

**Matteo Bollini**
Politecnico di Milano
matteo.bollini@polimi.it

**Matteo Castiglioni**
Politecnico di Milano
matteo.castiglioni@polimi.it

**Alberto Marchesi**
Politecnico di Milano
alberto.marchesi@polimi.it

**Nicola Gatti**
Politecnico di Milano
nicola.gatti@polimi.it

## Abstract

We study *online Bayesian persuasion* problems in which an informed sender repeatedly faces a receiver with the goal of influencing their behavior through the provision of payoff-relevant information. Previous works assume that the sender has *knowledge about* either the prior distribution over states of nature or receiver's utilities, or both. We relax such unrealistic assumptions by considering settings in which the sender does *not know anything* about the prior and the receiver. We design an algorithm that achieves sublinear—in the number of rounds—regret with respect to an optimal signaling scheme, and we also provide a collection of lower bounds showing that the guarantees of such an algorithm are tight. Our algorithm works by searching a suitable space of signaling schemes in order to learn receiver's best responses. To do this, we leverage a non-standard representation of signaling schemes that allows to cleverly overcome the challenge of *not* knowing anything about the prior over states of nature and receiver's utilities. Finally, our results also allow to derive lower/upper bounds on the *sample complexity* of learning signaling schemes in a related Bayesian persuasion PAC-learning problem.

## 1 Introduction

*Bayesian persuasion* has been introduced by Kamenica and Gentzkow [2011] to model how strategically disclosing information to decision makers influences their behavior. Over the last years, it has received a terrific attention in several fields of science, since it is particularly useful for understanding strategic interactions involving individuals with different levels of information, which are ubiquitous in the real world. As a consequence, Bayesian persuasion has been applied in several settings, such as online advertising [Emek et al., 2014, Badanidiyuru et al., 2018, Bacchiocchi et al., 2022, Agrawal et al., 2023], voting [Alonso and Câmara, 2016, Castiglioni et al., 2020a, Castiglioni and Gatti, 2021], traffic routing [Vasserman et al., 2015, Bhaskar et al., 2016, Castiglioni et al., 2021a], recommendation systems [Cohen and Mansour, 2019, Mansour et al., 2022], security [Rabinovich et al., 2015, Xu et al., 2016], e-commerce [Bro Miltersen and Sheffet, 2012, Castiglioni et al., 2022] medical research [Kolotilin, 2015], and financial regulation [Goldstein and Leitner, 2018].

In its simplest form, Bayesian persuasion involves a *sender* observing some information about the world, called *state of nature*, and a *receiver* who has to take an action. Agents' utilities are misaligned, but they both depend on the state of nature and receiver's action. Thus, sender's goal is to devise a

mechanism to (partially) disclose information to the receiver, so as to induce them to take a favorable action. This is accomplished by committing upfront to a *signaling scheme*, encoding a randomized policy that defines how to send informative signals to the receiver based on the observed state.

Classical Bayesian persuasion models (see, *e.g.*, [Dughmi and Xu, 2016, 2017, Xu, 2020]) rely on rather stringent assumptions that considerably limit their applicability in practice. Specifically, they assume that the sender perfectly knows the surrounding environment, including receiver's utilities and the probability distribution from which the state of nature is drawn, called *prior*. This has motivated a recent shift of attention towards Bayesian persuasion models that incorporate concepts and ideas from *online learning*, with the goal of relaxing some of such limiting assumptions. However, existing works only partially fulfill this goal, as they still assume some knowledge of either the prior (see, *e.g.*, [Castiglioni et al., 2020b, 2021b, 2023, Babichenko et al., 2022, Bernasconi et al., 2023]) or receiver's utilities (see, *e.g.*, [Zu et al., 2021, Bernasconi et al., 2022, Wu et al., 2022, Lin and Li, 2025]).

## 1.1 Original contributions

We address—for the first time to the best of our knowledge—Bayesian persuasion settings where the sender *has no clue* about the surrounding environment. In particular, we study the online learning problem faced by a sender who repeatedly interacts with a receiver over multiple rounds, *without* knowing anything about both the prior distribution over states of nature and receiver's utilities. At each round, the sender commits to a signaling scheme, and, then, they observe a state realization and send a signal to the receiver based on that. After each round, the sender gets *partial feedback*, namely, they only observe the best-response action played by the receiver in that round. In such a setting, the goal of the sender is to minimize their *regret*, which measures how much utility they lose with respect to committing to an optimal (*i.e.*, utility-maximizing) signaling scheme in every round.

We provide a learning algorithm that achieves regret of the order of $\widetilde{\mathcal{O}}(\sqrt{T})$, where $T$ is the number of rounds. We also provide lower bounds showing that the regret guarantees attained by our algorithm are tight in $T$ and in the parameters characterizing the Bayesian persuasion instance, *i.e.*, the number of states of nature $d$ and that of receiver's actions $n$. Our algorithm implements a sophisticated *explore-then-commit* scheme, with exploration being performed in a suitable space of signaling schemes so as to learn receiver's best responses *exactly*. This is crucial to attain tight regret guarantees, and it is made possible by employing a non-standard representation of signaling schemes, which allows to cleverly overcome the challenging lack of knowledge about both the prior and receiver's utilities.

Our results also allow us to derive lower/upper bounds on the *sample complexity* of learning signaling schemes in a related Bayesian persuasion PAC-learning problem, where the goal is to find, with high probability, an approximately-optimal signaling scheme in the minimum possible number of rounds.

## 1.2 Related works

Castiglioni et al. [2020b] were the first to introduce *online learning* problems in Bayesian persuasion scenarios, with the goal of relaxing sender's knowledge about receiver's utilities (see also follow-up works [Castiglioni et al., 2021b, 2023, Bernasconi et al., 2023]). In their setting, sender's uncertainty is modeled by means of an *adversary* selecting a receiver's *type* at each round, with types encoding information about receiver's utilities. However, in such a setting, the sender still needs knowledge about the finite set of possible receiver's types and their associated utilities, as well as about the prior.

A parallel research line has focused on relaxing sender's knowledge about the prior. Zu et al. [2021] study online learning in a repeated version of Bayesian persuasion. Differently from this paper, they consider the sender's learning problem of issuing *persuasive* action recommendations (corresponding to signals in their case), where persuasiveness is about correctly incentivizing the receiver to actually follow such recommendations. They provide an algorithm that attains sublinear regret while being persuasive at every round with high probability, despite having *no* knowledge of the prior. Wu et al. [2022], Gan et al. [2023], Bacchiocchi et al. [2024c] achieve similar results for Bayesian persuasion in episodic Markov decision processes, while Bernasconi et al. [2022] in non-Markovian environments. All these works crucially differ from ours, since they strongly rely on the assumption that receiver's utilities are known to the sender, which is needed in order to meet persuasiveness requirements. As a result, the techniques employed in such works are fundamentally different from ours as well.

Finally, learning receiver's best responses exactly (a fundamental component of our algorithm) is related to learning in Stackelberg games [Letchford et al., 2009, Peng et al., 2019, Bacchiocchi et al., 2024a]. For more details on these works and other related works, we refer the reader to Appendix A.

## 2 Preliminaries

In Section 2.1, we introduce all the needed ingredients of the classical *Bayesian persuasion* model by Kamenica and Gentzkow [2011], while, in the following Section 2.2, we formally define the Bayesian persuasion setting faced in the rest of the paper and its related *online learning* problem.

### 2.1 Bayesian persuasion

A Bayesian persuasion instance is characterized by a finite set $\Theta \coloneqq \{\theta_i\}_{i=1}^d$ of $d$ states of nature and a finite set $\mathcal{A} \coloneqq \{a_i\}_{i=1}^n$ of $n$ receiver's actions. Agents' payoffs are encoded by utility functions $u, u^{\mathrm{s}} : \Theta \times \mathcal{A} \to [0, 1]$, with $u_\theta(a) \coloneqq u(\theta, a)$, respectively $u_\theta^{\mathrm{s}}(a) \coloneqq u^{\mathrm{s}}(\theta, a)$, denoting the payoff of the receiver, respectively the sender, when action $a \in \mathcal{A}$ is played in state $\theta \in \Theta$. The sender observes a state of nature drawn from a commonly-known *prior* probability distribution $\mu \in \mathrm{int}(\Delta_\Theta)$,[1] with $\mu_\theta \in (0, 1]$ denoting the probability of $\theta \in \Theta$. To disclose information about the realized state, the sender can publicly commit upfront to a *signaling scheme* $\phi : \Theta \to \Delta_\mathcal{S}$, which defines a randomized mapping from states of nature to signals being sent to the receiver, for a finite set $\mathcal{S}$ of signals. For ease of notation, we let $\phi_\theta \coloneqq \phi(\theta)$ be the probability distribution over signals prescribed by $\phi$ when the the sate of nature is $\theta \in \Theta$, with $\phi_\theta(s) \in [0, 1]$ denoting the probability of sending signal $s \in \mathcal{S}$.

The sender-receiver interaction goes as follows: (1) the sender commits to a signaling scheme $\phi$; (2) the sender observes a state of nature $\theta \sim \mu$ and sends a signal $s \sim \phi_\theta$ to the receiver; (3) the receiver updates their belief over states of nature according to *Bayes rule*; and (4) the receiver plays a best-response action $a \in \mathcal{A}$, with sender and receiver getting payoffs $u_\theta(a)$ and $u_\theta^{\mathrm{s}}(a)$, respectively. Specifically, an action is a *best response* for the receiver if it maximizes their expected utility given the belief computed in step (3) of the interaction. Formally, given a signaling scheme $\phi : \Theta \to \Delta_\mathcal{S}$ and a signal $s \in \mathcal{S}$, we let $\mathcal{A}^\phi(s) \subseteq \mathcal{A}$ be the set of receivers' best-response actions, where:

$$\mathcal{A}^\phi(s) \coloneqq \left\{ a_i \in \mathcal{A} \mid \sum_{\theta \in \Theta} \mu_\theta \phi_\theta(s) u_\theta(a_i) \geq \sum_{\theta \in \Theta} \mu_\theta \phi_\theta(s) u_\theta(a_j) \quad \forall a_j \in \mathcal{A} \right\}. \tag{1}$$

As customary in the literature on Bayesian persuasion (see, *e.g.*, [Dughmi and Xu, 2016]), we assume that, when the receiver has multiple best responses available, they break ties in favor of the sender. In particular, we let $a^\phi(s)$ be the best response that is actually played by the receiver when observing signal $s \in \mathcal{S}$ under signaling scheme $\phi$, with $a^\phi(s) \in \arg\max_{a \in \mathcal{A}^\phi(s)} \sum_{\theta \in \Theta} \mu_\theta \phi_\theta(s) u_\theta^{\mathrm{s}}(a)$.

The goal of the sender is to commit to an *optimal* signaling scheme, namely, a $\phi : \Theta \to \Delta_\mathcal{S}$ that maximizes sender's expected utility, defined as $u^{\mathrm{s}}(\phi) \coloneqq \sum_{s \in \mathcal{S}} \sum_{\theta \in \Theta} \mu_\theta \phi_\theta(s) u_\theta^{\mathrm{s}}(a^\phi(s))$. In the following, we let $\mathrm{OPT} \coloneqq \max_\phi u^{\mathrm{s}}(\phi)$ be the optimal value of sender's expected utility. Moreover, given an additive error $\gamma \in (0, 1)$, we say that a signaling scheme $\phi$ is $\gamma$-optimal if $u^{\mathrm{s}}(\phi) \geq \mathrm{OPT} - \gamma$.

### 2.2 Learning in Bayesian persuasion

We study settings in which the sender *repeatedly* interacts with the receiver over multiple rounds, with each round involving a one-shot Bayesian persuasion interaction (as described in Section 2.1). We assume that the sender has *no* knowledge about both the prior $\mu$ and receiver's utility $u$, and that the only feedback they get after each round is the best-response action played by the receiver.

At each round $t \in [T]$,[2] the sender commits to a signaling scheme $\phi_t : \Theta \to \Delta_\mathcal{S}$ and observes a state of nature $\theta^t \sim \mu$. Then, they draw a signal $s^t \sim \phi_{t,\theta^t}$ and send it to the receiver, who plays a best-response action $a^t \coloneqq a^{\phi_t}(s^t)$. Finally, the sender gets payoff $u_t^{\mathrm{s}} \coloneqq u_{\theta^t}^{\mathrm{s}}(a^t)$ and observes a *feedback* consisting in the action $a^t$ played by the receiver. The goal of the sender is to learn how to maximize their expected utility while repeatedly interacting with the receiver. When the

---

[1]Given a finite set $X$, we denote by $\Delta_X$ the set of all the probability distributions over $X$.

[2]We denote by $[n] \coloneqq \{1, \ldots, n\}$ the set of the first $n \in \mathbb{N}$ natural numbers.

sender commits to a sequence $\{\phi_t\}_{t\in[T]}$ of signaling schemes, their performance over the $T$ rounds is measured by means of the following notion of *cumulative (Stackelberg) regret*:

$$R_T(\{\phi_t\}_{t\in[T]}) := T \cdot \text{OPT} - \mathbb{E}\left[\sum_{t=1}^{T} u^s(\phi_t)\right],$$

where the expectation is with respect to the randomness of the algorithm. In the following, for ease of notation, we omit the dependency on $\{\phi_t\}_{t\in[T]}$ from the cumulative regret, by simply writing $R_T$. Then, our goal is to design *no-regret* learning algorithms for the sender, which prescribe a sequence of signaling schemes $\phi_t$ that results in the regret $R_T$ growing sublinearly in $T$, namely $R_T = o(T)$.

## 3 Warm-up: A single signal is all you need

In order to design our learning algorithm in Section 4, we exploit a non-standard representation of signaling schemes, which we introduce in this section. Adopting such a representation is fundamental to be able to learn receiver's best responses *without* any knowledge of both the prior $\mu$ and receiver's utility function $u$. The crucial observation that motivates its adoption is that receiver's best responses $a^\phi(s)$ only depend on the components of $\phi$ associated with $s \in \mathcal{S}$, namely $\phi_\theta(s)$ for $\theta \in \Theta$. Thus, in order to learn them, it is sufficient to learn how $a^\phi(s)$ varies as a function of such components.

The signaling scheme representation introduced in this section revolves around the concept of *slice*.

**Definition 1** (Slice). *Given a signaling scheme $\phi : \Theta \to \Delta_S$, the* slice *of $\phi$ with respect to signal $s \in \mathcal{S}$ is the $d$-dimensional vector $x \in [0,1]^d$ with components $x_\theta := \phi_\theta(s)$ for $\theta \in \Theta$.*

In the following, we denote by $\mathcal{X}^\square := [0,1]^d$ the set of *all* the possible slices of signaling schemes. Moreover, we let $\mathcal{X}^\triangle$ be the set of *normalized* slices, which is simply obtained by restricting slices to lie in the $(d-1)$-dimensional simplex. Thus, it holds that $\mathcal{X}^\triangle := \{x \in [0,1]^d \mid \sum_{\theta \in \Theta} x_\theta = 1\}$.

Next, we show that receiver's actions induce particular coverings of the sets $\mathcal{X}^\square$ and $\mathcal{X}^\triangle$, which also depend on both the prior $\mu$ and receiver's utility $u$. First, we introduce $\mathcal{H}_{ij} \subseteq \mathbb{R}^d$ to denote the halfspace of slices under which action $a_i \in A$ is (weakly) better than action $a_j \in A$ for the receiver.

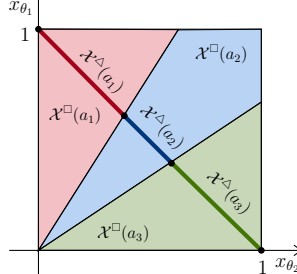

$$\mathcal{H}_{ij} := \left\{x \in \mathbb{R}^d \mid \sum_{\theta \in \Theta} x_\theta \mu_\theta \big(u_\theta(a_i) - u_\theta(a_j)\big) \geq 0\right\}.$$

Figure 1: Representation of sets $\mathcal{X}^\square(a_i)$ and $\mathcal{X}^\triangle(a_i)$ for an instance with $d = 2$ states of nature and $n = 3$ receivers' actions.

Moreover, we denote by $H_{ij} := \partial \mathcal{H}_{ij}$ the hyperplane constituting the boundary of the halfspace $\mathcal{H}_{ij}$, which we call the *separating hyperplane* between actions $a_i$ and $a_j$.[3] Then, for every $a_i \in \mathcal{A}$, we introduce the polytopes $\mathcal{X}^\square(a_i) \subseteq \mathcal{X}^\square$ and $\mathcal{X}^\triangle(a_i) \subseteq \mathcal{X}^\triangle$:

$$\mathcal{X}^\square(a_i) := \mathcal{X}^\square \cap \left(\bigcap_{a_j \in \mathcal{A}: a_i \neq a_j} \mathcal{H}_{ij}\right) \quad \text{and} \quad \mathcal{X}^\triangle(a_i) := \mathcal{X}^\triangle \cap \left(\bigcap_{a_j \in \mathcal{A}: a_i \neq a_j} \mathcal{H}_{ij}\right).$$

Clearly, the sets $\mathcal{X}^\square(a_i)$, respectively $\mathcal{X}^\triangle(a_i)$, define a cover of $\mathcal{X}^\square$, respectively $\mathcal{X}^\triangle$.[4] Intuitively, the set $\mathcal{X}^\square(a_i)$ encompasses all the slices under which action $a_i$ is a best response for the receiver. The set $\mathcal{X}^\triangle(a_i)$ has the same interpretation, but for normalized slices. Specifically, if $x \in \mathcal{X}^\square(a_i)$ is a slice of $\phi$ with respect to $s \in \mathcal{S}$, then $a_i \in \mathcal{A}^\phi(s)$. Notice that a slice $x \in \mathcal{X}^\square$ may belong to more than one polytope $\mathcal{X}^\square(a_i)$, when it is the case that $|\mathcal{A}^\phi(s)| > 1$ for signaling schemes $\phi$ having $x$

---

[3]We let $\partial \mathcal{H}$ be the boundary hyperplane of halfspace $\mathcal{H} \subseteq \mathbb{R}^d$. Notice that $H_{ij}$ and $H_{ji}$ actually refer to the same hyperplane. In this paper, we use both names for ease of presentation.

[4]In this paper, given a polytope $\mathcal{P} \subseteq \mathbb{R}^D$, we let $V(\mathcal{P})$ be its set of vertices, while we denote by $\text{vol}_m(\mathcal{P})$ its Lebesgue measure in $m$ dimensions. For ease of notation, whenever $m = D - 1$, we simply write $\text{vol}(\mathcal{P})$. Moreover, we let $\text{int}(\mathcal{P})$ be the interior of $\mathcal{P}$ relative to a subspace that fully contains $\mathcal{P}$ and has minimum dimension. In the case of polytopes $\mathcal{X}^\triangle(a_i)$, the $(d-1)$-dimensional simplex is one of such subspaces.

as slice with respect to $s \in \mathcal{S}$.[5] In order to denote the best-response action actually played by the receiver under a slice $x \in \mathcal{X}^\square$, we introduce the symbol $a(x)$, where $a(x) := a^\phi(s)$ for any $\phi$ having $x$ as slice with respect to $s \in \mathcal{S}$. Figure 1 depicts an example of the polytopes $\mathcal{X}^\square(a_i)$ and $\mathcal{X}^\triangle(a_i)$ in order to help the reader to grasp more intuition about them.

A crucial fact exploited by the learning algorithm developed in Section 4 is that knowing the separating hyperplanes $H_{ij}$ defining the polytopes $\mathcal{X}^\triangle(a_i)$ of normalized slices is *sufficient* to determine an optimal signaling scheme. Indeed, the polytopes $\mathcal{X}^\square(a_i)$ of unnormalized slices can be easily reconstructed by simply removing the normalization constraint $\sum_{\theta \in \Theta} x_\theta = 1$ from $\mathcal{X}^\triangle(a_i)$. Furthermore, as we show in Section 4 (see the proof of Lemma 4 in particular), there alway exits an optimal signaling scheme using at most one slice $x^a \in \mathcal{X}^\square(a)$ for each receiver's action $a \in \mathcal{A}$.

We conclude the section with some remarks that help to better clarify why we need to work with signaling scheme slices in order to design our learning algorithm in Section 4.

**Why we need slices for learning**  The coefficients of separating hyperplanes $H_{ij}$ are products between prior probabilities $\mu_\theta$ and receiver's utility differences $u_\theta(a_i) - u_\theta(a_j)$. In order to design a no-regret learning algorithm, it is fundamental that such coefficients are learned exactly, since even an arbitrarily small approximation error may result in "missing" some receiver's best responses, and this may potentially lead to a large loss in sender's expected utility (and, in its turn, to large regret). As a result, any naïve approach that learns the prior and receiver's payoffs separately is deemed to fail, as it would inevitably result in approximate separating hyperplanes being learned. Operating in the space of signaling scheme slices crucially allows us to learn separating hyperplanes exactly. As we show in Section 4, it makes it possible to directly learn the coefficients of separating hyperplanes without splitting them into products of prior probabilities and receiver's utility differences.

**Why we need normalized slices**  One may wonder why we cannot work with (unnormalized) slices in $\mathcal{X}^\square$, rather than with normalized ones in $\mathcal{X}^\triangle$. Indeed, as we show in Section 4, our procedure to learn separating hyperplanes crucially relies on the fact that we can restrict the search space to a suitable subset of the $(d-1)$-dimensional simplex. This makes it possible to avoid always employing a number of rounds exponential in $d$, which would lead to non-tight regret guarantees.

**Why *not* working with posteriors**  In Bayesian persuasion, it is oftentimes useful to work in the space of posterior distributions induced by sender's signals (see, *e.g.*, [Castiglioni et al., 2020b]). These are the beliefs computed by the receiver according to Bayes rule at step (3) of the interaction. Notice that posteriors do *not* only depend on the signaling scheme $\phi$ and the sent signal $s \in \mathcal{S}$, but also on the prior distribution $\mu$. Indeed, the same signaling scheme may induce different posteriors for different prior distributions. Thus, since in our setting the sender has no knowledge of $\mu$, we cannot employ posteriors. Looking at signaling scheme slices crucially allows us to overcome the lack of knowledge of the prior. Indeed, one way of thinking of them is as "prior-free" posterior distributions.

## 4  Learning to persuade without a clue

In this section, we design our no-regret algorithm (Algorithm 1). We adopt a sophisticated *explore-then-commit* approach that exploits the signaling scheme representation based on slices introduced in Section 3. Specifically, our algorithm works by first exploring the space $\mathcal{X} := \mathcal{X}^\triangle$ of normalized slices in order to learn satisfactory "approximations" of the polytopes $\mathcal{X}(a_i) := \mathcal{X}^\triangle(a_i)$.[6] Then, it exploits them in order to compute suitable approximately-optimal signaling schemes to be employed in the remaining rounds. Effectively implementing this approach raises considerable challenges.

The **first** challenge is that the algorithm cannot directly "query" a slice $x \in \mathcal{X}$ to know action $a(x)$, as it can only commit to fully-specified signaling schemes. Indeed, even if the algorithm commits to a signaling scheme including the selected slice $x \in \mathcal{X}$, the probability that the signal associated with $x$ is sent depends on the (unknown) prior, as it is equal to $\sum_{\theta \in \Theta} \mu_\theta x_\theta$. This probability can be

---

[5]Let us remark that, in this paper, we assume that there are *no* two equivalent receiver's actions $a_i \neq a_j \in \mathcal{A}$ such that $u_\theta(a_i) = u_\theta(a_j)$ for all $\theta \in \Theta$. Thus, a slice can belong to more than one polytope only if it lies on some separating hyperplane. This assumption is w.l.o.g. since it is possible to account for equivalent actions by introducing at most $n$ additional separating hyperplanes to consider tie breaking in favor of the sender.

[6]In the rest of the paper, for ease of notation, we write $\mathcal{X}$ and $\mathcal{X}(a_i)$ instead of $\mathcal{X}^\triangle$ and $\mathcal{X}^\triangle(a_i)$.

arbitrarily small. Thus, in order to observe $a(x)$, the algorithm may need to commit to the signaling scheme for an unreasonably large number of rounds. To circumvent this issue, we show that it is possible to focus on a subset $\mathcal{X}_\epsilon \subseteq \mathcal{X}$ of normalized slices "inducible" with at least a suitably-defined probability $\epsilon \in (0, 1)$. Such a set $\mathcal{X}_\epsilon$ is built by the algorithm in its first phase.

The **second** challenge that we face is learning the polytopes $\mathcal{X}_\epsilon(a_i) := \mathcal{X}(a_i) \cap \mathcal{X}_\epsilon$. This is done by means of a technically-involved procedure that learns the separating hyperplanes $H_{ij}$ needed to identify them. This procedure is an adaptation to Bayesian persuasion settings of an algorithm recently introduced for a similar problem in Stackelberg games [Bacchiocchi et al., 2024a].

Finally, the **third** challenge is how to use the polytopes $\mathcal{X}_\epsilon(a_i)$ to compute suitable approximately-optimal signaling schemes to commit to after exploration. We show that this can be done by solving an LP, which, provided that the set $\mathcal{X}_\epsilon$ is carefully constructed, gives signaling schemes with sender's expected utility sufficiently close to that of an optimal signaling scheme.

The pseudocode of our no-regret learning algorithm is provided in Algorithm 1. In the pseudocode, we assume that all the sub-procedures have access to the current round counter $t$, all the observed states of nature $\theta^t$, and all the feedbacks $a^t, u^s_t$ received by the sender.[7] Moreover, in Algorithm 1 and its sub-procedures, we use $\widehat{\mu}_t \in \Delta_\Theta$ to denote the *prior estimate* at any round $t > 1$, which is a vector with components defined as $\widehat{\mu}_{t,\theta} := N_{t,\theta}/(t-1)$ for $\theta \in \Theta$, where $N_{t,\theta} \in \mathbb{N}$ denotes the number of times that state of nature $\theta$ is observed up to round $t$ (excluded). Algorithm 1 can be conceptually divided into three phases. In *phase 1*, the algorithm employs the `Build-Search-Space` procedure (Algorithm 2) to build a suitable subset $\mathcal{X}_\epsilon \subseteq \mathcal{X}$ of "inducible" normalized slices. Then,

---

**Algorithm 1** `Learn-to-Persuade-w/o-Clue`

**Require:** $T \in \mathbb{N}$
1: $\delta \leftarrow \sfrac{1}{T}, \zeta \leftarrow \sfrac{1}{T}$
2: $\epsilon \leftarrow \left\lceil \frac{\sqrt{Bn}d^4}{\sqrt{T}} \right\rceil$
3: $T_1 \leftarrow \left\lceil \frac{12}{\epsilon} \log\left(\frac{2d}{\delta}\right) \right\rceil$
4: $t \leftarrow 1$
5: $\mathcal{X}_\epsilon \leftarrow$ `Build-Search-Space`$(T_1, \epsilon)$
6: $\mathcal{R}_\epsilon \leftarrow$ `Find-Polytopes`$(\mathcal{X}_\epsilon, \zeta)$
7: **while** $t \leq T$ **do**
8: $\quad \phi_t \leftarrow$ `Compute-Signaling`$(\mathcal{R}_\epsilon, \mathcal{X}_\epsilon, \widehat{\mu}_t)$
9: $\quad$ Commit to $\phi_t$, observe $\theta^t$, and send $s^t$
10: $\quad$ Observe feedback $a^t$ and receive $u^s_t$
11: $\quad$ Compute prior estimate $\widehat{\mu}_{t+1}$
12: $\quad t \leftarrow t + 1$

---

in *phase 2*, the algorithm employs the `Find-Polytopes` procedure (see Algorithm 6 in Appendix E) to find a collection of polytopes $\mathcal{R} := \{\mathcal{R}_\epsilon(a)\}_{a \in \mathcal{A}}$, where each $\mathcal{R}_\epsilon(a)$ is either $\mathcal{X}_\epsilon(a)$ or a suitable subset of $\mathcal{X}_\epsilon(a)$ that is sufficient for achieving the desired goals (see Section 4.2). Finally, *phase 3* uses the remaining rounds to exploit the knowledge acquired in the preceding two phases. Specifically, at each $t$, this phase employs the `Compute-Signaling` procedure (Algorithm 3) to compute an approximately-optimal signaling scheme, by using $\mathcal{R}_\epsilon$, the set $\mathcal{X}_\epsilon$, and the current prior estimate $\widehat{\mu}_t$.

In the rest of this section, we describe in detail the three phases of Algorithm 1, bounding the regret attained by each of them. This allows us to prove the following main result about Algorithm 1.[8]

**Theorem 1.** *The regret attained by Algorithm 1 is* $R_T \leq \widetilde{\mathcal{O}}\left(\binom{d+n}{d} n^{3/2} d^3 \sqrt{BT}\right)$.

We observe that the regret bound in Theorem 1 has an exponential dependence on the number of states of nature $d$ and the number of receiver's actions $n$, due to the binomial coefficient. Indeed, when $d$, respectively $n$, is constant, the regret bound is of the order of $\widetilde{\mathcal{O}}(n^d \sqrt{T})$, respectively $\widetilde{\mathcal{O}}(d^n \sqrt{T})$. Such a dependence is tight, as shown by the lower bounds that we provide in Section 5.

### 4.1 Phase 1: `Build-Search-Space`

Given an $\epsilon \in (0, \sfrac{1}{6d})$ and a number of rounds $T_1$, the `Build-Search-Space` procedure (Algorithm 2) computes a subset $\mathcal{X}_\epsilon \subseteq \mathcal{X}$ of normalized slices satisfying two crucial properties needed by the learning algorithm to attain the desired guarantees. Specifically, the first property is that any slice $x \in \mathcal{X}_\epsilon$ can be "induced" with sufficiently high probability by a signaling scheme, while the

---

[7]Notice that, in Algorithm 1, the sub-procedures `Build-Search-Space` and `Find-Polytopes` perform some rounds of interaction, and, thus, they update the current round counter $t$. For ease of presentation, we assume that, whenever $t > T$, their execution is immediately stopped (as well as the execution of Algorithm 1).

[8]Notice that the regret attained by Algorithm 6 (stated in Theorem 1) depends on $B$, which is the bit-complexity of numbers $\mu_\theta u_\theta(a_i)$, *i.e.*, the number of bits required to represent them. We refer the reader to Appendix E for more details about how we manage the bit-complexity of numbers in this paper.

second one is that, if $x \notin \mathcal{X}_\epsilon$, then signaling schemes "induce" such a slice with sufficiently small probability. Intuitively, the first property ensures that it is possible to associate any $x \in \mathcal{X}_\epsilon$ with the action $a(x)$ in a number of rounds of the order of $1/\epsilon$, while the second property is needed to bound the loss in sender's expected utility due to only considering signaling schemes with slices in $\mathcal{X}_\epsilon$.

Algorithm 2 works by simply observing realized states of nature $\theta^t$ for $T_1$ rounds, while committing to any signaling scheme meanwhile. This allows the algorithm to build a sufficiently accurate estimate $\widehat{\mu}$ of the true prior $\mu$. Then, the algorithm uses such an estimate to build the set $\mathcal{X}_\epsilon$. Specifically, it constructs $\mathcal{X}_\epsilon$ as the set containing all the normalized slices that are "inducible" with probability at least $2\epsilon$ under the estimated prior $\widehat{\mu}$, after filtering out all the states of nature whose estimated probability is *not* above the $2\epsilon$ threshold (see Lines 8–10 in Algorithm 2).

---

**Algorithm 2** `Build-Search-Space`

**Require:** $\epsilon \in (0, 1/6d)$, number of rounds $T_1$
1: $\widetilde{\Theta} \leftarrow \varnothing$
2: **while** $t \leq T_1$ **do**
3:      Commit to any $\phi_t$, observe $\theta^t$, and send $s^t$
4:      Observe feedback $a^t$ and receive $u_t^{\mathsf{s}}$
5:      $t \leftarrow t + 1$
6: Compute prior estimate $\widehat{\mu}_t$
7: $\widehat{\mu} \leftarrow \widehat{\mu}_t$
8: **for** $\theta \in \Theta$ **do**
9:      **if** $\widehat{\mu}_\theta > 2\epsilon$ **then** $\widetilde{\Theta} \leftarrow \widetilde{\Theta} \cup \{\theta\}$
10: $\mathcal{X}_\epsilon \leftarrow \left\{ x \in \mathcal{X} \mid \sum_{\theta \in \widetilde{\Theta}} \widehat{\mu}_\theta x_\theta \geq 2\epsilon \right\}$
11: **return** $\mathcal{X}_\epsilon$

---

The following lemma formally establishes the two crucial properties that are guaranteed by Algorithm 2, as informally described above.

**Lemma 1.** *Given $T_1 := \left\lceil \frac{12}{\epsilon} \log \left( 2d/\delta \right) \right\rceil$ and $\epsilon \in (0, 1/6d)$, Algorithm 2 employs $T_1$ rounds and terminates with a set $\mathcal{X}_\epsilon \subseteq \mathcal{X}$ such that, with probability at least $1 - \delta$: (i) $\sum_{\theta \in \Theta} \mu_\theta x_\theta \geq \epsilon$ for every slice $x \in \mathcal{X}_\epsilon$ and (ii) $\sum_{\theta \in \Theta} \mu_\theta x_\theta \leq 10\epsilon$ for every slice $x \in \mathcal{X} \setminus \mathcal{X}_\epsilon$.*

To prove Lemma 1, we employ the multiplicative version of Chernoff bound, so as to show that it is possible to distinguish whether prior probabilities $\mu_\theta$ are above or below suitable thresholds in a number of rounds of the order of $1/\epsilon$. Specifically, we show that, after $T_1$ rounds and with probability at least $1 - \delta$, the set $\widetilde{\Theta}$ in the definition of $\mathcal{X}_\epsilon$ does *not* contain states $\theta \in \Theta$ with $\mu_\theta \leq \epsilon$, while it contains all the states with a sufficiently large $\mu_\theta$. This immediately proves properties (i) and (ii) in Lemma 1. Notice that using a multiplicative Chernoff bound is a necessary technicality, since standard concentration inequalities would result in a number of needed rounds of the order of $1/\epsilon^2$, leading to a suboptimal regret bound in the number of rounds $T$.

For ease of presentation, we introduce the following *clean event* for phase 1 of Algorithm 1. This encompasses all the situations in which Algorithm 2 outputs a set $\mathcal{X}_\epsilon$ with the desired properties.

**Definition 2** (Phase 1 clean event). *$\mathcal{E}_1$ is the event in which $\mathcal{X}_\epsilon$ meets properties (i)–(ii) in Lemma 1.*

### 4.2 Phase 2: `Find-Polytopes`

Given a set $\mathcal{X}_\epsilon \subseteq \mathcal{X}$ computed by the `Build-Search-Space` procedure and $\zeta \in (0,1)$ as inputs, the `Find-Polytopes` procedure (Algorithm 6 in Appendix E) computes a collection $\mathcal{R}_\epsilon := \{\mathcal{R}_\epsilon(a)\}_{a \in \mathcal{A}}$ of polytopes enjoying suitable properties sufficient to achieve the desired goals.

Ideally, we would like $\mathcal{R}_\epsilon(a) = \mathcal{X}_\epsilon(a)$ for every $a \in \mathcal{A}$. However, it is *not* possible to completely identify the polytopes $\mathcal{X}_\epsilon(a)$ with $\text{vol}(\mathcal{X}_\epsilon(a)) = 0$. Indeed, if $\text{vol}(\mathcal{X}_\epsilon(a_i)) = 0$, then $\mathcal{X}_\epsilon(a_i) \subseteq \mathcal{X}_\epsilon(a_j)$ for some other polytope $\mathcal{X}_\epsilon(a_j)$ with positive volume. Thus, due to receiver's tie breaking, it could be impossible to identify the whole polytope $\mathcal{X}_\epsilon(a_i)$. As a result, the `Find-Polytopes` procedure can output polytopes $\mathcal{R}_\epsilon(a) = \mathcal{X}_\epsilon(a)$ only if $\text{vol}(\mathcal{X}_\epsilon(a)) > 0$. However, we show that, if $\text{vol}(\mathcal{X}_\epsilon(a)) = 0$, it is sufficient to guarantee that the polytope $\mathcal{R}_\epsilon(a)$ contains a suitable subset $\mathcal{V}_\epsilon(a)$ of the vertices of $\mathcal{X}_\epsilon(a)$; specifically, those in which the best response actually played by the receiver is $a$. For every $a \in \mathcal{A}$, such a set is formally defined as $\mathcal{V}_\epsilon(a) := \{x \in V(\mathcal{X}_\epsilon(a)) \mid a(x) = a\}$. Thus, we design `Find-Polytopes` so that it returns a collection $\mathcal{R}_\epsilon := \{\mathcal{R}_\epsilon(a)\}_{a \in \mathcal{A}}$ of polytopes such that:

    (i) if it holds $\text{vol}(\mathcal{X}_\epsilon(a)) > 0$, then $\mathcal{R}_\epsilon(a) = \mathcal{X}_\epsilon(a)$, while
    (ii) if $\text{vol}(\mathcal{X}_\epsilon(a)) = 0$, then $\mathcal{R}_\epsilon(a)$ is a (possible improper) face of $\mathcal{X}_\epsilon(a)$ with $\mathcal{V}_\epsilon(a) \subseteq \mathcal{R}_\epsilon(a)$.

As a result each polytope $\mathcal{R}_\epsilon(a)$ can be either $\mathcal{X}_\epsilon(a)$ or a face of $\mathcal{X}_\epsilon(a)$, or it can be empty, depending on receiver's tie breaking. In all these cases, it is always guaranteed that $\mathcal{V}_\epsilon(a) \subseteq \mathcal{R}_\epsilon(a)$.

To achieve its goal, the `Find-Polytopes` procedure works by searching over the space of normalized slices $\mathcal{X}_\epsilon$, so as to learn *exactly* all the separating hyperplanes $H_{ij}$ characterizing the needed vertices.

The algorithm does so by using and extending tools that have been developed for a related learning problem in Stackelberg games (see [Bacchiocchi et al., 2024a]). Notice that our Bayesian persuasion setting has some distinguishing features that do *not* allow us to use such tools off the shelf. We refer the reader to Appendix E for a complete description of the `Find-Polytopes` procedure.

A crucial component of `Find-Polytopes` is a tool to "query" a normalized slice $x \in \mathcal{X}_\epsilon$ in order to obtain the action $a(x)$. This is done by using a sub-procedure that we call `Action-Oracle` (see Algorithm 5 in Appendix E), which works by committing to a signaling scheme including slice $x$ until the signal corresponding to such a slice is actually sent. Under the clean event $\mathcal{E}_1$, the set $\mathcal{X}_\epsilon$ is built in such a way that `Action-Oracle` returns the desired action $a(x)$ in a number of rounds of the order of $1/\epsilon$, with high probability. This is made formal by the following lemma.

**Lemma 2.** *Under event $\mathcal{E}^1$, given any $\rho \in (0,1)$ and a normalized slice $x \in \mathcal{X}_\epsilon$, if the sender commits to a signaling scheme $\phi : \Theta \to \mathcal{S} := \{s_1, s_2\}$ such that $\phi_\theta(s_1) = x_\theta$ for all $\theta \in \Theta$ during $q := \left\lceil \frac{1}{\epsilon} \log(1/\rho) \right\rceil$ rounds, then, with probability at least $1 - \rho$, signal $s_1$ is sent at least once.*

Notice that the signaling scheme used to "query" an $x \in \mathcal{X}_\epsilon$ only employs *two* signals: one is associated with slice $x$, while the other crafted so as to make the signaling scheme well defined.

The following lemma provides the guarantees of Algorithm 6 in Appendix E.

**Lemma 3.** *Given inputs $\mathcal{X}_\epsilon \subseteq \mathcal{X}$ and $\zeta \in (0,1)$ for Algorithm 6, let $L := B + B_\epsilon + B_{\widehat{\mu}}$, where $B$, $B_\epsilon$, and $B_{\widehat{\mu}}$ denote the bit-complexity of numbers $\mu_\theta u_\theta(a_i)$, $\epsilon$, and $\widehat{\mu}$, respectively. Then, under event $\mathcal{E}_1$ and with at probability at least $1 - \zeta$, Algorithm 6 outputs a collection $\mathcal{R}_\epsilon := \{\mathcal{R}_\epsilon(a)\}_{a \in \mathcal{A}}$, where $\mathcal{R}_\epsilon(a)$ is a (possibly improper) face of $\mathcal{X}_\epsilon(a)$ such that $\mathcal{V}_\epsilon(a) \subseteq \mathcal{X}_\epsilon(a)$, in a number of rounds $T_2$:*

$$T_2 \leq \widetilde{\mathcal{O}}\left( \frac{n^2}{\epsilon} \log^2\left( \frac{1}{\zeta} \right) \left( d^7 L + \binom{d+n}{d} \right) \right).$$

For ease of presentation, we introduce the *clean event* for phase 2 of Algorithm 1, defined as follows:

**Definition 3** (Phase 2 clean event). *$\mathcal{E}_2$ is the event in which $\mathcal{V}_\epsilon(a) \subseteq \mathcal{R}_\epsilon(a)$ for every $a \in \mathcal{A}$.*

### 4.3 Phase 3: `Compute-Signaling`

Given the collection of polytopes $\mathcal{R}_\epsilon := \{\mathcal{R}_\epsilon(a)\}_{a \in \mathcal{A}}$ returned by the `Find-Polytopes` procedure and an estimated prior $\widehat{\mu}_t \in \Delta_\Theta$, the `Compute-Signaling` procedure (Algorithm 3) outputs an approximately-optimal signaling scheme by solving an LP (Program (2) in Algorithm 3).

Program (2) maximizes an approximated version of sender's expected utility over a suitable space of (partially-specified) signaling schemes. These are defined by tuples of slices $(x^a)_{a \in \mathcal{A}}$ containing an (unnormalized) slice $x^a \in \mathcal{R}_\epsilon^\square(a)$ for every receiver's action $a \in \mathcal{A}$. The objective function being maximized by Program (2) accounts for the sender's approximate utility under each of the slices $x^a$, where the approximation comes from the estimated prior $\widehat{\mu}_t$. The intuitive idea exploited by the LP formulation is that, under slice $x^a$, the receiver always plays the same action $a$ as best response, since $a(x^a) = a$ holds by the way in which $\mathcal{R}_\epsilon(a)$ is constructed by `Find-Polytopes`. In particular, each polytope $\mathcal{R}_\epsilon^\square(a)$ is built so as to include all the unnormalized slices corresponding to the normalized slices in the set $\mathcal{R}_\epsilon(a)$. Formally, for every receiver's action $a \in \mathcal{A}$, it holds:

---

**Algorithm 3** `Compute-Signaling`

---

**Require:** $\mathcal{R}_\epsilon := \{\mathcal{R}_\epsilon(a)\}_{a \in \mathcal{A}}, \mathcal{X}_\epsilon, \widehat{\mu}_t \in \Delta_\Theta$
 1: Solve Program (2) for $x^\star := (x^{\star,a})_{a \in \mathcal{A}}$:

$$\max_{(x^a)_{a \in \mathcal{A}}} \sum_{a \in \mathcal{A}} \sum_{\theta \in \Theta} \widehat{\mu}_{t,\theta} x_\theta^a u_\theta^{\mathrm{s}}(a) \quad \text{s.t.} \quad (2)$$

$$x^a \in \mathcal{R}_\epsilon^\square(a) \quad \forall a \in \mathcal{A}$$

$$\sum_{a \in \mathcal{A}} x_\theta^a \leq 1 \quad \forall \theta \in \Theta$$

 2: $\mathcal{S} \leftarrow \{s^\star\} \cup \{s^a \mid a \in \mathcal{A}\}$
 3: **for** $\theta \in \Theta$ **do**
 4: $\quad \phi_\theta \leftarrow \begin{cases} \phi_\theta(s^a) = x_\theta^{\star,a} & \forall a \in \mathcal{A} \\ \phi_\theta(s^\star) = 1 - \sum_{a \in \mathcal{A}} x_\theta^{\star,a} \end{cases}$
 5: **return** $\phi$

---

$$\mathcal{R}_\epsilon^\square(a) := \left\{ x \in \mathcal{X}^\square \mid x = \alpha x' \wedge x' \in \mathcal{R}_\epsilon(a) \wedge \alpha \in [0,1] \right\} \cup \{\mathbf{0}\},$$

where $\mathbf{0}$ denotes the vector of all zeros in $\mathbb{R}^d$. We observe that, since the polytopes $\mathcal{R}_\epsilon(a)$ are constructed as the intersection of some halfspaces $\mathcal{H}_{ij}$ and $\mathcal{X}_\epsilon$, it is possible to easily build polytopes $\mathcal{R}_\epsilon^\square(a_i)$ by simply removing the normalization constraint $\sum_{\theta \in \Theta} x_\theta = 1$. Notice that, if $\mathcal{R}_\epsilon(a) = \varnothing$, then $\mathcal{R}_\epsilon^\square(a) = \{\mathbf{0}\}$, which implies that action $a$ is never induced as a best response, since $x^a = \mathbf{0}$.

After solving Program (2) for an optimal solution $x^\star := (x^{\star,a})_{a \in \mathcal{A}}$, Algorithm 3 employs such a solution to build a signaling scheme $\phi$. This employs a signal $s^a$ for every action $a \in A$, plus an additional signal $s^\star$, namely $\mathcal{S} := \{s^\star\} \cup \{s^a \mid a \in \mathcal{A}\}$. Specifically, the slice of $\phi$ with respect to $s^a$ is set to be equal to $x^a$, while its slice with respect to $s^\star$ is set so as to render $\phi$ a valid signaling scheme (*i.e.*, probabilities over signal sum to one for every $\theta \in \Theta$). Notice that this is always possible thanks to the additional constraints $\sum_{a \in \mathcal{A}} x_\theta^a \leq 1$ in Program (2). Moreover, such a slice may belong to $\mathcal{X}^\square \setminus \mathcal{X}_\epsilon^\square$. Indeed, in instances where there are some $\mathcal{X}^\square(a)$ falling completely outside $\mathcal{X}_\epsilon^\square$, this is fundamental to build a valid signaling scheme. Intuitively, one may think of $s^\star$ as incorporating all the "missing" signals in $\phi$, namely those corresponding to actions $a \in \mathcal{A}$ with $\mathcal{X}^\square(a)$ outside $\mathcal{X}_\epsilon^\square$.

The following lemma formally states the theoretical guarantees provided by Algorithm 3.

**Lemma 4.** *Given inputs $\mathcal{R}_\epsilon := \{\mathcal{R}_\epsilon(a)\}_{a \in \mathcal{A}}$, $\mathcal{X}_\epsilon$, and $\widehat{\mu}_t \in \Delta_\Theta$ for Algorithm 3, under events $\mathcal{E}_1$ and $\mathcal{E}_2$, the signaling scheme $\phi$ output by the algorithm is $\mathcal{O}(\epsilon n d + \nu)$-optimal for $\nu \leq \left| \sum_{\theta \in \Theta} \widehat{\mu}_{t,\theta} - \mu_\theta \right|$.*

In order to provide some intuition on how Lemma 4 is proved, let us assume that each polytope $\mathcal{X}_\epsilon(a)$ is either empty or has volume larger than zero, implying that $\mathcal{R}_\epsilon(a) = \mathcal{X}_\epsilon(a)$. In Appendix D, we provide the complete formal proof of Lemma 4, working even with zero-measure non-empty polytopes. The first observation the we need is that sender's expected utility under a signaling scheme $\phi$ can be decomposed across its slices, with each slice $x$ providing a utility of $\sum_{\theta \in \Theta} \mu_\theta x_\theta u_\theta^s(a(x))$. The second crucial observation is that there always exists an optimal signaling scheme $\phi^\star$ that is *direct* and *persuasive*, which means that $\phi^\star$ employs only one slice $x^a$ for each action $a \in \mathcal{A}$, with $a$ being a best response for the receiver under $x^a$. It is possible to show that the slices $x^a$ that also belong to $\mathcal{X}_\epsilon$ can be used to construct a feasible solution to Program 2, since $x^a \in \mathcal{R}_\epsilon^\square(a)$ by definition. Thus, restricted to those slices, the signaling scheme $\phi$ computed by Algorithm 3 achieves an approximate sender's expected utility that is greater than or equal to the one achieved by $\phi^\star$. Moreover, the loss due to dropping the slices that are *not* in $\mathcal{X}_\epsilon$ can be bounded thanks to point (ii) in Lemma 1. Finally, it remains to account for the approximation due to using $\widehat{\mu}_t$ instead of the true prior in the objective of Program 2. All the observations above allow to bound sender's expected utility loss as in Lemma 4.

# 5 Lower bounds for online Bayesian persuasion

In this section, we present two lower bounds on the regret attainable in the setting faced by Algorithm 1. The first lower bound shows that an exponential dependence in the number of states of nature $d$ and the number of receiver's actions $n$ is unavoidable. This shows that one cannot get rid of the binomial coefficient in the regret bound of Algorithm 1 provided in Theorem 1. Formally:

**Theorem 2.** *For any sender's algorithm, there exists a Bayesian persuasion instance in which $n = d + 2$ and the regret $R_T$ suffered by the algorithm is at least $2^{\Omega(d)}$, or, equivalently, $2^{\Omega(n)}$.*

Theorem 2 is proved by constructing a collection of Bayesian persuasion instances in which an optimal signaling scheme has to induce the receiver to take an action that is a best response only for a unique posterior belief (among those computable by the receiver at step (3) of the interaction). This posterior belief belongs to a set of possible candidates having size exponential in the number of states of nature $d$. As a result, in order to learn such a posterior belief, any algorithm has to commit to a number of signaling schemes that is exponential in $d$ (and, given how the instances are built, in $n$).

The second lower bound shows that the regret bound attained by Algorithm 1 is tight in $T$.

**Theorem 3.** *For any sender's algorithm, there exists a Bayesian persuasion instance in which the regret $R_T$ suffered by the algorithm is at least $\Omega(\sqrt{T})$.*

To prove Theorem 3, we construct two Bayesian persuasion instances with $\Theta = \{\theta_1, \theta_2\}$ such that, in the first instance, $\mu_{\theta_1}$ is slightly greater than $\mu_{\theta_2}$, while the opposite holds in the second instance. Furthermore, the two instances are built so that the sender does *not* gain any information that helps to distinguish between them by committing to signaling schemes. As a consequence, to make a distinction, the sender can only leverage the information gained by observing the states of nature realized at each round, and this clearly results in the regret being at least $\Omega(\sqrt{T})$.

# 6 The sample complexity of Bayesian persuasion

In this section, we show how the no-regret learning algorithm developed in Section 4 can be easily adapted to solve a related *Bayesian persuasion PAC-learning problem*. Specifically, given an (additive) approximation error $\gamma \in (0, 1)$ and a probability $\eta \in (0, 1)$, the goal of such a problem is to learn a $\gamma$-optimal signaling scheme with probability at least $1 - \eta$, by using the minimum possible number of rounds. This can be also referred to as the *sample complexity* of learning signaling schemes.

As in the regret-minimization problem addressed in Section 4, we assume that the sender does *not* know anything about both the prior distribution $\mu$ and receiver's utility function $u$.

We tackle the Bayesian persuasion PAC-learning problem with a suitable adaptation of Algorithm 1, provided in Algorithm 4. The first two phases of the algorithm follow the line of Algorithm 1, with the `Build-Search-Space` and `Find-Polytopes` procedures being called for suitably-defined parameters $\epsilon, \delta, T_1$, and $\zeta$ (taking different values with respect to their counterparts in Algorithm 1). In particular, the value of $\epsilon$ depends on $\gamma$ and is carefully computed so as to control the bit-complexity of numbers used in the `Find-Polytopes` procedure (see Lemma 3), as detailed in Appendix G. Finally, in its third phase,

---

**Algorithm 4** `PAC-Persuasion-w/o-Clue`

**Require:** $\gamma \in (0, 1), \eta \in (0, 1)$
1: $\delta \leftarrow \eta/2, \zeta \leftarrow \eta/2, \epsilon_1 \leftarrow \gamma/12nd, t \leftarrow 1$
2: $\epsilon \leftarrow$ `Compute-Epsilon`$(\epsilon_1)$
3: $T_1 \leftarrow \left\lceil \frac{1}{2\epsilon^2} \log\left(\frac{2d}{\delta}\right) \right\rceil$
4: $\mathcal{X}_\epsilon \leftarrow$ `Build-Search-Space`$(T_1, \epsilon)$
5: $\mathcal{R}_\epsilon \leftarrow$ `Find-Polytopes`$(\mathcal{X}_\epsilon, \zeta)$
6: $\phi \leftarrow$ `Compute-Signaling`$(\mathcal{R}_\epsilon, \mathcal{X}_\epsilon, \widehat{\mu})$
7: **return** $\phi$

---

the algorithm calls `Compute-Signaling` to compute a signaling scheme $\phi$ that can be proved to $\gamma$-optimal with probability at least $1 - \eta$.

The most relevant difference between Algorithm 4 and Algorithm 1 is the number of rounds used to build the prior estimate defining $\mathcal{X}_\epsilon$. Specifically, while the latter has to employ $T_1$ of the order of $1/\epsilon$ and rely on a multiplicative Chernoff bound to get tight regret guarantees, the former has to use $T_1$ of the order of $1/\epsilon^2$ and standard concentration inequalities to get an $\mathcal{O}(\epsilon)$-optimal solution. Formally:

**Lemma 5.** *Given $T_1 := \left\lceil \frac{1}{2\epsilon^2} \log\left(2d/\delta\right) \right\rceil$ and $\epsilon \in (0, 1)$, Algorithm 2 employs $T_1$ rounds and outputs $\mathcal{X}_\epsilon \subseteq \mathcal{X}$ such that, with probability at least $1 - \delta$: (i) $\sum_{\theta \in \Theta} \mu_\theta x_\theta \geq \epsilon$ for every slice $x \in \mathcal{X}_\epsilon$, (ii) $\sum_{\theta \in \Theta} \mu_\theta x_\theta \leq 6\epsilon$ for every slice $x \in \mathcal{X} \setminus \mathcal{X}_\epsilon$, and (iii) $|\widehat{\mu}_\theta - \mu_\theta| \leq \epsilon$ for every $\theta \in \Theta$.*

By Lemma 5, it is possible to show that the event $\mathcal{E}^1$ holds. Hence, the probability that a signaling scheme including a slice $x \in \mathcal{X}_\epsilon$ actually "induces" such a slice is at least $\epsilon$, and, thus, the results concerning the second phase of Algorithm 1 are valid also in this setting. Finally, whenever the events $\mathcal{E}_1$ and $\mathcal{E}_2$ hold, we can provide an upper bound on the number of rounds required by Algorithm 4 to compute a $\gamma$-optimal signaling scheme as desired. Formally:

**Theorem 4.** *Given $\gamma \in (0, 1)$ and $\eta \in (0, 1)$, with probability at least $1 - \eta$, Algorithm 4 outputs a $\gamma$-optimal signaling scheme in a number of rounds $T$ such that:*

$$T \leq \widetilde{\mathcal{O}}\left(\frac{n^3}{\gamma^2} \log^2\left(\frac{1}{\eta}\right)\left(d^8 B + d\binom{d+n}{d}\right)\right).$$

We conclude by providing two negative results showing that the result above is tight.

**Theorem 5.** *There exist two absolute constants $\kappa, \lambda > 0$ such that no algorithm is guaranteed to return a $\kappa$-optimal signaling scheme with probability of at least $1 - \lambda$ by employing less than $2^{\Omega(n)}$ and $2^{\Omega(d)}$ rounds, even when the prior distribution $\mu$ is known to the sender.*

**Theorem 6.** *Given $\gamma \in (0, 1/8)$ and $\eta \in (0, 1)$, no algorithm is guaranteed to return a $\gamma$-optimal signaling scheme with probability at least $1 - \eta$ by employing less than $\Omega\left(\frac{1}{\gamma^2}\log(1/\eta)\right)$ rounds.*

In Appendix H , we also study the case in which the prior $\mu$ is known to the sender. In such a case, we show that the sample complexity can be improved by a factor $1/\gamma$, which is tight.

## Acknowledgments

This work was supported by the Italian MIUR PRIN 2022 Project "Targeted Learning Dynamics: Computing Efficient and Fair Equilibria through No-Regret Algorithms", by the FAIR (Future

Artificial Intelligence Research) project, funded by the NextGenerationEU program within the PNRR-PE-AI scheme (M4C2, Investment 1.3, Line on Artificial Intelligence), and by the EU Horizon project ELIAS (European Lighthouse of AI for Sustainability, No. 101120237). This work was also partially supported by project SERICS (PE00000014) under the MUR National Recovery and Resilience Plan funded by the European Union - NextGenerationEU. Francesco Bacchiocchi was also supported by the G-Research November Grant.

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

# Appendix

The appendixes are organized as follows:

- Appendix A provides a discussion of the previous works most related to ours.
- Appendix B presents additional preliminaries.
- Appendix C presents the omitted proofs from Section 4.
- Appendix D presents the proof of Lemma 4 from Section 4.3.
- Appendix E provides a description of the results and the procedures employed in *phase 2*.
- Appendix F presents the omitted proofs from Section 5.
- Appendix G presents the omitted proofs and some technical details from Section 6.
- Appendix H discusses the PAC-learning problem when the prior is known.

## A   Additional Related Works

**Learning in Bayesian persuasion settings**   In addition to the works presented in Section 1.2, the problem of learning optimal signaling schemes in Bayesian persuasion settings has received growing attention over the last few years. Camara et al. [2020] study an adversarial setting where the receiver does not know the prior, and the receiver's behavior is aimed at minimizing internal regret. Babichenko et al. [2022] consider an online Bayesian persuasion setting with binary actions when the prior is known, and the receiver's utility function has some regularities. Feng et al. [2022] study the online Bayesian persuasion problem faced by a platform that observes some relevant information about the state of a product and repeatedly interacts with a population of myopic receivers through a recommendation mechanism. Agrawal et al. [2023] design a regret-minimization algorithm in an advertising setting based on the Bayesian persuasion framework, assuming that the receiver's utility function satisfies some regularity conditions.

**Online learning in problems with commitment**   From a technical point of view, our work is related to the problem of learning optimal strategies in Stackelberg games when the leader has no knowledge of the follower's utility. Letchford et al. [2009] propose the first algorithm to learn optimal strategies in Stackelberg games. Their algorithm is based on an initial random sampling that may require an exponential number of samples, both in the number of leader's actions $m$ and in the representation precision $L$. Peng et al. [2019] improve the algorithm of Letchford et al. [2009], while Bacchiocchi et al. [2024a] further improve the approach by Peng et al. [2019] by relaxing some of their assumptions.

Furthermore, our work is also related to the problem of learning optimal strategies in Stackelberg games where the leader and the follower interaction is modelled by a Markov Decision Process. Lauffer et al. [2024] study Stackelberg games with a state that influences the leader's utility and available actions. Bai et al. [2021] consider a setting where the leader commits to a pure strategy and observes a noisy measurement of their utility.

Finally, our work is also related to online hidden-action principal-agent problems, in which a principal commits to a contract at each round to induce an agent to take favorable actions. Ho et al. [2014] initiated the study by proposing an algorithm that adaptively refines a discretization over the space of contracts, framing the model as a multi-armed bandit problem where the discretization provides a finite number of arms to play with. Cohen et al. [2022] similarly work in a discretized space but with milder assumptions. Zhu et al. [2023] provide a more general algorithm that works in hidden-action principal-agent problems with multiple agent types. Finally, Bacchiocchi et al. [2024b] study the same setting and propose an algorithm with smaller regret when the number of agent actions is small.

## B   Additional preliminaries

### B.1   Additional preliminaries on Bayesian persuasion

In step (3) of the sender-receiver interaction presented in Section 2.1, after observing $s \in \mathcal{S}$, the receiver performs a Bayesian update and infers a posterior belief $\xi^s \in \Delta_\Theta$ over the states of nature,

according to the following equation:

$$\xi_\theta^s = \frac{\mu_\theta \, \phi_\theta(s)}{\sum_{\theta' \in \Theta} \mu_{\theta'} \, \phi_{\theta'}(s)} \qquad \forall \theta \in \Theta. \tag{3}$$

Consequently, given a signaling scheme $\phi$, we can equivalently represent it as a distribution over the set of posteriors it induces. Formally, we say that $\phi$ induces $\gamma : \Delta_\Theta \to [0, 1]$ if, for each posterior distribution $\xi \in \Delta_\Theta$, we have:

$$\gamma(\xi) = \sum_{s \in \mathcal{S} : \xi^s = \xi} \sum_{\theta \in \Theta} \mu_\theta \phi_\theta(s) \qquad \text{and} \qquad \sum_{\xi \in \text{supp}(\gamma)} \gamma(\xi) = 1.$$

Furthermore, we say that a distribution over a set of posteriors $\gamma$ is consistent, *i.e.*, there exists a valid signaling scheme $\phi$ inducing $\gamma$ if the following holds:

$$\sum_{\xi \in \text{supp}(\gamma)} \gamma(\xi)\xi_\theta = \mu_\theta \quad \forall \theta \in \Theta.$$

With an abuse of notation, we will sometimes refer to a consistent distribution over a set of posteriors $\gamma$ as a signaling scheme. This is justified by the fact that there exists a signaling scheme $\phi$ inducing such distribution, but we are interested only in the distribution over the set of posteriors that $\phi$ induces.

## B.2 Additional preliminaries on the representation of numbers

In the following, we assume that all the numbers manipulated by our algorithms are rational. Furthermore, we assume that rational numbers are represented as fractions, by specifying two integers which encode their numerator and denominator. Given a rational number $q \in \mathbb{Q}$ represented as a fraction $b/c$ with $b, c \in \mathbb{Z}$, we denote the number of bits that $q$ occupies in memory, called *bit-complexity*, as $B_q := B_b + B_c$, where $B_b$ ($B_c$) is the number of bits required to represent the numerator (denominator). For the sake of the presentation, with an abuse of terminology, given a vector in $\mathbb{Q}^D$ of $D$ rational numbers represented as fractions, we let its bit-complexity be the maximum bit-complexity among its entries.

Furthermore, we assume that the bit-complexity encoding both the receiver's utility and the prior distribution is bounded. Formally, we denote by $B_\mu$ the bit-complexity of the prior $\mu$, while we assume $B_u$ to be an upper bound to the bit-complexity of each $d$-dimensional vector $u_\theta(a)$ with $\theta \in \Theta$. Moreover, we let $B := B_\mu + B_u$. Finally, we also denote with $B_\epsilon$ the bit-complexity of the parameter $\epsilon$ computed by our algorithms, while we denote with $B_{\widehat{\mu}}$ the bit-complexity of the estimator $\widehat{\mu}$ computed by Algorithm 2.

## C Omitted proofs from Section 4

**Lemma 1.** *Given* $T_1 := \left\lceil \frac{12}{\epsilon} \log \left(2d/\delta\right) \right\rceil$ *and* $\epsilon \in (0, 1/6d)$, *Algorithm 2 employs* $T_1$ *rounds and terminates with a set* $\mathcal{X}_\epsilon \subseteq \mathcal{X}$ *such that, with probability at least* $1 - \delta$: *(i)* $\sum_{\theta \in \Theta} \mu_\theta x_\theta \geq \epsilon$ *for every slice* $x \in \mathcal{X}_\epsilon$ *and (ii)* $\sum_{\theta \in \Theta} \mu_\theta x_\theta \leq 10\epsilon$ *for every slice* $x \in \mathcal{X} \setminus \mathcal{X}_\epsilon$.

*Proof.* For each $\theta \in \Theta$ we consider two possible cases.

1. If $\mu_\theta > \epsilon$, then we employ the multiplicative Chernoff inequality as follows:

$$\mathbb{P}\left(|\mu_\theta - \widehat{\mu}_\theta| \geq \frac{1}{2}\mu_\theta\right) \leq 2e^{-\frac{T_1 \mu_\theta}{12}},$$

where $T_1 \in \mathbb{N}_+$ is the number of rounds employed to estimate $\widehat{\mu}_\theta$. As a result, by setting the number of rounds to estimate $\mu_\theta$ equal to $T_1 = \lceil 12/\epsilon \log (2d/\delta) \rceil$ we get:

$$\mathbb{P}\left(|\mu_\theta - \widehat{\mu}_\theta| \geq \frac{1}{2}\mu_\theta\right) \leq 2\left(\frac{\delta}{2d}\right)^{\frac{\mu_\theta}{\epsilon}} \leq \frac{\delta}{d},$$

since $\mu_\theta > \epsilon$. Then, we get:

$$\mathbb{P}\left(\frac{\mu_\theta}{2} \leq \widehat{\mu}_\theta \leq \frac{3\mu_\theta}{2}\right) \geq 1 - \frac{\delta}{d}. \tag{4}$$

Consequently, with a probability of at least $1 - \delta/d$ the estimator $\widehat{\mu}_\theta$ is such that $\widehat{\mu}_\theta \in [\mu_\theta/2, 3\mu_\theta/2]$. Thus, if $\mu_\theta \geq 6\epsilon$, then $\widehat{\mu}_\theta \geq 3\epsilon > 2\epsilon$ and $\theta \in \widetilde{\Theta}$. We also notice that there always exits a $\theta \in \Theta$ such that $\mu_\theta \geq 1/d \geq 6\epsilon$, since $\epsilon \in (0, 1/6d)$. Consequently, with a probability of at least $1 - \delta/d$, there always exist a $\theta \in \widetilde{\Theta}$.

2. If $\mu_\theta \leq \epsilon$, then we employ the multiplicative Chernoff inequality as follows:

$$\mathbb{P}\left(\widehat{\mu}_\theta \geq (1 + c)\mu_\theta\right) \leq e^{-\frac{c^2 T_1 \mu_\theta}{2+c}},$$

with $c = \epsilon/\mu_\theta$. Thus, by setting the number of rounds employed to estimate $\mu_\theta$ equal to $T_1 = \lceil 12/\epsilon \log(2d/\delta) \rceil$, we get:

$$\mathbb{P}\left(\widehat{\mu}_\theta \geq \mu_\theta + \epsilon\right) \leq \exp\left(-\frac{\frac{\epsilon^2}{\mu_\theta^2}\left(\frac{12}{\epsilon}\log\left(\frac{2d}{\delta}\right)\right)\mu_\theta}{2 + \frac{\epsilon}{\mu_\theta}}\right)$$

$$\leq \exp\left(-\frac{\frac{12\epsilon}{\mu_\theta}\log\left(\frac{2d}{\delta}\right)}{2 + \frac{\epsilon}{\mu_\theta}}\right)$$

$$\leq \left(\frac{\delta}{2d}\right)^4 \leq \frac{\delta}{d},$$

since $x/(x+2) \geq 1/3$, for each $x \geq 1$. As a result, we have:

$$\mathbb{P}\left(\widehat{\mu}_\theta \leq \mu_\theta + \epsilon\right) \geq 1 - \frac{\delta}{d}.$$

Thus, with a probability of at least $1 - \delta/d$, if $\mu_\theta \leq \epsilon$, then $\widehat{\mu}_\theta \leq \mu_\theta + \epsilon$, which implies that $\widehat{\mu}_\theta \leq 2\epsilon$. Furthermore, if $\mu_\theta \leq \epsilon$, then $\widehat{\mu}_\theta \leq 2\epsilon$ and $\theta \notin \widetilde{\Theta}$.

Thus, by employing a union bound over the set of natures, we have that if $\mu_\theta \leq \epsilon$, then its corresponding estimate $\widehat{\mu}_\theta$ falls within the interval $[0, 2\epsilon]$ and $\theta \notin \widetilde{\Theta}$, while if $\mu_\theta \geq 6\epsilon$, then its corresponding estimate is such that $\widehat{\mu}_\theta > 2\epsilon$ and $\theta \in \widetilde{\Theta}$, with a probability of at least $1 - \delta$. We also notice that, with the same probability, the set $\widetilde{\Theta}$ is always non empty.

Consequently, for each slice $x \in \mathcal{X}_\epsilon$ with respect to a signal $s$, the probability of observing $s$ can be lower bounded as follows:

$$\epsilon \leq \frac{1}{2}\sum_{\theta \in \widetilde{\Theta}}\widehat{\mu}_\theta x_\theta \leq \frac{3}{4}\sum_{\theta \in \widetilde{\Theta}}\mu_\theta x_\theta \leq \sum_{\theta \in \widetilde{\Theta}}\mu_\theta x_\theta \leq \sum_{\theta \in \Theta}\mu_\theta x_\theta,$$

where the inequalities above hold because of the definition of $\mathcal{X}_\epsilon$ and observing that each $\theta \in \widetilde{\Theta}$ satisfies Equation 4 with probability at least $1 - \delta$.

Furthermore, for each $x \notin \mathcal{X}_\epsilon$, the two following conditions hold:

$$\frac{1}{2}\sum_{\theta \in \widetilde{\Theta}}\mu_\theta x_\theta \leq \sum_{\theta \in \widetilde{\Theta}}\widehat{\mu}_\theta x_\theta \leq 2\epsilon \quad \text{and} \quad \sum_{\theta \notin \widetilde{\Theta}}\mu_\theta x_\theta \leq 6\epsilon\sum_{\theta \notin \widetilde{\Theta}}x_\theta \leq 6\epsilon,$$

with probability at least $1 - \delta$. Thus, by putting the two inequalities above together, for each $x \notin \mathcal{X}_\epsilon$, we have:

$$\sum_{\theta \in \Theta}\mu_\theta x_\theta \leq 10\epsilon,$$

with probability at least $1 - \delta$, concluding the proof. $\qquad\square$

**Lemma 2.** *Under event $\mathcal{E}^1$, given any $\rho \in (0,1)$ and a normalized slice $x \in \mathcal{X}_\epsilon$, if the sender commits to a signaling scheme $\phi : \Theta \to \mathcal{S} := \{s_1, s_2\}$ such that $\phi_\theta(s_1) = x_\theta$ for all $\theta \in \Theta$ during $q := \lceil \frac{1}{\epsilon}\log(1/\rho) \rceil$ rounds, then, with probability at least $1 - \rho$, signal $s_1$ is sent at least once.*

*Proof.* In the following, we let $\tau$ be the first round in which the sender commits to $\phi$. The probability of observing the signal $s_1$ at a given round $t \geq \tau$ is can be lower bounded as follows:

$$\mathbb{P}\left(s^t = s_1\right) = \sum_{\theta \in \Theta} \mu_\theta x_\theta \geq \epsilon,$$

where the inequality holds under the event $\mathcal{E}_1$. Thus, at each round, the probability of sampling the signal $s_1 \in \mathcal{S}$ is greater or equal to $\epsilon > 0$. Consequently, the probability of never observing the signal $s_1 \in \mathcal{S}$ in $q$ rounds is given by:

$$\mathbb{P}\left(\bigcap_{t=\tau}^{\tau+q-1} \{s^t \neq s_1\}\right) \leq (1 - \epsilon)^q \leq \rho,$$

where the last inequality holds by taking $q = \left\lceil \frac{\log(\rho)}{\log(1-\epsilon)} \right\rceil \leq \left\lceil \frac{\log(1/\rho)}{\epsilon} \right\rceil$, for each $\epsilon \in (0, 1)$.

As a result, the probability of observing the signal $s_1$ at least once in $q$ rounds is greater or equal to:

$$\mathbb{P}\left(\bigcup_{t=\tau}^{\tau+q-1} \{s^t \neq s_1\}\right) = 1 - \mathbb{P}\left(\bigcap_{t=\tau}^{\tau+q-1} \{s^t \neq s_1\}\right) \geq 1 - \rho,$$

concluding the proof. $\qquad\square$

**Theorem 1.** *The regret attained by Algorithm 1 is* $R_T \leq \widetilde{\mathcal{O}}\left(\binom{d+n}{d} n^{3/2} d^3 \sqrt{BT}\right)$.

*Proof.* In the following, we let $\delta = \zeta = \frac{1}{T}$ and $\epsilon = \frac{\lceil \sqrt{Bn} d^4 \rceil}{\lceil \sqrt{T} \rceil}$, as defined in Algorithm 1. To prove the theorem, we decompose the regret suffered in the three phases of Algorithm 1:

1. Phase 1. We observe that the number of rounds to execute the `Build-Search-Space` procedure (Algorithm 2) is equal to $T_1 = \mathcal{O}\left(1/\epsilon \log(1/\delta) \log(d)\right)$. Thus, the cumulative regret of Phase 1 can be upper bounded as follows:

$$R_T^1 \leq \widetilde{\mathcal{O}}\left(\frac{1}{\epsilon} \log(T) \log(d)\right) \leq \widetilde{\mathcal{O}}\left(\frac{\sqrt{T}}{d^4 \sqrt{nB}} \log(d)\right) \leq \widetilde{\mathcal{O}}\left(\sqrt{T}\right).$$

   This is because, at each round, the regret suffered during the execution of Algorithm 2 is at most one.

2. Phase 2. Under the event $\mathcal{E}_1$, which holds with probability $1 - \delta$, Algorithm 6 correctly terminates with probability $1 - \zeta$. Thus, with probability at least $1 - \delta - \zeta$, the number of rounds employed by such algorithm is of the order:

$$T_2 \leq \widetilde{\mathcal{O}}\left(\frac{n^2}{\epsilon} \log^2\left(\frac{1}{\zeta}\right)\left(d^7(B + B_\epsilon + B_{\widehat{\mu}}) + \binom{d+n}{d}\right)\right)$$
$$= \widetilde{\mathcal{O}}\left(\frac{n^2}{\epsilon} \log^2(T)\left(d^7(B + B_\epsilon) + \binom{d+n}{d}\right)\right),$$

   where the last equality holds because $B_{\widehat{\mu}} = \mathcal{O}(\log(1/\epsilon) + \log(d) + \log(T))$. As a result, by taking the expectation, the regret suffered in Phase 2 by Algorithm 1 can be upper bounded as follows:

$$R_T^2 \leq \widetilde{\mathcal{O}}\left(n^{3/2} d^3 \binom{d+n}{d} \sqrt{BT}\right),$$

   since, at each round, the regret suffered during the execution of Algorithm 6 is at most one.

3. Phase 3. Let $\tau$ be the number of rounds required by Phase 1 and Phase 2 to terminate. Under the events $\mathcal{E}_1$ and $\mathcal{E}_2$, which hold with probability at least $1 - \delta - \zeta$, thanks to Lemma 4, the solution returned by Algorithm 3 at each round $t > \tau$ is $\mathcal{O}(dn\epsilon + \nu_t)$-optimal, where we define $\nu_t = \left|\sum_{\theta \in \Theta} \mu_\theta - \widehat{\mu}_{t,\theta}\right|$. We introduce the following event:

$$E_t = \{|\mu_\theta - \widehat{\mu}_{t,\theta}| \leq \epsilon_t \quad \forall \theta \in \Theta\},$$

where we let $\epsilon_t > 0$ be defined as follows:

$$\epsilon_t = \sqrt{\frac{\log\left(2dT/\iota\right)}{2(t-\tau)}}, \quad t > \tau.$$

Then, by Hoeffding's inequality and a union bound we have:

$$\mathbb{P}\Big(\bigcap_{t>\tau} E_t\Big) \geq 1 - \iota.$$

Thus, by setting $\iota = 1/T$, the regret suffered in Phase 3 by Algorithm 1 can be upper bounded as follows:

$$
\begin{aligned}
R_T^3 &\leq \sum_{t=\tau+1}^{T} \left|\sum_{\theta\in\Theta} \mu_\theta - \widehat{\mu}_{t,\theta}\right| + \mathcal{O}\left(d\epsilon T\right) \\
&\leq d \sum_{t=\tau+1}^{T} \epsilon_t + \mathcal{O}\left(dn\epsilon T\right) \\
&\leq \widetilde{\mathcal{O}}\left(d\sqrt{T} + dn\epsilon T\right) \\
&= \widetilde{\mathcal{O}}\left(d\sqrt{T} + n^{3/2}d^5\sqrt{BT}\right).
\end{aligned}
$$

As a result, the regret of Algorithm 1 is in the order of:

$$R_T \leq \widetilde{\mathcal{O}}\left(n^{3/2}d^3\binom{d+n}{d}\sqrt{BT} + n^{3/2}d^5\sqrt{BT}\right) = \widetilde{\mathcal{O}}\left(n^{3/2}d^3\binom{d+n}{d}\sqrt{BT}\right),$$

concluding the proof. $\qquad\square$

## D  Proof of Lemma 4 from Section 4.3

In order to prove Lemma 4, we first consider an auxiliary LP (Program 5a) that works on the vertices of the regions $\mathcal{X}_\epsilon(a)$. This is useful to take into account the polytopes $\mathcal{X}_\epsilon(a)$ with null volume. Indeed, for every action $a \in \mathcal{A}$ such that $\mathrm{vol}(\mathcal{X}_\epsilon(a)) = 0$, Algorithm 3 takes in input only a face $\mathcal{R}_\epsilon(a)$ of $\mathcal{X}_\epsilon(a)$ such that $\mathcal{V}_\epsilon(a) \subseteq V(\mathcal{R}_\epsilon(a))$. By working on the vertices of the regions $\mathcal{X}_\epsilon(a)$, we can show that the vertices in $\mathcal{V}_\epsilon(a)$ are sufficient to compute an approximately optimal signaling scheme.

The auxiliary LP that works on the vertices is the following:

$$
\max_{\alpha\geq\mathbf{0}} \quad \sum_{x\in\mathcal{V}_\epsilon} \alpha_x \sum_{\theta\in\Theta} \mu_\theta x_\theta u_\theta^{\mathrm{s}}(a(x)) \tag{5a}
$$

$$
\text{s.t.} \quad \sum_{x\in\mathcal{V}_\epsilon} \alpha_x x_\theta \leq 1 \quad \forall\theta\in\Theta. \tag{5b}
$$

Program 5a takes in input the set of vertices $\mathcal{V}_\epsilon := \bigcup_{a\in\mathcal{A}} V(\mathcal{X}_\epsilon(a))$, along with the corresponding best-responses $(a(x))_{x\in\mathcal{V}_\epsilon}$ and the exact prior $\mu$. It then optimizes over the non-negative variables $\alpha_x \geq 0$, one for vertex $x \in \mathcal{V}_\epsilon$. These variables $\alpha_x$ act as weights for the corresponding slices $x \in \mathcal{V}_\epsilon$, identifying a non-normalized slice $\alpha_x x \in \mathcal{X}^{\square}$.

In the following, we show that the value of an optimal solution $v^\star$ to Program 5a is at least $v^\star \geq \mathrm{OPT} - \mathcal{O}(\epsilon nd)$. Then, we prove that the signaling scheme $\phi$ computed by Algorithm 3 achieves a principal's expected utility of at least $v^\star$ minus a quantity related to the difference between the estimated prior $\widehat{\mu}_t$ and with the actual prior $\mu$. Thus, by considering that $v^\star \geq \mathrm{OPT} - \mathcal{O}(\epsilon nd)$, we will be able to prove Lemma 4.

As a first step, we show that it is possible to decompose each slice $x \in \mathcal{X}_\epsilon$ into a weighted sum of the vertices $x' \in \mathcal{V}_\epsilon$ without incurring a loss in the sender's utility. Thus, a generic slice $x$ of an optimal signaling scheme can be written as a convex combination of slices $x'$ with $x' \in \mathcal{V}_\epsilon$. This property is formalized in the following lemma.

**Lemma 6.** *For every $x \in \mathcal{X}_\epsilon$, there exists a distribution $\alpha \in \Delta_{\mathcal{V}_\epsilon}$ such that:*

$$x_\theta = \sum_{x' \in \mathcal{V}_\epsilon} \alpha_{x'} x'_\theta \quad \forall \theta \in \Theta.$$

*Furthermore, the following holds:*

$$\sum_{\theta \in \Theta} \mu_\theta x_\theta u_\theta^{\mathrm{s}}(a(x)) \leq \sum_{x' \in \mathcal{V}_\epsilon} \alpha_{x'} \sum_{\theta \in \Theta} \mu_\theta x'_\theta u_\theta^{\mathrm{s}}(a(x')).$$

*Proof.* Let $a := a(x)$. Since $x \in \mathcal{X}_\epsilon(a)$, by the Carathéodory theorem, there exists an $\alpha \in \Delta_{V(\mathcal{X}_\epsilon(a))}$ such that:

$$\sum_{x' \in V(\mathcal{X}_\epsilon(a))} \alpha_{x'} x'_\theta = x_\theta \quad \forall \theta \in \Theta.$$

Furthermore:

$$\sum_{\theta \in \Theta} \mu_\theta x_\theta u_\theta^{\mathrm{s}}(a) = \sum_{\theta \in \Theta} \mu_\theta \left( \sum_{x' \in V(\mathcal{X}_\epsilon(a))} \alpha_{x'} x'_\theta \right) u_\theta^{\mathrm{s}}(a).$$

$$= \sum_{x' \in V(\mathcal{X}_\epsilon(a))} \alpha_{x'} \sum_{\theta \in \Theta} \mu_\theta x'_\theta u_\theta^{\mathrm{s}}(a)$$

$$\leq \sum_{x' \in V(\mathcal{X}_\epsilon(a))} \alpha_{x'} \sum_{\theta \in \Theta} \mu_\theta x'_\theta u_\theta^{\mathrm{s}}(a(x'))$$

where the inequality holds because the receiver breaks ties in favor of the sender. Finally, we observe that for each distribution over the set $V(\mathcal{X}_\epsilon(a))$ for a given $\mathcal{X}_\epsilon(a)$, we can always recover a probability distribution supported in $\mathcal{V}_\epsilon$, since $V(\mathcal{X}_\epsilon(a)) \subseteq \mathcal{V}_\epsilon$ by construction. $\square$

Thanks to the the result above, in the next lemma (Lemma 7) we prove that an optimal solution of Program 5a has value at least $v^\star \geq \mathrm{OPT} - 10\epsilon nd$. To show this, we begin by observing that there exists a set $\mathcal{J}$ of slices of the optimal signaling scheme that belong to the search space $\mathcal{X}_\epsilon$. By applying Lemma 6 to each of these slices, we obtain a feasible solution for Program 5a. The value of this solution is at least the sender's expected utility given by the slices in $\mathcal{J}$. Finally, thanks to the properties of the search space $\mathcal{X}_\epsilon$, we can bound the expected sender's utility provided by the slices that lie outside the search space.

**Lemma 7.** *Under the event $\mathcal{E}_1$, the optimal solution of Program 5a has value at least $v^\star \geq \mathrm{OPT} - 10\epsilon nd$.*

*Proof.* In the following we let $\phi_\theta \in \Delta_\mathcal{A}$ for each $\theta \in \Theta$ be an optimal signaling scheme, where we assume, without loss of generality, such a signaling scheme to be direct, meaning that $\mathcal{S} = \mathcal{A}$ and $a \in \mathcal{A}^\phi(a)$ for every action $a \in \mathcal{A}$. Furthermore, for each action $a \in \mathrm{supp}(\phi)$, we define:

$$x_\theta^a = \frac{\phi_\theta(a)}{\sum_{\theta \in \Theta} \phi_\theta(a)} \quad \forall \theta \in \Theta \quad \text{and} \quad \alpha_{x^a} = \sum_{\theta \in \Theta} \phi_\theta(a).$$

We observe that each $x^a \in \mathcal{X}(a)$, indeed we have:

$$\sum_{\theta \in \Theta} \mu_\theta x_\theta u_\theta(a) = \frac{1}{\alpha_{x^a}} \sum_{\theta \in \Theta} \mu_\theta \phi_\theta(a) u_\theta(a) \geq \frac{1}{\alpha_{x^a}} \sum_{\theta \in \Theta} \mu_\theta \phi_\theta(a) u_\theta(a') = \sum_{\theta \in \Theta} \mu_\theta x_\theta u_\theta(a').$$

for every action $a' \in \mathcal{A}$. We define the subset of actions $\mathcal{A}' \subseteq \mathcal{A}$ in a way that if $a \in \mathcal{A}'$, then $x^a \in \mathcal{X}_\epsilon$, *i.e.*, $\mathcal{A}' := \{a \in \mathcal{A} \mid x^a \in \mathcal{X}_\epsilon\}$. Then, we let $\mathcal{A}'' := \mathrm{supp}(\phi) \setminus \mathcal{A}'$. Furthermore, for each $a \in \mathcal{A}'$, thanks to Lemma 6, there exists a distribution $\alpha^a \in \Delta_{\mathcal{V}_\epsilon}$ such that:

$$x_\theta^a = \sum_{x' \in \mathcal{V}_\epsilon} \alpha_{x'}^a x'_\theta,$$

and the following holds:

$$\sum_{\theta \in \Theta} \mu_\theta x_\theta^a u_\theta(a) = \sum_{\theta \in \Theta} \mu_\theta \left( \sum_{x' \in \mathcal{V}_\epsilon} \alpha_{x'}^a x_\theta' \right) u_\theta^{\mathrm{s}}(a).$$

$$= \sum_{\theta \in \Theta} \sum_{x' \in \mathcal{V}_\epsilon} \mu_\theta \alpha_{x'}^a x_\theta' u_\theta^{\mathrm{s}}(a)$$

$$\leq \sum_{\theta \in \Theta} \sum_{x' \in \mathcal{V}_\epsilon} \mu_\theta \alpha_{x'}^a x_\theta' u_\theta^{\mathrm{s}}(a(x')). \tag{6}$$

We also define $\alpha^\star : \mathcal{V}_\epsilon \to \mathbb{R}_+$ as follows:

$$\alpha_{x'}^\star = \sum_{a \in \mathcal{A}'} \alpha_{x'}^a \alpha_{x^a},$$

for each $x' \in \mathcal{V}_\epsilon$. First, we show that $\alpha^\star : \mathcal{V}_\epsilon \to \mathbb{R}_+$ is a feasible solution to LP 5a. Indeed, for each $\theta \in \Theta$, it holds:

$$\sum_{x' \in \mathcal{V}_\epsilon} \alpha_{x'}^\star x_\theta' = \sum_{x' \in \mathcal{V}_\epsilon} \sum_{a \in \mathcal{A}'} \alpha_{x'}^a \alpha_{x^a} x_\theta'$$

$$= \sum_{a \in \mathcal{A}'} \alpha_{x^a} \sum_{x' \in \mathcal{V}_\epsilon} \alpha_{x'}^a x_\theta'$$

$$= \sum_{a \in \mathcal{A}'} \alpha_{x^a} x_\theta^a$$

$$= \sum_{a \in \mathcal{A}'} \phi_\theta(a) \leq 1$$

Where the equalities above holds thanks to Equation 6 and the definition of $\alpha^\star$. Then, we show that the utility achieved by $\alpha^\star : \mathcal{V}_\epsilon \to \mathbb{R}_+^m$ is greater or equal to $\mathrm{OPT} - 10\epsilon d$. Formally, we have:

$$\sum_{x' \in \mathcal{V}_\epsilon} \alpha_{x'}^\star \sum_{\theta \in \Theta} \mu_\theta x_\theta' u_\theta^{\mathrm{s}}(a(x')) = \sum_{x' \in \mathcal{V}_\epsilon} \sum_{a \in \mathcal{A}'} \alpha_{x^a} \alpha_{x'}^a \sum_{\theta \in \Theta} \mu_\theta x_\theta' u_\theta^{\mathrm{s}}(a(x'))$$

$$= \sum_{a \in \mathcal{A}'} \alpha_{x^a} \sum_{x' \in \mathcal{V}_\epsilon} \sum_{\theta \in \Theta} \mu_\theta \alpha_{x'}^a x_\theta' u_\theta^{\mathrm{s}}(a(x'))$$

$$\geq \sum_{a \in \mathcal{A}'} \alpha_{x^a} \sum_{\theta \in \Theta} x_\theta^a \mu_\theta u_\theta^{\mathrm{s}}(a)$$

$$= \sum_{a \in \mathcal{A}'} \sum_{\theta \in \Theta} \mu_\theta \phi_\theta(a) u_\theta^{\mathrm{s}}(a)$$

$$= \mathrm{OPT} - \sum_{a \in \mathcal{A}''} \sum_{\theta \in \Theta} \mu_\theta \phi_\theta(a) u_\theta^{\mathrm{s}}(a)$$

$$= \mathrm{OPT} - \sum_{a \in \mathcal{A}''} \sum_{\theta \in \Theta} \mu_\theta \alpha_{x^a} x_\theta^a u_\theta^{\mathrm{s}}(a)$$

$$\geq \mathrm{OPT} - d \sum_{a \in \mathcal{A}''} \sum_{\theta \in \Theta} \mu_\theta x_\theta^a u_\theta^{\mathrm{s}}(a)$$

$$\geq \mathrm{OPT} - 10\epsilon nd.$$

Where the first inequality holds thanks to Inequality (6), the second inequality holds since $\alpha_{x^a} \leq d$ and the last inequality holds since, for each $x \notin \mathcal{X}_\epsilon$, it holds $\sum_{\theta \in \Theta} \mu_\theta x_\theta^a(a) \leq 10\epsilon$, under the event $\mathcal{E}_1$ and $x^a \notin \mathcal{X}_\epsilon$ for every $a \in \mathcal{A}''$. Consequently, $\alpha^\star$ is a feasible solution to LP 5a and provides, under the event $\mathcal{E}_1$, a value of at least $\mathrm{OPT} - 10\epsilon nd$. As a result, under the event $\mathcal{E}_1$ the optimal solution of LP 5a has value $v^\star \geq \mathrm{OPT} - 10\epsilon nd$, concluding the proof. $\qquad \square$

**Lemma 4.** *Given inputs $\mathcal{R}_\epsilon := \{\mathcal{R}_\epsilon(a)\}_{a \in \mathcal{A}}$, $\mathcal{X}_\epsilon$, and $\widehat{\mu}_t \in \Delta_\Theta$ for Algorithm 3, under events $\mathcal{E}_1$ and $\mathcal{E}_2$, the signaling scheme $\phi$ output by the algorithm is $\mathcal{O}(\epsilon nd + \nu)$-optimal for $\nu \leq \left| \sum_{\theta \in \Theta} \widehat{\mu}_{t,\theta} - \mu_\theta \right|$.*

*Proof.* We observe that under the events $\mathcal{E}_1$ and $\mathcal{E}_2$, the collection $\mathcal{R}_\epsilon = \{\mathcal{R}_\epsilon(a)\}_{a \in \mathcal{A}}$ is composed of faces $\mathcal{R}_\epsilon(a)$ of $\mathcal{X}_\epsilon(a)$ (possibly the improper face $\mathcal{X}\epsilon(a)$ itself) such that every vertex $x \in V(\mathcal{X}_\epsilon(a))$ that satisfies $a(x) = a$ belongs to $\mathcal{R}_\epsilon(a)$. The following statements hold under these two events.

As a first step, we prove that given a feasible solution $\alpha = (\alpha_x)_{x \in \mathcal{V}_\epsilon}$ to Program 5a, one can construct a feasible solution $\varphi$ to Program 2 with the same value. In particular, we consider a solution $\varphi = (\widetilde{x}^a)_{a \in \mathcal{A}}$ defined as:

$$\widetilde{x}^a_\theta := \sum_{x \in \mathcal{V}_\epsilon(a)} \alpha_x x_\theta \quad \forall a \in \mathcal{A}, \theta \in \Theta.$$

We observe that for every $a \in \mathcal{A}$ and $\theta \in \Theta$ we can bound $\widetilde{x}^a_\theta$ as follows:

$$0 \leq \widetilde{x}^a_\theta = \sum_{x \in \mathcal{V}_\epsilon(a)} \alpha_x x_\theta \leq \sum_{x \in \mathcal{V}_\epsilon} \alpha_x x_\theta \leq 1,$$

where the last inequality holds due to the constraints of Program 5a. Consequently, the vectors $\widetilde{x}^a$ belong to $\mathcal{X}^\square$.

Now we show that $\widetilde{x}^a$ belongs to $\mathcal{R}^\square_\epsilon(a)$ for every $a \in \mathcal{A}$. This holds trivially for every action $a \in \mathcal{A}$ such that $\sum_{x' \in \mathcal{V}_\epsilon(a)} \alpha_{x'} = 0$, as $\widetilde{x}^a = \mathbf{0} \in \mathcal{R}^\square_\epsilon(a)$. Consider instead an action $a \in \mathcal{A}$ such that $\sum_{x' \in \mathcal{V}_\epsilon(a)} \alpha_{x'} > 0$, and let us define the coefficient:

$$\beta^a_x := \frac{\alpha_x}{\sum_{x' \in \mathcal{V}_\epsilon(a)} \alpha_{x'}},$$

for every vertex $x \in \mathcal{V}_\epsilon(a)$. One can easily verify that $\beta^a \in \Delta_{\mathcal{V}_\epsilon(a)}$. Now consider the normalized slice:

$$\widetilde{x}^{\mathrm{N},a} := \sum_{x \in \mathcal{V}_\epsilon(a)} \beta^a_x x.$$

This slice belongs to $\mathcal{R}^\triangle_\epsilon(a) := \mathcal{R}_\epsilon(a)$, as it is the weighted sum of the vertices $\mathcal{V}_\epsilon(a) \subseteq V(\mathcal{R}^\triangle_\epsilon(a))$ with weights $\beta^a \in \Delta_{\mathcal{V}_\epsilon(a)}$. Furthermore, we can rewrite the component $\widetilde{x}^a$ of the solution $\phi$ as:

$$\widetilde{x}^a = \widetilde{x}^{\mathrm{N},a} \sum_{x \in \mathcal{V}_\epsilon(a)} \alpha_x.$$

Thus, by considering that $\widetilde{x}^{\mathrm{N},a} \in \mathcal{R}^\triangle_\epsilon(a)$ and $\widetilde{x}^a \in \mathcal{X}^\square$, we have that $\widetilde{x}^a \in \mathcal{R}^\square_\epsilon(a)$.

Finally, since $\alpha$ is a feasible solution for Program 5a, we can observe that:

$$\sum_{a \in \mathcal{A}} \widetilde{x}^a_\theta = \sum_{a \in \mathcal{A}} \sum_{x \in \mathcal{V}_\epsilon(a)} \alpha_x x = \sum_{x \in \mathcal{V}_\epsilon} \alpha_x x \leq 1.$$

As a result, $\varphi = (\widetilde{x}^a)_{a \in \mathcal{A}}$ is a feasible solution to Program 2.

If the estimator $\widehat{\mu}_t$ coincides with the exact prior $\mu$, then direct calculations show that the solution $\varphi$ to Program 2 achieves the same value of the solution $\alpha$ to Program 5a.

It follows that, when $\widehat{\mu}_t = \mu$, the optimal solution of Program 2 has at least the same value of the optimal solution of Program 5a.

In order to conclude the proof, we provide a lower bound on the utility of the signaling scheme computed by Algorithm 3. Let $\phi^{\mathrm{LP}}$ be the signaling scheme computed by Algorithm 3, while let $\Psi = (x^{\mathrm{LP},a})_{a \in \mathcal{A}}$ be the optimal solution to Program 2. Furthermore, we let $\psi = (x^{\mathrm{E},a})_{a \in \mathcal{A}}$ be the optimal solution of Program 2 and $\phi^{\mathrm{E}}$ the signaling scheme computed by Algorithm 3 when the prior estimator coincides *exactly* with the prior itself, *i.e.*, $\widehat{\mu} := \widehat{\mu}_t = \mu$.

Since $x^{\mathrm{LP},a} \in \mathcal{R}^\square_\epsilon(a)$ for every $a \in \mathcal{A}$, we have that $a \in \mathcal{A}^{\phi^{\mathrm{LP}}}(a)$.[9] Breaking ties in favor of the sender, the action $a^{\phi^{\mathrm{LP}}}(s^a)$ is such that:

$$\sum_{\theta \in \Theta} \mu_\theta x^{\mathrm{LP},a}_\theta u^{\mathrm{s}}_\theta(a^{\phi^{\mathrm{LP}}}(s^a)) \geq \sum_{\theta \in \Theta} \mu_\theta x^{\mathrm{LP},a}_\theta u^{\mathrm{s}}_\theta(a).$$

---

[9] Observe that when $\mathcal{R}^\triangle_\epsilon(a) = \emptyset$ and $\mathcal{R}^\square_\epsilon(a) = \{\mathbf{0}\}$, we have $x^{\mathrm{LP},a} = \mathbf{0}$. Thus, $x^{\mathrm{LP},a}$ does not contribute to the sender's utility, $s^a \notin \mathrm{supp}(\phi^{\mathrm{LP}})$ and $\mathcal{A}^{\phi^{\mathrm{LP}}}(s^a) = \mathcal{A}$ by definition.

Then:

$$
\begin{aligned}
u(\phi^{\mathrm{LP}}) &= \sum_{s\in\mathcal{S}}\sum_{\theta\in\Theta}\mu_\theta\phi_\theta^{\mathrm{LP}}(s)u_\theta^{\mathrm{s}}(a^{\phi^{\mathrm{LP}}}(s)) \\
&\geq \sum_{s\in\mathcal{S}\setminus\{s^\star\}}\sum_{\theta\in\Theta}\mu_\theta\phi_\theta^{\mathrm{LP}}(s)u_\theta^{\mathrm{s}}(a^{\phi^{\mathrm{LP}}}(s)) \\
&= \sum_{a\in\mathcal{A}}\sum_{\theta\in\Theta}\mu_\theta x_\theta^{\mathrm{LP},a}u_\theta^{\mathrm{s}}(a^{\phi^{\mathrm{LP}}}(s^a)) \\
&\geq \sum_{a\in\mathcal{A}}\sum_{\theta\in\Theta}\mu_\theta x_\theta^{\mathrm{LP},a}u_\theta^{\mathrm{s}}(a) \\
&\geq \sum_{a\in\mathcal{A}}\sum_{\theta\in\Theta}\widehat{\mu}_\theta x_\theta^{\mathrm{LP},a}u_\theta^{\mathrm{s}}(a) - \left|\sum_{\theta\in\Theta}\widehat{\mu}_\theta - \mu_\theta\right| \\
&\geq \sum_{a\in\mathcal{A}}\sum_{\theta\in\Theta}\widehat{\mu}_\theta x_\theta^{\mathrm{E},a}u_\theta^{\mathrm{s}}(a) - \left|\sum_{\theta\in\Theta}\widehat{\mu}_\theta - \mu_\theta\right| \\
&\geq \sum_{a\in\mathcal{A}}\sum_{\theta\in\Theta}\mu_\theta x_\theta^{\mathrm{E},a}u_\theta^{\mathrm{s}}(a) - 2\left|\sum_{\theta\in\Theta}\widehat{\mu}_\theta - \mu_\theta\right|.
\end{aligned}
$$

We recall that the value of $\psi = (x^{\mathrm{E},a})_{a\in\mathcal{A}}$ is at least the optimal value of Program 5a when $\widehat{\mu} = \mu$, and such a value is at least $v^\star \geq \mathrm{OPT} - 10\epsilon nd$ according to Lemma 7, Thus, we have that:

$$
u(\phi^{\mathrm{LP}}) \geq \sum_{a\in\mathcal{A}}\sum_{\theta\in\Theta}\mu_\theta x_\theta^{\mathrm{E},a}u_\theta^{\mathrm{s}}(a) - 2\left|\sum_{\theta\in\Theta}\widehat{\mu}_\theta - \mu_\theta\right| \geq \mathrm{OPT} - 10\epsilon nd - 2\left|\sum_{\theta\in\Theta}\widehat{\mu}_\theta - \mu_\theta\right|,
$$

concluding the proof. $\qquad\square$

## E  Omitted proofs and sub-procedures of Phase 2

### E.1  `Action-Oracle`

The goal of the `Action-Oracle` procedure (Algorithm 5) is to assign the corresponding best-response to a slice $x \in \mathcal{X}_\epsilon$ received as input. In order to do so, it repeatedly commits to a signaling scheme $\phi$ such that $x$ is the slice of $\phi$ with respect to the signal $s_1$. When the signal $s_1$ is sampled, the procedure returns the best-response $a(x)$.

---
**Algorithm 5** `Action-Oracle`

---
**Require:** $x \in \mathcal{X}_\epsilon$
 1: $\phi_\theta(s_1) \leftarrow x_\theta$ and $\phi_\theta(s_2) \leftarrow 1 - x_\theta \ \forall \theta \in \Theta$
 2: **do**
 3:     Commit to $\phi^t = \phi$, observe $\theta^t$, and send $s^t$
 4:     Observe feedback $a^t$
 5:     $a \leftarrow a^t$
 6: **while** $s^t \neq s_1$
 7: **return** $a$

---

In the following, for the sake of analysis, we introduce the definition of a *clean event* under which the `Action-Oracle` procedure always returns the action $a(x) \in \mathcal{A}$, as formally stated below.

**Definition 4** (Clean event of `Action-Oracle`). *We denote $\mathcal{E}^{\mathrm{a}}$ as the event in which Algorithm 5 correctly returns the follower's best response $a(x)$ whenever executed.*

In the proof of Lemma 3 we show that, thanks to Lemma 2 and the definition of $\mathcal{X}_\epsilon$, it is possible to bound the number of rounds required to Algorithm 5 to ensure that it always returns the best response $a(x) \in \mathcal{A}$ with high probability.

---

**Algorithm 6** `Find-Polytopes`

---

**Require:** Search space $\mathcal{X}_\epsilon \subseteq \mathcal{X}$, parameter $\zeta \in (0,1)$
1: $(\mathcal{C}, \{\mathcal{X}_\epsilon(a)\}_{a\in\mathcal{C}}) \leftarrow$ `Find-Fully-Dimensional-Regions`$(\mathcal{X}_\epsilon, \zeta)$
2: **for all** $a_j \in \mathcal{A} \setminus \mathcal{C}$ **do**
3:     $\mathcal{R}_\epsilon(a_j) \leftarrow$ `Find-Face`$(\mathcal{C}, \{\mathcal{X}_\epsilon(a)\}_{a\in\mathcal{C}}, a_j)$
4: $\mathcal{R}_\epsilon(a_k) \leftarrow \mathcal{X}_\epsilon(a_k) \quad \forall a_k \in \mathcal{C}$
5: **return** $\{\mathcal{R}_\epsilon(a)\}_{a\in\mathcal{A}}$.

---

Algorithm 6 can be divided in two parts. First, by means of Algorithm 7, it computes the polytopes $\mathcal{R}_\epsilon(a) = \mathcal{X}_\epsilon(a)$ with volume strictly larger than zero. This procedure is based on the algorithm developed by Bacchiocchi et al. [2024a] for Stackelberg games. The main differences are the usage of `Action-Oracle` to query a generic normalized slice, and some technical details to account for the shape of the search space $\mathcal{X}_\epsilon$.

In the second part (loop at Line 2 Algorithm 6), it finds, for every polytope $\mathcal{X}_\epsilon(a)$ with null volume, a face $\mathcal{R}_\epsilon(a)$ such that $\mathcal{V}_\epsilon(a) \subseteq \mathcal{R}_\epsilon(a)$. Let us remark that $\mathcal{R}_\epsilon(a)$ could be the improper face $\mathcal{X}_\epsilon(a)$ itself, and it is empty if $\mathcal{V}_\epsilon(a) = \emptyset$.

**Lemma 3.** *Given inputs $\mathcal{X}_\epsilon \subseteq \mathcal{X}$ and $\zeta \in (0,1)$ for Algorithm 6, let $L := B + B_\epsilon + B_{\widehat{\mu}}$, where $B$, $B_\epsilon$, and $B_{\widehat{\mu}}$ denote the bit-complexity of numbers $\mu_\theta u_\theta(a_i)$, $\epsilon$, and $\widehat{\mu}$, respectively. Then, under event $\mathcal{E}_1$ and with at probability at least $1 - \zeta$, Algorithm 6 outputs a collection $\mathcal{R}_\epsilon := \{\mathcal{R}_\epsilon(a)\}_{a\in\mathcal{A}}$, where $\mathcal{R}_\epsilon(a)$ is a (possibly improper) face of $\mathcal{X}_\epsilon(a)$ such that $\mathcal{V}_\epsilon(a) \subseteq \mathcal{X}_\epsilon(a)$, in a number of rounds $T_2$:*

$$T_2 \leq \widetilde{\mathcal{O}}\left(\frac{n^2}{\epsilon}\log^2\left(\frac{1}{\zeta}\right)\left(d^7 L + \binom{d+n}{d}\right)\right).$$

*Proof.* In the following we let $L = B + B_\epsilon + B_{\widehat{\mu}}$.

As a first step, Algorithm 6 invokes the procedure `Find-Fully-Dimensional-Regions`. Thus, according to Lemma 8, under the event $\mathcal{E}^{\text{a}}$, with probability at least $1 - \zeta/2$, Algorithm 6 computes every polytope $\mathcal{X}_\epsilon(a)$ with volume larger than zero by performing at most:

$$C_1 = \widetilde{\mathcal{O}}\left(n^2\left(d^7 L \log(1/\zeta) + \binom{d+n}{d}\right)\right).$$

calls to the `Action-Oracle` procedure. Together with these polytopes, it computes the set $\mathcal{C} \subseteq \mathcal{A}$ containing the actions $a$ such that $\text{vol}(\mathcal{X}_\epsilon(a)) > 0$.

Subsequently, Algorithm 6 employs the procedure `Find-Face` at most $n$ times and, according to Lemma 12, it computes the polytopes $\mathcal{R}_\epsilon(a_j)$ for every $a_j \notin \mathcal{C}$. Overall, this computation requires:

$$C_2 = n^2\binom{d+n}{d}$$

calls to the `Action-Oracle` procedure. Thus, under the event $\mathcal{E}^{\text{a}}$ and with probability at least $1 - \zeta/2$, Algorithm 6 correctly computes the polytopes $\mathcal{R}_\epsilon(a_j)$ for every action $a_j \in \mathcal{A}$. Furthermore, the number of calls $C \geq 0$ to the `Action-Oracle` procedure can be upper bounded as:

$$C := C_1 + C_2 \leq \widetilde{\mathcal{O}}\left(n^2\left(d^7 L \log(1/\zeta) + \binom{d+n}{d}\right)\right).$$

Moreover, by setting $\rho = \zeta/2C$, and thanks to Lemma 2, with a probability of at least $1 - \rho$, every execution of `Action-Oracle` requires at most $N \geq 0$ rounds, where $N$ can be bounded as follows:

$$N \leq \mathcal{O}\left(\frac{\log(1/\rho)}{\epsilon}\right) = \mathcal{O}\left(\frac{1}{\epsilon}\log\left(\frac{2C}{\zeta}\right)\right).$$

Consequently, since the number of calls to the `Action-Oracle` procedure is equal to $C$, by employing a union bound, the probability that each one of these calls requires $N$ rounds to terminate is greater than or equal to:

$$1 - C\rho = 1 - \frac{\zeta}{2C}C = 1 - \frac{\zeta}{2}.$$

To conclude the proof, we employ an union bound over the probability that every execution of `Action-Oracle` terminates in at most $N$ rounds, and the probability that Algorithm 6 performs $C$ calls to the `Action-Oracle` procedure. Since the probability that each one of these two events hold is at least $1 - \varsigma/2$, with probability at least $1 - \zeta$, Algorithm 6 correctly terminates by using a number of samples of the order:

$$\widetilde{\mathcal{O}}\left(\frac{C}{\epsilon}\log\left(\frac{2C}{\zeta}\right)\right) \leq \widetilde{\mathcal{O}}\left(\frac{n^2}{\epsilon}\log^2\left(\frac{1}{\zeta}\right)\left(d^7 L + \binom{d+n}{d}\right)\right),$$

concluding the proof. $\qquad\square$

### E.3 `Find-Fully-Dimensional-Regions`

---
**Algorithm 7** `Find-Fully-Dimensional-Regions`

---
**Require:** Search space $\mathcal{X}_\epsilon \subseteq \mathcal{X}$, parameter $\delta \in (0,1)$
1: $\delta \leftarrow \varsigma/2n^2(2(d+n)+n)$
2: $\mathcal{C} \leftarrow \varnothing$
3: **while** $\bigcup_{a_j \in \mathcal{C}} \mathcal{U}(a_j) \neq \mathcal{X}_\epsilon$ **do**
4:      $x^{\text{int}} \leftarrow$ Sample a point from $\text{int}\left(\mathcal{X}_\epsilon \setminus \bigcup_{a_k \in \mathcal{C}} \mathcal{U}(a_k)\right)$
5:      $a_j \leftarrow$ `Action-Oracle`$(x^{\text{int}})$
6:      $\mathcal{U}(a_j) \leftarrow \mathcal{X}_\epsilon$
7:      $B_x \leftarrow$ Bit-complexity of $x^{\text{int}}$
8:      $\lambda \leftarrow d2^{-d(B_x + 4(B + B_\epsilon + B_{\widehat{\mu}})) - 1}$
9:      **for all** $v \in V(\mathcal{U}(a_j))$ **do**
10:          $x \leftarrow \lambda x^{\text{int}} + (1 - \lambda)v$
11:          $a \leftarrow$ `Action-Oracle`$(x)$
12:          **if** $a \neq a_j$ **then**
13:              $H_{jk} \leftarrow$ `Find-Hyperplane`$(a_j, \mathcal{U}(a_j), x^{\text{int}}, v, \delta)$
14:              $\mathcal{U}(a_j) \leftarrow \mathcal{U}(a_j) \cap \mathcal{H}_{jk}$
15:          **else**
16:              restart the for-loop at Line 9
17:      $\mathcal{C} \leftarrow \mathcal{C} \cup \{a_j\}$
18:      $\mathcal{X}_\epsilon(a_j) \leftarrow \mathcal{U}(a_j)$
19: **return** $\mathcal{C}, \{\mathcal{X}_\epsilon(a_j)\}_{a_j \in \mathcal{C}}$

---

At a high level, Algorithm 7 works by keeping track of a set $\mathcal{C} \subseteq \mathcal{A}$ of closed actions, meaning that the corresponding polytope $\mathcal{X}_\epsilon(a)$ has been completely identified. First, at Line 4, Algorithm 7 samples at random a normalized slice $x^{\text{int}}$ from the interior of one of the polytopes $\mathcal{X}_\epsilon(a_j)$ that have not yet been closed and queries it, observing the best-response $a_j \in \mathcal{A}$. Then, it initializes the entire $\mathcal{X}_\epsilon$ as the upper bound $\mathcal{U}(a_j)$ of the region $\mathcal{X}_\epsilon(a_j)$. As a further step, to verify whether the upper bound $\mathcal{U}(a_j)$ coincides with $\mathcal{X}_\epsilon(a_j)$, Algorithm 7 queries at Line 11 one of the vertices of the upper bound $\mathcal{U}(a_j)$. Since the same vertex may belong to the intersection of multiple regions, the `Action-Oracle` procedure is called upon an opportune convex combination of $x^{\text{int}}$ and the vertex $v$ itself (Line 10). If the vertex does not belong to $\mathcal{X}_\epsilon(a_j)$, then a new separating hyperplane can be computed (Line 13) and the upper bound $\mathcal{U}(a_j)$ is updated accordingly. In this way, the upper bound $\mathcal{U}(a_j)$ is refined by finding new separating hyperplanes until it coincides with the polytope $\mathcal{X}_\epsilon(a_j)$. Finally, such a procedure is iterated for all the receiver's actions $a_j \in \mathcal{A}$ such that $\text{vol}(\mathcal{X}_\epsilon(a_j)) > 0$, ensuring that all actions corresponding to polytopes with volume larger than zero are *closed*.

We observe that the estimator $\widehat{\mu}_t$ is updated during the execution of Algorithm 7 according to the observed states. However, let us remark that the search space $\mathcal{X}_\epsilon = \{x \in \mathcal{X} \mid \sum_{\theta \in \widetilde{\Theta}} \widehat{\mu}_\theta x_\theta \geq 2\epsilon\}$ does *not* change during the execution of this procedure.

**Lemma 8.** *Given in input $\mathcal{X}_\epsilon \subseteq \mathcal{X}$ and $\zeta \in (0,1)$, then under the event $\mathcal{E}^{\text{a}}$ with probability at least $1 - \varsigma/2$ Algorithm 7 computes the collection of polytopes $\{\mathcal{X}_\epsilon(a_j)\}_{a_j \in \mathcal{C}}$ with volume larger than zero, and the corresponding set of actions $\mathcal{C}$. Furthermore, it employs at most:*

$$\widetilde{\mathcal{O}}\left(n^2\left(d^7 L \log(1/\varsigma) + \binom{d+n}{d}\right)\right)$$

*calls to the* `Action-Oracle` *procedure.*

*Proof.* Thanks to Lemma 9 and Lemma 10, with an approach similar to the one proposed in Theorem 4.3 by Bacchiocchi et al. [2024a], we can prove that, under the event $\mathcal{E}^{\mathrm{a}}$, with probability at least $1 - \delta n^2(2(d+n)^2 + n)$, Algorithm 7 computes every polytope $\mathcal{X}_\epsilon(a)$ with volume larger than zero by performing at most:

$$\mathcal{O}\left(n^2\left(d^7 L \log(1/\delta) + \binom{d+n}{d}\right)\right)$$

calls to the `Action-Oracle` procedure. Together with these polytopes, it computes the set $\mathcal{C} \subseteq \mathcal{A}$ containing the actions $a$ such that $\mathrm{vol}(\mathcal{X}_\epsilon(a)) > 0$.

Furthermore, we observe that $\zeta = 2\delta n^2(2(d+n)+n)$, as defined at Line 1 in Algorithm 7. As a result, under the event $\mathcal{E}^{\mathrm{a}}$, with probability at least $1 - \zeta/2$, the number of calls $C_1 \geq 0$ performed by Algorithm 7 to the `Action-Oracle` procedure can be bounded as follows:

$$C_1 \leq \widetilde{\mathcal{O}}\left(n^2\left(d^7 L \log(1/\zeta) + \binom{d+n}{d}\right)\right),$$

concluding the proof. □

## E.4 `Find-Hyperplane`

---
**Algorithm 8** `Find-Hyperplane`

---
**Require:** $a_j, \mathcal{U}(a_j), x^{\mathrm{int}}, v, \delta$
1: $x \leftarrow$ `Sample-Int`$(\mathcal{U}(a_j), \delta)$
2: $x^1 \leftarrow x$
3: **if** `Action-Oracle`$(x) = a_j$ **then**
4:      $x^2 \leftarrow v$
5: **else**
6:      $x^2 \leftarrow x^{\mathrm{int}}$
7: $x^\circ \leftarrow$ `Binary-Search`$(a_j, x^1, x^2)$
8: $\alpha \leftarrow 2^{-4d(B_x + B + B_{\widehat{\mu}} + B_\epsilon)}/d$
9: $\mathcal{S}_j \leftarrow \varnothing; \mathcal{S}_k \leftarrow \varnothing$
10: **for** $i = 1 \ldots d$ **do**
11:      $x \leftarrow$ `Sample-Int`$(H_i \cap \mathcal{X}, \delta)$
12:      $x^{+i} \leftarrow x^\circ + \alpha(x - x^\circ)$
13:      $x^{-i} \leftarrow x^\circ - \alpha(x - x^\circ)$
14:      **if** `Action-Oracle`$(x^{+i}) = a_j$ **then**
15:          $\mathcal{S}_j \leftarrow \mathcal{S}_j \cup \{x^{+i}\} \wedge \mathcal{S}_k \leftarrow \mathcal{S}_k \cup \{x^{-i}\}$
16:      **else**
17:          $\mathcal{S}_k \leftarrow \mathcal{S}_k \cup \{x^{+i}\} \wedge \mathcal{S}_j \leftarrow \mathcal{S}_j \cup \{x^{-i}\}$
18: Build $H_{jk}$ by `Binary-Search`$(a_j, x^1, x^2)$ for $d - 1$ pairs of linearly-independent points $x^1 \in \mathcal{S}_j, x^2 \in \mathcal{S}_k$

---

The goal of the `Find-Hyperplane` procedure is to compute a new separating hyperplane $H_{jk}$ between a given region $\mathcal{X}_\epsilon(a_j)$ and some other polytope $\mathcal{X}_\epsilon(a_k)$, with $a_j, a_k \in \mathcal{A}$. To do so, it receives as input an upper bound $\mathcal{U}(a_j)$ of some polytope $\mathcal{X}_\epsilon(a_j)$, an interior point $x^{\mathrm{int}} \in \mathrm{int}(\mathcal{U}(a_j))$, a vertex $v \in \mathrm{V}(\mathcal{U}(a_j))$ that does not belong to $\mathcal{X}_\epsilon(a_j)$, and a parameter $\delta > 0$ as required by the `Sample-Int` procedure. As a first step, Algorithm 8 samples at random a slice $x$ from the interior of the upper bound $\mathcal{U}(a_j)$. Subsequently, it performs a binary search on the segment between $x$ and either $v$ or $x^{\mathrm{int}}$, depending on the best response $a(x)$ in $x$. This binary search returns a point $x^\circ$ on some new separating hyperplane $H_{jk}$ (Line 7). As a further step, the algorithm computes two sets of normalized slices, $\mathcal{S}_j \subseteq \mathcal{X}_\epsilon(a_j)$ and $\mathcal{S}_k \subseteq \mathcal{X}_\epsilon(a_k)$. Finally, it performs $d - 1$ binary searches between different couples of points, one in $\mathcal{S}_j$ and the other in $\mathcal{S}_k$, in order to completely identify the separating hyperplane.

**Lemma 9.** *With probability at least $1 - (d+n)^2\delta$, under the event $\mathcal{E}^{\mathrm{a}}$ Algorithm 8 returns a separating hyperplane $H_{jk}$ by using $\mathcal{O}(d^7(B + B_\epsilon + B_{\widehat{\mu}}) + d^4 \log(1/\delta))$ calls to Algorithm 5.*

*Proof.* We observe that, with the same analysis provided in Lemma 4.7 by Bacchiocchi et al. [2024a], we can prove that, under the event $\mathcal{E}^{\mathrm{a}}$, the binary-search procedure described in Algorithm 9 correctly computes a point on a separating hyperplane by calling the `Action-Oracle` procedure at most $\mathcal{O}(d(B_x + B))$ times.

With the same reasoning applied in Lemma 4.4 and 4.5 by Bacchiocchi et al. [2024a], we can prove that with probability at least $1 - (d+n)^2\delta$, under the event $\mathcal{E}^{\mathrm{a}}$, the points $x^{+i}$ are linearly independent and do not belong to $H_{jk}$. To conclude the proof, we have to show that every $x^{+i}$ belongs to either $\mathcal{X}_\epsilon(a_j)$ or $\mathcal{X}_\epsilon(a_k)$. This is because, if the previous condition holds, under the event $\mathcal{E}^{\mathrm{a}}$, Algorithm 8 correctly computes a new separating hyperplane with probability at least $1 - (d+n)^2\delta$.

To do that, we show that the constant $\alpha$ defined at Line 8 in Algorithm 8 is such that all the points $x^{+i}$ and $x^{-i}$ either belong to $\mathcal{X}_\epsilon(a_j)$ or $\mathcal{X}_\epsilon(a_k)$, given that $x^\circ$ belongs to the hyperplane between these polytopes. With an argument similar to the one proposed in Bacchiocchi et al. [2024a], the distance between $x^i$ and any separating hyperplane can be lower bounded by $2^{-d(B_x+4B)}$, where $B_x$ is the bit-complexity of $x^\circ$.

Similarly, the distance between $x^\circ$ and the hyperplane $\widehat{H} = \{x \in \mathbb{R}^d \mid \sum_{\theta \in \widetilde{\Theta}} \widehat{\mu}_\theta x_\theta \geq 2\epsilon\}$ can be lower bounded as follows:

$$d(x^\circ, \widehat{H}) = \frac{\left|\sum_{\theta \in \widetilde{\Theta}} x_\theta \widehat{\mu}_\theta + 2\epsilon\right|}{\sqrt{\sum_{\theta \in \widetilde{\Theta}} \widehat{\mu}_\theta^2}} \geq \frac{1}{d2^{3d(B_x + B_{\widehat{\mu}} + B_\epsilon)}}, \tag{7}$$

where $B_{\widehat{\mu}}$ is the bit-complexity of $\widehat{\mu}$. The inequality follows by observing that the denominator of the fraction above is at most $d$, while to lower bound the numerator we define the following quantities:

$$\sum_{\theta \in \widetilde{\Theta}} x_\theta \widehat{\mu}_\theta = \frac{\alpha}{\beta} \quad \text{and} \quad \epsilon = \frac{\gamma}{\nu},$$

where $\alpha$ and $\beta$ are integers numbers, while $\gamma$ and $\nu$ are natural numbers. Thus, the numerator $\left|\sum_{\theta \in \widetilde{\Theta}} x_\theta \widehat{\mu}_\theta + 2\epsilon\right|$ of the fraction defined in Equation 7 can be lower bounded as follows:

$$\left|\sum_{\theta \in \widetilde{\Theta}} x_\theta \widehat{\mu}_\theta + 2\epsilon\right| = \left|\frac{\alpha\nu + 2\beta\gamma}{\beta\nu}\right| \geq \left|\frac{1}{\beta\nu}\right| \geq 2^{-3d(B_x + B_{\widehat{\mu}} + B_\epsilon)},$$

where the last inequality follows from the fact that the bit-complexity of $\nu$ is at most $B_\epsilon$, while the bit-complexity of $\beta$ cannot exceed $3d(B_x + B_{\widehat{\mu}})$ as stated by Lemma D.1 of Bacchiocchi et al. [2024a].

Overall, the distance between $x^\circ$ and the boundary of the polytope $\mathcal{X}_\epsilon(a_j) \cap \mathcal{X}_\epsilon(a_k)$ is strictly larger than $\alpha := 2^{-4d(B_x + B + B_{\widehat{\mu}} + B_\epsilon) - \log_2(d)}$. Thus, every signaling scheme $x^{+i}$ and $x^{-i}$ belongs to $\mathcal{X}_\epsilon$ and either $\mathcal{X}(a_j)$ or $\mathcal{X}(a_k)$.

Finally, we observe that the bit-complexity of $x$ is bounded by $B_x = \mathcal{O}(d^3(B + B_\epsilon + B_{\widehat{\mu}}) + \log(1/\delta))$, as stated by Lemma 10. Thus, the first binary-search requires $\mathcal{O}(d(B_x + B)) = \mathcal{O}(d^4 L + d\log(1/\delta))$ calls to `Action-Oracle`, where $L = B + B_\epsilon + B_{\widehat{\mu}}$.

Furthermore, the bit-complexity of $x^\circ$ is bounded by $\mathcal{O}(d^4 L + d\log(1/\delta))$. As a result, the bit-complexity of the slices $x^{+i}$ and $x^{-i}$ is bounded by $\mathcal{O}(d^5 L + d\log(1/\delta))$, given that the bit-complexity of $\alpha$ is $\mathcal{O}(dL)$. It follows that each binary search between two points in $\mathcal{S}_j$ and $\mathcal{S}_k$ requires $\mathcal{O}(d^6 L + d^2 \log(1/\delta))$ calls to `Action-Oracle`.

Overall, Algorithm 8 invokes the `Action-Oracle` procedure at most $\mathcal{O}(d^7 L + d^4 \log(1/\delta))$ times, accounting for the $d - 1$ binary searches, concluding the proof. $\qquad\square$

### E.5 `Binary-Search`

The `Binary-Search` procedure performs a binary search on the segment connecting two points, $x^1, x^2 \in \mathcal{X}_\epsilon$ such that $x^1 \in \mathcal{X}_\epsilon(a_j)$ and $x^2 \notin \mathcal{X}_\epsilon(a_j)$ for some $a_j \in \mathcal{A}$, in order to find a point $x^\circ$ on some separating hyperplane $H_{jk}$. At each iteration, the binary search queries the middle point of the segment. Depending on the receiver's best-response in such a point, it keeps one of the two halves of the segment for the subsequent iteration. The binary search ends when the segment is

sufficiently small, so that it contains a single point with a bit-complexity appropriate for a point that lies on both the hyperplane $H_{jk}$ and the segment connecting $x^1$ and $x^2$. Such a point can be found traversing the Stern-Brocot-Tree. For an efficient implementation, see Forišek [2007]. Overall, Algorithm 9 performs $\mathcal{O}(d(B_x + B))$ calls to the `Action-Oracle` procedure, and returns a normalized slice $x^\circ$ with bit-complexity bounded by $\mathcal{O}(d(B_x + B))$, where $B_x$ is the bit-complexity of the points $x^1$ and $x^2$.

---

**Algorithm 9** `Binary-Search`

---

**Require:** $a_j, x^1, x^2$ of bit-complexity bounded by some $B_x > 0$
1: $\lambda_1 \leftarrow 0; \lambda_2 \leftarrow 1$
2: **while** $|\lambda_2 - \lambda_1| \geq 2^{-6d(5B_x + 8B)}$ **do**
3:      $\lambda \leftarrow (\lambda_1 + \lambda_2)/2; x^\circ \leftarrow x^1 + \lambda(x^2 - x^1)$
4:      **if** `Action-Oracle`$(x^\circ) = a_j$ **then**
5:          $\lambda_1 \leftarrow \lambda$
6:      **else**
7:          $\lambda_2 \leftarrow \lambda$
8: $\lambda \leftarrow$ `Stern-Brocot-Tree`$(\lambda_1, \lambda_2, 3d(5B_x + 8B))$
9: $x^\circ \leftarrow \lambda x^1 + (1 - \lambda)x^2$

---

## E.6 `Sample-Int`

---

**Algorithm 10** `Sample-Int`

---

**Require:** $\mathcal{P} \subseteq \mathcal{X}_\epsilon : \text{vol}_{d-1}(\mathcal{P}) > 0$, and $\delta$
1: $\mathcal{V} \leftarrow d$ linearly-independent vertexes of $\mathcal{P}$
2: $x^\diamond \leftarrow \frac{1}{d} \sum_{v \in \mathcal{V}} v$
3: $\rho \leftarrow \left( d^3 2^{9d^3 L + 4dL} \right)^{-1}; M \leftarrow \lceil \sqrt{d}/\delta \rceil$
4: $y \sim \text{Uniform}(\{-1, -\frac{M-1}{M}, \ldots, 0, \ldots, \frac{M-1}{M}, 1\}^{d-1})$
5: **for all** $i \in 1, \ldots, d-1$ **do**
6:      $x_i \leftarrow x_i^\diamond + \rho y_i$
7: $x_d \leftarrow 1 - \sum_{i=1}^{d-1} x_i$

---

The `Sample-Int` procedure (Algorithm 10) samples at random a normalized slice from the interior of a given polytope $\mathcal{P}$. We observe that each polytope Algorithm 7 is required to sample from is defined as the intersection of $\mathcal{X}_\epsilon$ with some separating half-spaces as the ones defined in Section 3. This procedure provides theoretical guarantees both on the bit-complexity of the point $x$ being sampled and on the probability that such a point belongs to a given hyperplane. Furthermore, it can be easily modified to sample a point from a facet of the simplex $\mathcal{X} = \Delta_d$ (intuitively, this is equivalent to sample a point from $\Delta_{d-1}$). As a first step, Algorithm 10 computes a normalized slice $x^\diamond$ in the interior of $\mathcal{P}$ (Line 2). Subsequently, it samples randomly a vector $y$ from a suitable grid belonging to the $(d-1)$-dimensional hypercube with edges of length 2. As a further step, it sums each component $x_i^\diamond$ of the normalized slice $x^\diamond$ with the corresponding component $y_i$ of $y$, scaled by an opportune factor $\rho$, where the constant $\rho$ is defined to ensure that $x$ belongs to the interior of $\mathcal{P}$.

The theoretical guarantees provided by Algorithm 10 are formalized in the following lemma:

**Lemma 10.** *Given a polytope $\mathcal{P} \subseteq \mathcal{X}_\epsilon : \text{vol}_{d-1}(\mathcal{P}) > 0$ defined by separating or boundary hyperplanes, Algorithm 10 computes $x \in \text{int}(\mathcal{P})$ such that, for every linear space $H \subset \mathbb{R}^d : \mathcal{P} \not\subseteq H$ of dimension at most $d-1$, the probability that $x \in H$ is at most $\delta$. Furthermore, the bit-complexity of $x$ is $\mathcal{O}(d^3(B + B_\epsilon + B_{\widehat{\mu}}) + \log(1/\delta))$.*

*Proof.* In the following, we prove that $x$ belongs to the interior of the polytope $\mathcal{P}$. To do that, we observe that the point $x^\diamond$ belongs to the interior of $\mathcal{P}$, while, with the same analysis proposed in Lemma 4.8 by Bacchiocchi et al. [2024a], the distance between $x^\diamond$ and $x$ can be upper bounded by $\rho n$. To ensure that $x$ belongs to $\text{int}(\mathcal{P})$, we have to show that $d(x^\diamond, x)$ is smaller than the distance between $x^\diamond$ and any hyperplane defining the boundary of $\mathcal{P}$.

Let us denote with $v^h$ the $h$-vertex of the set $\mathcal{V}$, so that $\mathcal{V} = \{v^1, v^2, \ldots, v^d\}$. Furthermore, we define:

$$v_\theta^h := \frac{\gamma_\theta^h}{\nu_h}$$

for each $h \in [d]$ and $\theta \in \Theta$. Since $x^\diamond$ belongs to $\text{int}(\mathcal{P})$, then the distance between $x^\diamond$ and any separating hyperplane $H_{jk}$ can be lower bounded as follows:

$$
\begin{aligned}
d(x^\diamond, H_{jk}) &= \left| \frac{\sum_{\theta \in \Theta} x_\theta^\diamond \mu_\theta(u_\theta(a_j) - u_\theta(a_k))}{\sqrt{\sum_{\theta \in \Theta} \mu_\theta^2 (u_\theta(a_j) - u_\theta(a_k))^2}} \right| \\
&= \left| \frac{\sum_{\theta \in \Theta} \sum_{h=1}^d v_\theta^h \mu_\theta(u_\theta(a_j) - u_\theta(a_k))}{d\sqrt{\sum_{\theta \in \Theta} \mu_\theta^2(u_\theta(a_j) - u_\theta(a_k))^2}} \right| \\
&\geq \frac{1}{d^2} 2^{-4dB - 9d^3(B + B_\epsilon + B_{\widehat{\mu}})}.
\end{aligned}
$$

To prove the last inequality, we observe that the denominator of the fraction above can be upper bounded by $d^2$. To lower bound the nominator, we define:

$$\mu_\theta(u_\theta(a_j) - u_\theta(a_k)) := \frac{\alpha_\theta}{\beta_\theta}$$

for each $\theta \in \Theta$. As a result, we have:

$$
\begin{aligned}
\left| \sum_{\theta \in \Theta} \sum_{h=1}^d v_\theta^h \mu_\theta(u_\theta(a_j) - u_\theta(a_k)) \right| &= \left| \sum_{\theta \in \Theta} \sum_{h=1}^d \frac{\alpha_\theta \gamma_\theta^h}{\beta_\theta \nu_h} \right| \\
&= \left| \frac{\sum_{\theta \in \Theta} \sum_{h=1}^d \alpha_\theta \gamma_\theta^h \left( \prod_{\theta' \neq \theta} \beta_{\theta'} \prod_{h' \neq h} \nu_{h'} \right)}{\prod_{\theta \in \Theta} \beta_\theta \prod_{h=1}^d \nu_h} \right| \\
&\geq \left( \prod_{\theta \in \Theta} \beta_\theta \prod_{h=1}^d \nu_h \right)^{-1} \\
&\geq 2^{-4dB - 9d^3(B + B_\epsilon + B_{\widehat{\mu}})}.
\end{aligned}
$$

The first inequality holds because the numerator of the fraction above can be lower bounded by one. The denominator can be instead upper bounded observing that the bit-complexity of each $\beta_\theta$ is at most $4B$ while the bit-complexity of each $\nu_h$ is at most $9d^2(B + B_\epsilon + B_{\widehat{\mu}})$, as stated by Lemma 11.

In a similar way, we can lower bound the distance between $x^\diamond$ and $\widehat{H} := \{x \in \mathbb{R}^d \mid \sum_{\theta \in \widetilde{\Theta}} \widehat{\mu}_\theta x_\theta \geq 2\epsilon\}$.

$$
\begin{aligned}
d(x^\diamond, \widehat{H}) &= \left| \frac{\sum_{\theta \in \widetilde{\Theta}} \widehat{\mu}_\theta x_\theta^\diamond + 2\epsilon}{\sqrt{\sum_{\theta \in \widetilde{\Theta}} \widehat{\mu}^2}} \right| \\
&= \left| \frac{\sum_{\theta \in \widetilde{\Theta}} \sum_{h=1}^d \widehat{\mu}_\theta v_\theta^h + 2\epsilon}{d\sqrt{\sum_{\theta \in \widetilde{\Theta}} \widehat{\mu}^2}} \right| \\
&\geq \frac{1}{d^2} 2^{-B_{\widehat{\mu}} - B_\epsilon - 9d^3(B + B_\epsilon + B_{\widehat{\mu}})}.
\end{aligned}
$$

To prove the last inequality, we observe that the denominator of the fraction above can be upper bounded by $d^2$, while to lower bound the numerator, we define:

$$\widehat{\mu}_\theta := \frac{N_\theta}{p} \quad \text{and} \quad \epsilon = \frac{\alpha}{\beta}$$

for each $\theta \in \widetilde{\Theta}$. As a result, we have:

$$
\left| \sum_{\theta \in \widetilde{\Theta}} \sum_{h=1}^d \widehat{\mu}_\theta v_\theta^h + 2\epsilon \right| = \left| \sum_{\theta \in \widetilde{\Theta}} \sum_{h=1}^d \frac{N_\theta \gamma_\theta^h}{p \nu_h} + 2\epsilon \right|
$$

$$
= \left| \frac{\sum_{\theta \in \widetilde{\Theta}} \sum_{h=1}^{d} \beta N_\theta \gamma_\theta^h \prod_{h' \neq h} \nu_{h'} + 2\alpha p \prod_{h=1}^{d} \nu_h}{p \prod_{h=1}^{d} \nu_h \beta} \right|
$$

$$
\geq 2^{-B_{\widehat{\mu}} - B_\epsilon - 9d^3 (B + B_\epsilon + B_{\widehat{\mu}})}
$$

Thus, the distance between $x^\diamond$ and any hyperplane $H$ defining the boundary of $\mathcal{P}$ can be lower bounded as follows:

$$
d(x^\diamond, H) \geq \frac{1}{d^2} 2^{-9d^3 (B + B_\epsilon + B_{\widehat{\mu}}) - 4dB - B_{\widehat{\mu}} - B_\epsilon} \geq \frac{1}{d^2} 2^{-10d^3 (B + B_\epsilon + B_{\widehat{\mu}})}.
$$

We observe that, given the definition of $\rho$ at Line 3, the distance $d(x^\diamond, x)$ is strictly smaller than $d(x^\diamond, H)$, showing that $x \in \mathrm{int}(\mathcal{P})$.

Furthermore, we can prove that the bit-complexity of $x$ is bounded by $\mathcal{O}(d^3(B + B_\epsilon + B_{\widehat{\mu}}) + \log(1/\delta))$. To do so, we observe that the denominator of $x^\diamond$ is equal to $d \prod_{h=1}^{d} \nu_h$, while the denominator of $y_i$ is equal to $M = \lceil \sqrt{d}/\delta \rceil$. As a result, the denominator of every $x_i = x_i^\circ + \rho y_i$, with $i \in [d-1]$, can be written as follows:

$$
D = d \prod_{h=1}^{d} \nu_h D_\rho M,
$$

where $D_\rho$ is the denominator of the rational number $\rho$. Similarly, the last component $x_d$ can be written with the same denominator. As a result, the bit complexity of $x \in [0,1]^d$ can be upper bounded as follows:

$$
\begin{aligned}
B_x &\leq 2 \lceil \log(D) \rceil \\
&= \mathcal{O}(\log(d \prod_{h=1}^{d} \nu_h D_\rho M)) \\
&= \mathcal{O}\left( \log\left( \prod_{h=1}^{d} 2^{9d^2(B + B_\epsilon + B_{\widehat{\mu}})} \right) + \log(d 2^{10d^3(B + B_\epsilon + B_{\widehat{\mu}})}) + \log(\sqrt{d}/\delta) \right) \\
&= \mathcal{O}\left( d^3(B + B_\epsilon + B_{\widehat{\mu}}) + \log\left( \frac{1}{\delta} \right) \right).
\end{aligned}
$$

Finally, with the same analysis performed in Lemma 4.8 by Bacchiocchi et al. [2024a], we can show that the probability that $x$ belongs to a given hyperplane $H$ is at most $\delta$. $\qquad \square$

**Lemma 11.** *Each vertex $v$ of a polytope $\mathcal{P} \subseteq \mathcal{X}_\epsilon : \mathrm{vol}_{d-1}(\mathcal{P}) > 0$, defined by separating or boundary hyperplanes, has bit-complexity at most $9d^2(B + B_\epsilon + B_{\widehat{\mu}})$. Furthermore, with a bit-complexity of $9d^2(B + B_\epsilon + B_{\widehat{\mu}})$, all the components of the vector $v$ identifying a vertex can be written as fractions with the same denominator.*

*Proof.* We follow a line of reasoning similar to the proof of Lemma D.2 in Bacchiocchi et al. [2024a]. Let $v$ be a vertex of the polytope $\mathcal{P}$. Then such a vertex lies on the hyperplane $H'$ ensuring that the sum of its components is equal to one. Furthermore, it also belongs to a subset of $d-1$ linearly independent hyperplanes. These can be separating hyperplanes:

$$
H_{ij} = \left\{ x \in \mathbb{R}^d \mid \sum_{\theta \in \Theta} \mu_\theta x_\theta (u_\theta(a_i) - u_\theta(a_j)) = 0 \right\}
$$

with $a_i, a_j \in \mathcal{A}$, boundary hyperplanes of the form $H_i = \{x \in \mathbb{R}^d \mid x_i > 0\}$, or the hyperplane $\widehat{H} := \{x \in \mathbb{R}^d \mid \sum_{\theta \in \widetilde{\Theta}} \widehat{\mu}_\theta x_\theta \geq 2\epsilon\}$. Consequently, there exists a matrix $A \in \mathbb{Q}^{d \times d}$ and a vector $b \in \mathbb{Q}^d$ such that $Av = b$.

Suppose that $v$ is not defined by the hyperplane $\widehat{H} := \{x \in \mathbb{R}^d \mid \sum_{\theta \in \Theta} \widehat{\mu}_\theta x_\theta \geq 2\epsilon\}$. Then, each entry of the matrix $A$ is either equal to one or the quantity $\mu_\theta(u_\theta(a) - u_\theta(a'))$ for some $\theta \in \Theta$ and $a, a' \in \mathcal{A}$. Thus, its bit-complexity is bounded by $B$. Similarly, each entry of the vector $b$ is either

equal to one or zero. With a reasoning similar to the one applied in Bacchiocchi et al. [2024a], the bit-complexity of $v$ is at most $9d^2(B_\mu + B_u)$.

Suppose instead that $v$ is defined also by the hyperplane $\widehat{H}$, corresponding to the last row of the matrix $A$ and the last component of the vector $b$. This hyperplane can be rewritten as:

$$\widehat{H} = \left\{ x \in \mathbb{R}^d \mid \sum_{\theta \in \widetilde{\Theta}} \frac{\widehat{\mu}_\theta}{2\epsilon} x_\theta \geq 1 \right\}.$$

Thus, each element of the last row of $A$ is either zero or the quantity $\widehat{\mu}_\theta/2\epsilon$ for some $\theta \in \widetilde{\Theta}$. We observe that $\widehat{\mu}_\theta/2\epsilon$ is a rational number with numerator bounded by $2^{B_{\widehat{\mu}}+B_\epsilon}$ and denominator bounded by $2^{B_{\widehat{\mu}}+B_\epsilon+1}$. Thus, we multiply the last row of $A$ and the last component of $b$ by a constant bounded by $2^{d(B_{\widehat{\mu}}+B_\epsilon+1)}$. The other rows of $A$ and the corresponding components of $b$ are multiplied instead by some constants bounded by $2^{4dB}$. This way, we obtain an equivalent system $A'v = b'$ with integer coefficients.

We define $A'(j)$ as the matrix obtained by substituting the $j$-th column of $A'$ with $b'$. Then, by Cramer's rule, the value of the $j$-th component of $v_j$ can be computed as follows:

$$v_j = \frac{\det(A'(j))}{\det(A')} \quad \forall j \in [d].$$

We observe that both determinants are integer numbers as the entries of both $A'$ and $b'$ are all integers, thus by Hadamard's inequality we have:

$$
\begin{aligned}
|\det(A')| &\leq \prod_{i \in [d]} \sqrt{\sum_{j \in [d]} {a'_{ji}}^2} \\
&\leq \left( \prod_{i \in [d-1]} \sqrt{\sum_{j \in [d]} (2^{4dB})^2} \right) \sqrt{\sum_{j \in [d]} (2^{d(B_{\widehat{\mu}}+B_\epsilon+1)})^2} \\
&= \left( \prod_{i \in [d-1]} \sqrt{d(2^{4dB})^2} \right) \sqrt{d 2^{2d(B_{\widehat{\mu}}+B_\epsilon+1)}} \\
&= \left( \prod_{i \in [d-1]} d^{\frac{1}{2}}(2^{4dB}) \right) d^{\frac{1}{2}} 2^{d(B_{\widehat{\mu}}+B_\epsilon+1)} \\
&= d^{\frac{d}{2}}(2^{4d(d-1)B}) 2^{d(B_{\widehat{\mu}}+B_\epsilon+1)} \\
&\leq d^{\frac{d}{2}}(2^{4d^2 B}) 2^{d(B_{\widehat{\mu}}+B_\epsilon+1)}
\end{aligned}
$$

With a reasoning similar to Bacchiocchi et al. [2024a], we can show that that the bit-complexity $D_v$ of the vertex $v$ is bounded by:

$$D_v \leq 9Bd^2 + 2(d(B_{\widehat{\mu}} + B_\epsilon + 1)) \leq 9d^2(B + B_\epsilon + B_{\widehat{\mu}})$$

Furthermore, this result holds when the denominator of every component $v_j$ of the vertex $v$ is written with the same denominator $\det(A')$, concluding the proof. $\qquad\square$

## E.7 Find-Face

---

**Algorithm 11** Find-Face

---

**Require:** The set of polytopes $\{\mathcal{X}_\epsilon(a_i)\}_{a_i \in \mathcal{C}}$, with volume larger than zero, and action $a_j \notin \mathcal{C}$
1: Compute the minimal H-representation $\mathcal{M}(\mathcal{X}_\epsilon(a_i))$ for every polytope $\mathcal{X}_\epsilon(a_i), a_i \in \mathcal{C}$
2: $\mathcal{H}(a_i) \leftarrow \emptyset \quad \forall a_i \in \mathcal{C}$
3: First $\leftarrow$ True
4: **for all** $x \in \bigcup_{a_i \in \mathcal{C}} V(\mathcal{X}_\epsilon(a_i))$ **do**
5:      $a \leftarrow$ Action-Oracle$(x)$
6:      **if** $a = a_j$ **then**
7:          **if** First $=$ False **then**
8:              $\mathcal{H}(a_i) \leftarrow \{H \in \mathcal{M}(\mathcal{X}_\epsilon(a_i)) \mid x \in H, H \in \mathcal{H}(a_i)\} \quad \forall a_i \in \mathcal{C}$
9:          **else**
10:             $\mathcal{H}(a_i) \leftarrow \{H \in \mathcal{M}(\mathcal{X}_\epsilon(a_i)) \mid x \in H\} \quad \forall a_i \in \mathcal{C}$
11:             First $\leftarrow$ False
12: **if** First $=$ True **then**
13:      $\mathcal{F}_\epsilon(a_j) \leftarrow \emptyset$
14: $\mathcal{F}_\epsilon(a_j) \leftarrow \mathcal{X}_\epsilon(a_i) \cap \bigcap_{H \in \mathcal{H}(a_i)} H$ for any $a_i$ such that $\mathcal{X}_\epsilon(a_i) \cap \bigcap_{H \in \mathcal{H}(a_i)} H \neq \emptyset$
15: **Return** $\mathcal{F}_\epsilon(a_j)$

---

Algorithm 11 takes in input the collection of polytopes $\{\mathcal{X}_\epsilon(a_i)\}_{a_i \in \mathcal{C}}$ and another action $a_j \notin \mathcal{C}$ such that $\text{vol}(\mathcal{X}_\epsilon(a_j)) = 0$, and outputs the H-representation of a (possibly improper) face of $\mathcal{X}_\epsilon(a_j)$ that contains all those vertices $x \in V(\mathcal{X}_\epsilon(a_j))$ where $a(x) = a_j$. As we will show by means of a pair of technical lemmas, the polytope $\mathcal{X}_\epsilon(a_j)$ is a face of some other polytope $\mathcal{X}_\epsilon(a_k)$, with $a_k \in \mathcal{C}$. Consequently, Algorithm 11 looks for a face of some polytope $\mathcal{X}_\epsilon(a_k)$ containing the set of vertices $\mathcal{V}_\epsilon(a_j)$.

As a first step, Algorithm 11 computes, for every action $a_i \in \mathcal{C}$, the set of hyperplanes $\mathcal{M}(\mathcal{X}_\epsilon(a_i))$ corresponding to the minimal H-representation of $\mathcal{X}_\epsilon(a_i)$. This set includes every separating hyperplane $H_{ik}$ found by Algorithm 7, together with the non-redundant boundary hyperplanes that delimit $\mathcal{X}_\epsilon$. Subsequently, Algorithm 11 iterates over the vertices of the regions with volume larger than zero, which we prove to include all the vertices of the region $\mathcal{X}_\epsilon(a_j)$. While doing so, it builds a set of hyperplanes $\mathcal{H}(a_i) \subseteq \mathcal{M}(\mathcal{X}_\epsilon(a_i))$ for every action $a_i \in \mathcal{C}$. Such a (possibly empty) set includes all and only the hyperplanes in $\mathcal{M}(\mathcal{X}_\epsilon(a_i))$ that contain all the vertices where the action $a_j$ has been observed, *i.e*, $a(x) = a_j$.

Finally, at Line 14 Algorithm 11 intersects every region $\mathcal{X}_\epsilon(a_i)$ with the corresponding hyperplanes in $\mathcal{H}(a_i)$, obtaining a (possibly empty) face for every polytope $\mathcal{X}_\epsilon(a_i), a_i \in \mathcal{C}$. At least one of these faces is the face the algorithm is looking for, and corresponds to the output of Algorithm 11.

The main result concerning Algorithm 11 is the following:

**Lemma 12.** *Given the collection of polytopes $\{\mathcal{X}_\epsilon(a_i)\}_{a_i \in \mathcal{C}}$ with volume larger than zero and an another action $a_j$, then, under the event $\mathcal{E}^a$, Algorithm 11 returns a (possibly improper) face $\mathcal{F}_\epsilon(a_j)$ of $\mathcal{X}_\epsilon(a_j)$ such that $\mathcal{V}_\epsilon(a_j) \subseteq \mathcal{F}_\epsilon(a_j)$. Furthermore, the Algorithm requires $\mathcal{O}(n\binom{d+n}{d})$ calls to Algorithm 5.*

In order to prove it, we first need to introduce two technical lemmas to characterize the relationship between regions with null volume and those with volume larger than zero.

**Lemma 13.** *Let $a_i, a_j \in \mathcal{A}$ such that $\text{vol}(\mathcal{X}_\epsilon(a_i)) > 0$ and $\text{vol}(\mathcal{X}_\epsilon(a_j)) = 0$. Then $\mathcal{X}_\epsilon(a_i) \cap \mathcal{X}_\epsilon(a_j)$ is a (possibly improper) face of $\mathcal{X}_\epsilon(a_i)$ and $\mathcal{X}_\epsilon(a_j)$.*

*Proof.* In the following we assume that $\mathcal{X}_\epsilon(a_i) \cap \mathcal{X}_\epsilon(a_j)$ is non-empty, as the empty set is an improper face of every polytope.

We prove that $\mathcal{X}_\epsilon(a_i) \cap \mathcal{X}_\epsilon(a_j) = \mathcal{X}_\epsilon(a_i) \cap H_{ij} = \mathcal{X}_\epsilon(a_j) \cap H_{ij}$. In order to do that, we first show that $\mathcal{X}_\epsilon(a_i) \cap H_{ij} \subseteq \mathcal{X}_\epsilon(a_i) \cap \mathcal{X}_\epsilon(a_j)$. Consider a normalized slice $x \in \mathcal{X}_\epsilon(a_i) \cap H_{ij}$. Then we have that $x \in \mathcal{X}_\epsilon(a_j)$. As this holds for every $x \in \mathcal{X}_\epsilon(a_i) \cap H_{ij}$, it follows that $\mathcal{X}_\epsilon(a_i) \cap H_{ij} \subseteq \mathcal{X}_\epsilon(a_i) \cap \mathcal{X}_\epsilon(a_j)$.

Similarly, we show that $\mathcal{X}_\epsilon(a_i) \cap \mathcal{X}_\epsilon(a_j) \subseteq \mathcal{X}_\epsilon(a_i) \cap H_{ij}$. Take any normalized slice $x \in \mathcal{X}_\epsilon(a_i) \cap \mathcal{X}_\epsilon(a_j)$. Then $x \in H_{ij}$ as it belongs to both $\mathcal{X}_\epsilon(a_i)$ and $\mathcal{X}_\epsilon(a_j)$, thus $x \in \mathcal{X}_\epsilon(a_i) \cap H_{ij}$. This implies that $\mathcal{X}_\epsilon(a_i) \cap \mathcal{X}_\epsilon(a_j) \subseteq \mathcal{X}_\epsilon(a_i) \cap H_{ij}$.

Consequently, we have that $\mathcal{X}_\epsilon(a_i) \cap \mathcal{X}_\epsilon(a_j) = \mathcal{X}_\epsilon(a_i) \cap H_{ij}$. With a similar argument, we can prove that $\mathcal{X}_\epsilon(a_i) \cap \mathcal{X}_\epsilon(a_j) = \mathcal{X}_\epsilon(a_j) \cap H_{ij}$. As a result, we have that $\mathcal{X}_\epsilon(a_i) \cap \mathcal{X}_\epsilon(a_j) = \mathcal{X}_\epsilon(a_i) \cap H_{ij} = \mathcal{X}_\epsilon(a_j) \cap H_{ij}$.

In order to conclude the proof, we show that $\mathcal{X}_\epsilon(a_i) \cap \mathcal{X}_\epsilon(a_j) = \mathcal{X}_\epsilon(a_i) \cap H_{ij} = \mathcal{X}_\epsilon(a_j) \cap H_{ij}$ is a face of both $\mathcal{X}_\epsilon(a_i)$ and $\mathcal{X}_\epsilon(a_j)$. We observe that $\mathcal{X}_\epsilon(a_i) \subseteq \mathcal{H}_{ij}$, thus the non-empty region $\mathcal{X}_\epsilon(a_i) \cap H_{ij}$ is by definition a face of $\mathcal{X}_\epsilon(a_i)$. Similarly, $\mathcal{X}_\epsilon(a_j) \subseteq \mathcal{H}_{ji}$, thus the non-empty region $\mathcal{X}_\epsilon(a_j) \cap H_{ij}$ is a face of $\mathcal{X}_\epsilon(a_j)$ (possibly the improper face $\mathcal{X}_\epsilon(a_j)$ itself). □

**Lemma 14.** *Let $\mathcal{X}_\epsilon(a_j)$ be a polytope such that $\mathrm{vol}(\mathcal{X}_\epsilon(a_j)) = 0$. Then $\mathcal{X}_\epsilon(a_j)$ is a face of some polytope $\mathcal{X}_\epsilon(a_i)$ with $\mathrm{vol}(\mathcal{X}_\epsilon(a_i)) > 0$.*

*Proof.* First, we observe that if $\mathcal{X}_\epsilon(a_j)$ is empty, then it is the improper face of any region $\mathcal{X}_\epsilon(a_i)$ with $\mathrm{vol}(\mathcal{X}_\epsilon(a_i)) > 0$. Thus, in the following, we consider $\mathcal{X}_\epsilon(a_j)$ to be non-empty.

As a first step, we observe that any normalized slice $x \in \mathcal{X}_\epsilon(a_j)$ belongs also to some region $\mathcal{X}_\epsilon(a_k)$, where $a_k \in \mathcal{A}$ depends on $x$, such that $\mathrm{vol}(\mathcal{X}_\epsilon(a_k)) > 0$. Suppose, by contradiction, that $x \in \mathrm{int}(\mathcal{X}_\epsilon(a_i))$. Then $a_i$ is a best-response in $\mathcal{X}_\epsilon(a_j) \cap \mathcal{H}_{ij}$, i.e., $\mathcal{X}_\epsilon(a_j) \cap \mathcal{H}_{ij} \subseteq \mathcal{X}_\epsilon(a_i)$. One can easily observe that such a region has positive volume, thus contradicting the hypothesis that $\mathrm{vol}(\mathcal{X}_\epsilon(a_j)) = 0$.

Now we prove that there exists some $\mathcal{X}_\epsilon(a_i)$ with $\mathrm{vol}(\mathcal{X}_\epsilon(a_i)) > 0$ such that $\mathcal{X}_\epsilon(a_j) \subseteq \mathcal{X}_\epsilon(a_i)$. If $\mathcal{X}_\epsilon(a_j)$ is a single normalized slice $x$, then this trivially holds.

Suppose instead that $\mathcal{X}_\epsilon(a_j)$ has dimension at least one. Consider a fixed normalized slice $\bar{x} \in \mathrm{int}(\mathcal{X}_\epsilon(a_j))$, where the interior is taken relative to the subspace that contains $\mathcal{X}_\epsilon(a_j)$ and has minimum dimension. There exists a region $\mathcal{X}_\epsilon(a_i)$ with $\mathrm{vol}(\mathcal{X}_\epsilon(a_i)) > 0$ such that $\bar{x} \in \mathcal{X}_\epsilon(a_i)$.

We prove that $\mathcal{X}_\epsilon(a_j) \subseteq \mathcal{X}(a_i)$. Suppose, by contradiction, that there exists a normalized slice $x \in \mathcal{X}_\epsilon(a_j)$ such that $x \notin \mathcal{X}_\epsilon(a_i)$. It follows that the line segment $\mathrm{co}(\bar{x}, x)$ intersect the separating hyperplane $H_{ij}$ in some normalized slice $\widetilde{x} \in \mathrm{co}(\bar{x}, x) \cap H_{ij}$. Furthermore, since $\widetilde{x} \neq x$ and $\bar{x} \in \mathrm{int}(\mathcal{X}_\epsilon(a_j))$, then $\widetilde{x} \in \mathrm{int}(\mathcal{X}_\epsilon(a_j))$. However, if the internal point $\widetilde{x}$ belongs to the hyperplane $H_{ij}$ and $\mathcal{X}_\epsilon(a_j) \subseteq \mathcal{H}_{ji}$, then it must be the case that $\mathcal{X}_\epsilon(a_j) \subseteq H_{ij}$. This implies that $\mathcal{X}_\epsilon(a_j) \subseteq \mathcal{X}_\epsilon(a_i)$ and thus $x \in \mathcal{X}_\epsilon(a_i)$, which contradicts the hypothesis..

Given that there exists some $\mathcal{X}_\epsilon(a_i)$ with $\mathrm{vol}(\mathcal{X}_\epsilon(a_i)) > 0$ such that $\mathcal{X}_\epsilon(a_j) \subseteq \mathcal{X}_\epsilon(a_i)$, then $\mathcal{X}_\epsilon(a_j) = \mathcal{X}_\epsilon(a_i) \cap \mathcal{X}_\epsilon(a_j)$ is a face of $\mathcal{X}_\epsilon(a_i)$ by Lemma 13. □

**Lemma 12.** *Given the collection of polytopes $\{\mathcal{X}_\epsilon(a_i)\}_{a_i \in \mathcal{C}}$ with volume larger than zero and an another action $a_j$, then, under the event $\mathcal{E}^a$, Algorithm 11 returns a (possibly improper) face $\mathcal{F}_\epsilon(a_j)$ of $\mathcal{X}_\epsilon(a_j)$ such that $\mathcal{V}_\epsilon(a_j) \subseteq \mathcal{F}_\epsilon(a_j)$. Furthermore, the Algorithm requires $\mathcal{O}(n\binom{d+n}{d})$ calls to Algorithm 5.*

*Proof.* In the following, for the sake of notation, given a polytope $\mathcal{X}_\epsilon(a)$ and the a set of hyperplanes $\mathcal{H}(a)$, with an abuse of notation we denote with $\mathcal{X}_\epsilon(a) \cap \mathcal{H}(a)$ the intersection of $\mathcal{X}_\epsilon(a)$ with every hyperplane in $\mathcal{H}(a)$. Formally:

$$\mathcal{X}_\epsilon(a) \cap \mathcal{H}(a) := \mathcal{X}_\epsilon(a) \cap \bigcap_{H \in \mathcal{H}(a)} H. \tag{8}$$

Suppose that $\mathcal{X}_\epsilon(a_j) = \emptyset$. Then, one can easily verify that Algorithm 11 returns $\emptyset$. Thus, in the following we assume $\mathcal{X}_\epsilon(a_j) \neq \emptyset$.

Let $a_i$ be action selected at Line 14 Algorithm 11. We denote with $\mathcal{F}_\epsilon(a_i)$ the face returned by Algorithm 11:

$$\mathcal{F}_\epsilon(a_i) := \mathcal{X}_\epsilon(a_i) \cap \mathcal{H}(a_i).$$

We observe that by Lemma 14, there exists an action $a_k \in \mathcal{C}$ such that $\mathcal{X}_\epsilon(a_j)$ is a face of $\mathcal{X}_\epsilon(a_k)$. Consequently, Algorithm 11 queries every vertex $x \in \mathcal{V}_\epsilon(a_j)$.

As a first step we show that $\mathcal{F}_\epsilon(a_i)$ actually is a face of $\mathcal{X}_\epsilon(a_i)$ and contains every vertex of $\mathcal{V}_\epsilon(a_j)$. Being the non-empty intersection of $\mathcal{X}_\epsilon(a_i)$ with some hyperplanes in $\mathcal{M}(\mathcal{X}_\epsilon(a_i))$, $\mathcal{F}_\epsilon(a_i)$ is a face of $\mathcal{X}_\epsilon(a_i)$. One can easily prove by induction that $\mathcal{H}(a_i)$ includes all and only the hyperplanes within $\mathcal{M}(\mathcal{X}_\epsilon(a_i))$ containing every vertex in $\mathcal{V}_\epsilon(a_j)$. Thus, $\mathcal{V}_\epsilon(a_j) \subseteq \mathcal{F}_\epsilon(a_i)$.

Now we show that $\mathcal{F}_\epsilon(a_i)$ is not only a (proper) face of $\mathcal{X}_\epsilon(a_i)$ containing the set $\mathcal{V}_\epsilon(a_j)$, but also a face of $\mathcal{X}_\epsilon(a_j)$ (possibly the improper face $\mathcal{X}_\epsilon(a_j)$ itself). We consider the set $\mathcal{X}_\epsilon(a_i) \cap \mathcal{X}_\epsilon(a_j)$, which is a face of both $\mathcal{X}_\epsilon(a_i)$ and $\mathcal{X}_\epsilon(a_j)$ thanks to Lemma 13. Thus, there exists some set of hyperplanes $\mathcal{H}'(a_i) \subset \mathcal{M}(\mathcal{X}_\epsilon(a_i))$ such that:

$$\mathcal{X}_\epsilon(a_i) \cap \mathcal{H}'(a_i) = \mathcal{X}_\epsilon(a_i) \cap \mathcal{X}_\epsilon(a_j). \tag{9}$$

Furthermore, we observe that $\mathcal{V}_\epsilon(a_j) \subseteq \mathcal{X}_\epsilon(a_i) \cap \mathcal{X}_\epsilon(a_j)$. Indeed, we have that $\mathcal{V}_\epsilon(a_j) \subseteq \mathcal{X}_\epsilon(a_i)$ since $\mathcal{V}_\epsilon(a_j) \subseteq \mathcal{F}_\epsilon(a_i)$ and $\mathcal{F}_\epsilon(a_i)$ is a face of $\mathcal{X}_\epsilon(a_i)$, and $\mathcal{V}_\epsilon(a_j) \subseteq \mathcal{X}_\epsilon(a_j)$ by definition.

We want to prove that $\mathcal{H}'(a_i) \subseteq \mathcal{H}(a_i)$, where the set $\mathcal{H}(a_i)$ contains all and only the hyperplanes within $\mathcal{M}(\mathcal{X}_\epsilon(a_i))$ that include the whole set $\mathcal{V}_\epsilon(a_j)$. In order to do that, suppose, by contradiction, that there exists a vertex $x \in \mathcal{V}_\epsilon(a_j)$ such that $x \notin H$ for some $H \in \mathcal{H}'(a_i)$. Then, $x \notin \mathcal{X}_\epsilon(a_i) \cap \mathcal{H}'(a_i) \subseteq H$. However, we proved that $\mathcal{V}_\epsilon(a_j) \subseteq \mathcal{X}_\epsilon(a_i) \cap \mathcal{X}_\epsilon(a_j)$ and $\mathcal{X}_\epsilon(a_i) \cap \mathcal{X}_\epsilon(a_j) = \mathcal{X}_\epsilon(a_i) \cap \mathcal{H}'(a_i)$ by definition of $\mathcal{H}'(a_i)$, reaching a contradiction.

Consequently:

$$\mathcal{F}_\epsilon(a_i) := \mathcal{X}_\epsilon(a_i) \cap \mathcal{H}(a_i) = \mathcal{X}_\epsilon(a_i) \cap \bigcap_{H \in \mathcal{H}(a_i)} H$$
$$\subseteq \mathcal{X}_\epsilon(a_i) \cap \bigcap_{H \in \mathcal{H}'(a_i)} H$$
$$= \mathcal{X}_\epsilon(a_i) \cap \mathcal{H}'(a_i)$$
$$= \mathcal{X}_\epsilon(a_i) \cap \mathcal{X}_\epsilon(a_j),$$

where we applied Equation 8, the fact that $\mathcal{H}'(a_i) \subseteq \mathcal{H}(a_i)$, and Equation 9.

Finally, we can show that $\mathcal{F}_\epsilon(a_i)$ is a face of $\mathcal{X}_\epsilon(a_j)$. We have that $\mathcal{F}_\epsilon(a_i)$ is a face of $\mathcal{X}_\epsilon(a_i)$ and $\mathcal{F}_\epsilon(a_i) \subseteq \mathcal{X}_\epsilon(a_i) \cap \mathcal{X}_\epsilon(a_j)$, which is itself a face of $\mathcal{X}_\epsilon(a_i)$ by Lemma 13. Thus, $\mathcal{F}_\epsilon(a_i)$ is a face of $\mathcal{X}_\epsilon(a_i) \cap \mathcal{X}_\epsilon(a_j)$. Furthermore, Lemma 13 states that $\mathcal{X}_\epsilon(a_i) \cap \mathcal{X}_\epsilon(a_j)$ is also a face of $\mathcal{X}_\epsilon(a_j)$. This implies that $\mathcal{F}_\epsilon(a_i)$ is a face of a face of $\mathcal{X}_\epsilon(a_j)$, and thus a face of $\mathcal{X}_\epsilon(a_j)$ itself.

In order to conclude the proof, we have to prove that at Line 14 Algorithm 11 can actually find an action $a_i$ such that $\mathcal{F}_\epsilon(a_i) = \mathcal{X}_\epsilon(a_i) \cap \mathcal{H}(a_i)$ is non-empty. Let $a_k \in \mathcal{C}$ be such that $\mathcal{X}_\epsilon(a_j)$ is a face of $\mathcal{X}_\epsilon(a_k)$, which exists thanks to Lemma 14. Let $x$ be any vertex in the set $\mathcal{V}_\epsilon(a_j)$ and define $\mathcal{H}''(a_j)$ as:
$$\mathcal{H}''(a_k) := \{H \in \mathcal{R}^H(\mathcal{X}_\epsilon(a_k)) \mid x \in H\}.$$
Then $\mathcal{X}_\epsilon(a_k) \cap \mathcal{H}''(a_k) = \{x\}$. Consequently, $\mathcal{H}(a_k) \subseteq \mathcal{H}''(a_k)$, and thus:

$$\mathcal{X}_\epsilon(a_k) \cap \mathcal{H}(a_k) = \mathcal{X}_\epsilon(a_k) \cap \bigcap_{H \in \mathcal{H}(a_k)} H$$
$$\subseteq \mathcal{X}_\epsilon(a_k) \cap \bigcap_{H \in \mathcal{H}''(a_k)} H$$
$$= \mathcal{X}_\epsilon(a_k) \cap \mathcal{H}''(a_k)$$
$$= \{x\} \neq \emptyset.$$

As a result, there is always an action $a_k \in \mathcal{C}$ such that $\mathcal{X}_\epsilon(a_k) \cap \mathcal{H}(a_k) \neq \emptyset$.

Finally, we observe that Algorithm 11 executes Algorithm 5 once for every vertex in the set $\bigcup_{a_i \in \mathcal{C}} V(\mathcal{X}_\epsilon(a_i))$, which has size $\mathcal{O}(n\binom{d+n}{d})$. □

## F  Omitted proofs from Section 5

**Theorem 2.** *For any sender's algorithm, there exists a Bayesian persuasion instance in which $n = d + 2$ and the regret $R_T$ suffered by the algorithm is at least $2^{\Omega(d)}$, or, equivalently, $2^{\Omega(n)}$.*

*Proof.* In the following, for the sake of the presentation, we consider a set of instances characterized by an even number $d \in \mathbb{N}_+$ of states of nature and $n = d + 2$ receiver's actions. All the instances share the same uniform prior distribution and the same sender's utility, given by $u_\theta^s(a_{d+1}) = 1$ for all $\theta \in \Theta$, and $u_\theta^s(a) = 0$ for all $\theta \in \Theta$ and $\forall a \in \mathcal{A} \setminus \{a_{d+1}\}$. Each instance is parametrized by a vector $p$ belonging to a set $\mathcal{P}$ defined as follows:

$$\mathcal{P} := \left\{ p \in \{0,1\}^d \,\Big|\, \sum_{i=1}^d p_i = \frac{d}{2} \right\}.$$

Furthermore, we assume that the receiver's utility in each instance $I_p$ is given by:

$$\textcircled{$I_p$} \begin{cases} u_{\theta_i}(a_j) = \delta_{ij} \ \forall i, j \in [d], \\ u_{\theta_i}(a_{d+1}) = \frac{2}{d} p_i \ \forall i \in [d], \\ u_{\theta_i}(a_{d+2}) = \frac{2}{d} \ \forall i \in [d]. \end{cases}$$

We show that $\xi'_{\theta_i} := \frac{2}{d} p_i$ for each $i \in [d]$ is the only posterior inducing the receiver's action $a_{d+1} \in \mathcal{A}$ in the instance $I_p$, since the receiver breaks ties in favor of the sender. To prove that, we observe that the action $a_{d+1}$ is preferred to the action $a_{d+2}$ only in those posteriors that satisfy the condition $\xi_{\theta_i} = 0$ for each $i \in [d]$ with $p_i = 0$. Furthermore, to incentivize the action $a_{d+1}$ over the set of actions $a_i$ with $i \in [d]$, the following condition must hold:

$$\sum_{i \in [d]:p_i>0} \xi_{\theta_i} u_{\theta_i}(a_{n+1}) = \frac{2}{d} \sum_{i \in [d]:p_i>0} \xi_{\theta_i} = \frac{2}{d} \geq \max_{i \in [d]:p_i>0} \xi_{\theta_i}.$$

We notice that the last step holds only if $\xi_{\theta_i} \leq 2/d$ for each $i \in [d]$ such that $p_i > 0$. Consequently, since the number of $\xi_{\theta_i} > 0$ is equal to $d/2$, it holds $\xi_{\theta_i} = 2/d$ for each $i \in [d]$ such that $p_i > 0$. Thus, the only posterior inducing action $a_{d+1}$ is equal to $\xi'_{\theta_i} := \frac{2}{d} p_i$.

We also notice that, given $p \in \mathcal{P}$, the optimal signaling scheme $\gamma$ is defined as $\gamma(\xi') = 1/2$ and $\gamma(\xi'') = 1/2$, with $\xi''_\theta = \mu_\theta - \frac{1}{2}\xi'_\theta$ for each $\theta \in \Theta$. With a simple calculation, we can show that the expected sender's utility in $\gamma$ is equal to $1/2$.

We set the time horizon $T = \lfloor |\mathcal{P}|/4 \rfloor$ to show that any algorithm suffers regret of at least $2^{\Omega(d)}$. This is sufficient to prove the statement. We start by making the following simplifications about the behavior of the algorithm. First, we observe that if the algorithm can choose any posterior (instead of a signaling scheme), then this will only increase the performance of the algorithm. Consequently, we assume that the algorithm chooses a posterior $\xi_t$ at each round $t \in [T]$.

Thus, we can apply Yao's minimax principle to show that any deterministic algorithm fails against a distribution over instances. In the following, we consider a uniform distribution over instances $I_p$ with $p \in \mathcal{P}$. Furthermore, we observe that the feedback of any algorithm is actually binary. Thus, it is easy to see that an optimal algorithm works as follows: (i) it ignores the feedback whenever the action is not $a_{d+1}$, and (ii) it starts to play the optimal posterior when the action is $a_{d+1}$ since it found an optimal posterior.

This observation is useful for showing that any deterministic algorithm does not find a posterior that induces action $a_{d+1}$ with a probability of at least $1 - |\mathcal{P}|/(4|\mathcal{P}|) = 3/4$ (since it can choose only $\lfloor |\mathcal{P}|/4 \rfloor$ posteriors among the $|\mathcal{P}|$ possible optimal posteriors). Hence, by Yao's minimax principle, for any (randomized) algorithm there exists an instance such that the regret suffered in the $T$ rounds is at least:

$$R_T \geq \frac{3}{4} \frac{T}{2} \geq \frac{1}{4} \left\lfloor \frac{|\mathcal{P}|}{4} \right\rfloor \geq \frac{|\mathcal{P}|}{32},$$

since $\lfloor x \rfloor \geq x - 1 \geq x/2$, for each $x \geq 2$. Finally, we notice that $|\mathcal{P}| = \binom{d}{d/2} = 2^{\Omega(d)}$, which concludes the proof. $\square$

**Theorem 3.** *For any sender's algorithm, there exists a Bayesian persuasion instance in which the regret $R_T$ suffered by the algorithm is at least $\Omega(\sqrt{T})$.*

*Proof.* To prove the theorem, we introduce two instances characterized by two states of nature and four receiver's actions. In both the instances the sender's utility is given by $u_\theta^s(a_1) = u_\theta^s(a_2) = 0$ for

all $\theta \in \Theta$, while $u^{\text{s}}_{\theta_1}(a_3) = 1$, $u^{\text{s}}_{\theta_2}(a_3) = 0$ and $u^{\text{s}}_{\theta_2}(a_4) = 0$ $u^{\text{s}}_{\theta_2}(a_4) = 1$. The receiver's utilities and the prior distributions in the two instances are:

$$
① \begin{cases}
\mu^1_{\theta_1} = \frac{1}{2} + \epsilon, \ \mu^1_{\theta_2} = \frac{1}{2} - \epsilon \\
u^1_{\theta_1}(a_1) = \frac{1}{(2+4\epsilon)}, \ u^1_{\theta_2}(a_1) = \frac{1}{(10-20\epsilon)} \\
u^1_{\theta_1}(a_2) = \frac{1}{(10+20\epsilon)}, \ u^1_{\theta_2}(a_2) = \frac{1}{(2-4\epsilon)} \\
u^1_{\theta_1}(a_3) = \frac{3}{10}, \ u^1_{\theta_2}(a_3) = \frac{3}{10} \\
u^1_{\theta_1}(a_4) = 0, \ u^1_{\theta_2}(a_4) = \frac{1}{(2-4\epsilon)}
\end{cases}
\qquad
② \begin{cases}
\mu^2_{\theta_1} = \frac{1}{2} - \epsilon, \ \mu^2_{\theta_2} = \frac{1}{2} + \epsilon \\
u^2_{\theta_1}(a_1) = \frac{1}{(2-4\epsilon)}, \ u^2_{\theta_2}(a_1) = \frac{1}{(10+20\epsilon)} \\
u^2_{\theta_1}(a_2) = \frac{1}{(10-20\epsilon)}, \ u^2_{\theta_2}(a_2) = \frac{1}{(2+4\epsilon)} \\
u^2_{\theta_1}(a_3) = \frac{3}{10}, \ u^2_{\theta_2}(a_3) = \frac{3}{10} \\
u^2_{\theta_1}(a_4) = 0, \ u^2_{\theta_2}(a_4) = \frac{1}{(2+4\epsilon)}
\end{cases}
$$

with $\epsilon \in (0, 1/4)$. With a simple calculation, we can show that, in both the two instances, for any signaling scheme $\phi$ employing a generic set of signals $\mathcal{S}$, the sender receives the following feedback:

1. $\forall s \in \mathcal{S}$ such that $\phi_{\theta_1}(s) > \phi_{\theta_2}(s)$, then $a^\phi(s) = a_1$.

2. $\forall s \in \mathcal{S}$ such that $0 < \phi_{\theta_1}(s) < \phi_{\theta_2}(s)$, then $a^\phi(s) = a_2$.

3. $\forall s \in \mathcal{S}$ such that $\phi_{\theta_1}(s) = \phi_{\theta_2}(s)$, then $a^\phi(s) = a_3$.

4. $\forall s \in \mathcal{S}$ such that $0 = \phi_{\theta_1}(s)$, then $a^\phi(s) = a_4$.

As a result, for any signaling scheme the sender may commit to, the resulting feedback in each signal of the signaling scheme is the same. Thus, we assume without loss of generality, that the sender only commits to signaling schemes that maximizes the probability of inducing actions $a_3$ or $a_4$. This is because, the sender does not gain any information by committing to one signaling scheme over another, while the signaling schemes that induce these two actions are the only ones that provide the sender with strictly positive expected utility.

Furthermore, thanks to what we have observed before, we can restrict our attention to direct signaling schemes, *i.e.*, those in which the set of signals coincides with the set of actions. Thus, at each round, we assume that the sender commits to a signaling scheme $\phi^t$ of the form:

$$
\phi^t := \begin{cases}
\phi^t_{\theta_1}(a_3) = \phi^t_{\theta_2}(a_3) := \phi^t_1, \\
\phi^t_{\theta_1}(a_4) = 0, \ \phi^t_{\theta_2}(a_4) = 1 - \phi^t_1, \\
\phi^t_{\theta_1}(a_1) = 1 - \phi^t_1, \ \phi^t_{\theta_2}(a_1) = 0,
\end{cases}
\tag{10}
$$

with $\phi^t_1 \in [0, 1]$. We also notice that, in each round, the optimal signaling scheme in the first instance is the one that induces action $a_3$, meaning $\phi^t_1 = 1$ for each $t \in [T]$. While the optimal signaling scheme in the second instance at each round is the one that reveals the state of nature to the receiver, meaning $\phi^t_1 = 0$ for each $t \in [T]$. In such a way, the learning task undertaken by the sender reduces to select a value of $\phi^t_1 \in [0, 1]$ for each $t \in [T]$ controlling the probability of inducing action $a_3$ over action $a_4$.

In the following, we define $\mathbb{P}^1$ ($\mathbb{P}^2$) as the probability distribution generated by the execution of a given algorithm in the first (second) instance and we let $\mathbb{E}^1$ ($\mathbb{E}^2$) be the expectation induced by such a distribution.

The cumulative regret in the first instance can be written as follows:

$$
R^1_T = \mathbb{E}^1 \left[ \sum_{t=1}^{T} \left( \frac{1}{2} + \epsilon - \phi^t_1 \left( \frac{1}{2} + \epsilon \right) - \left( 1 - \phi^t_1 \right) \left( \frac{1}{2} - \epsilon \right) \right) \right]
$$

$$
= 2\epsilon \mathbb{E} \left[ \sum_{t=1}^{T} \left( 1 - \phi^t_1 \right) \right].
$$

Similarly, in the second instance, the cumulative regret is given by:

$$
R^2_T = 2\epsilon \mathbb{E} \left[ \sum_{t=1}^{T} \phi^t_1 \right].
$$

Furthermore, it is easy to check that:

$$
R^1_T \geq \mathbb{P}^1 \left( \sum_{t=1}^{T} \phi^1_t \leq T/2 \right) \epsilon T \quad \text{and} \quad R^2_T \geq \mathbb{P}^2 \left( \sum_{t=1}^{T} \phi^1_t \geq T/2 \right) \epsilon T.
$$

By employing the relative entropy identities divergence decomposition we also have that:

$$\mathcal{KL}\left(\mathbb{P}^1, \mathbb{P}^2\right) = T \cdot \mathcal{KL}\left(\mu^1, \mu^2\right)$$
$$\leq \frac{64}{3}T\epsilon^2 \leq 22T\epsilon^2,$$

where we employed the fact that for two Bernoulli distribution it holds

$$\mathcal{KL}(\mathrm{Be}(p), \mathrm{Be}(q)) \leq \frac{(p-q)^2}{q(1-q)}.$$

Then, by employing the Bretagnolle–Huber inequality we have that,

$$R_T^1 + R_T^2 \geq \epsilon T \left(\mathbb{P}^1\left(\sum_{t=1}^T \phi_t^1 \leq T/2\right) + \mathbb{P}^2\left(\sum_{t=1}^T \phi_t^1 \geq T/2\right)\right)$$
$$\geq \frac{1}{2}\epsilon T \exp\left(-\mathcal{KL}\left(\mathbb{P}^1, \mathbb{P}^2\right)\right)$$
$$\geq \frac{1}{2}\epsilon T \exp\left(-22T\epsilon^2\right).$$

By taking $\epsilon = \sqrt{\frac{1}{22T}}$ we get:

$$R_T^1 + R_T^2 \geq C_1\sqrt{T}.$$

with $C_1 = {e^{-1}}/{(2\sqrt{22})}$ Thus, we have:

$$R_T^1 \geq \frac{C_1}{2}\sqrt{T} \quad \vee \quad R_T^2 \geq \frac{C_1}{2}\sqrt{T},$$

concluding the proof. $\qquad\square$

## G   Details and omitted proofs from Section 6

### G.1   `Compute-Threshold` **procedure**

The `Compute-Threshold` procedure takes as input a real parameter $\epsilon_1 > 0$. Then, it iteratively halves the value of a different parameter $\epsilon$, initially set to one, until it is smaller than or equal to $\epsilon_1$. In this way, Algorithm 12 computes a parameter $\epsilon \in [\epsilon_1/2, \epsilon_1]$ in $\mathcal{O}(\log(^1/\epsilon_1))$ rounds with bit complexity $B_\epsilon = \mathcal{O}(\log(^1/\epsilon_1))$. This technical component is necessary to ensure that the bit-complexity of the parameter $\epsilon$ is not too large while guaranteeing that the solution returned by Algorithm 4 is still $\gamma$-optimal with probability at least $1 - \eta$.

---

**Algorithm 12** `Compute-Threshold`

---

**Require:** $\epsilon_1 \in (0, 1)$
 1: $\epsilon \leftarrow 1$
 2: **while** $\epsilon \geq \epsilon_1$ **do**
 3: $\quad \epsilon \leftarrow \epsilon/2$
 4: **Return** $\epsilon$

---

### G.2   Omitted proofs from Section 6

**Lemma 5.** *Given $T_1 := \left\lceil \frac{1}{2\epsilon^2} \log\left(2d/\delta\right)\right\rceil$ and $\epsilon \in (0, 1)$, Algorithm 2 employs $T_1$ rounds and outputs $\mathcal{X}_\epsilon \subseteq \mathcal{X}$ such that, with probability at least $1 - \delta$: (i) $\sum_{\theta \in \Theta} \mu_\theta x_\theta \geq \epsilon$ for every slice $x \in \mathcal{X}_\epsilon$, (ii) $\sum_{\theta \in \Theta} \mu_\theta x_\theta \leq 6\epsilon$ for every slice $x \in \mathcal{X} \setminus \mathcal{X}_\epsilon$, and (iii) $|\widehat{\mu}_\theta - \mu_\theta| \leq \epsilon$ for every $\theta \in \Theta$.*

*Proof.* Thanks to the definition of $T_1 := \left\lceil {1}/{2\epsilon^2} \log\left(2d/\delta\right)\right\rceil$ in Algorithm 4 and employing both a union bound and the Hoeffding bound we have:

$$\mathbb{P}\left(|\widehat{\mu}_\theta - \mu_\theta| \leq \epsilon\right) \geq 1 - \delta, \quad \forall \theta \in \Theta.$$

Consequently, if $|\widehat{\mu}_\theta - \mu_\theta| \le \epsilon$ for each $\theta \in \Theta$, then for each $x \in \mathcal{X}_\epsilon$, the probability of inducing the slice $x$ can be lower bounded as follows:

$$\epsilon \le \sum_{\theta \in \Theta'} \widehat{\mu}_\theta x_\theta - \epsilon \le \sum_{\theta \in \Theta'} (\widehat{\mu}_\theta - \epsilon) x_\theta \le \sum_{\theta \in \Theta'} \mu_\theta x_\theta \le \sum_{\theta \in \Theta} \mu_\theta x_\theta$$

where the above inequalities hold because each $x \in \mathcal{X}_\epsilon$ satisfies the constraint $\sum_{\theta \in \Theta'} \widehat{\mu}_\theta x_\theta \ge 2\epsilon$. Furthermore, if $|\widehat{\mu}_\theta - \mu_\theta| \le \epsilon$ for each $\theta \in \Theta$, then for each $x \notin \mathcal{X}_\epsilon$ the following holds:

$$\sum_{\theta \in \Theta'} \mu_\theta x_\theta \le \epsilon + \sum_{\theta \in \Theta'} (\mu_\theta - \epsilon) x_\theta \le \epsilon + \sum_{\theta \in \Theta'} \widehat{\mu}_\theta x_\theta \le 3\epsilon, \tag{11}$$

since, if $x \notin \mathcal{X}_\epsilon$, it holds $\sum_{\theta \in \Theta'} \widehat{\mu}_\theta x_\theta \le 2\epsilon$, and,

$$\sum_{\theta \notin \Theta'} \mu_\theta x_\theta \le \sum_{\theta \notin \Theta'} (\mu_\theta - \epsilon) x_\theta + \epsilon \le \sum_{\theta \notin \Theta'} \widehat{\mu}_\theta x_\theta + \epsilon \le 3\epsilon. \tag{12}$$

Thus, by combining Inequality (11) and Inequality (12), we have:

$$\sum_{\theta \in \Theta} \mu_\theta x_\theta \le 6\epsilon,$$

when $x \notin \mathcal{X}_\epsilon$, concluding the proof. $\qquad\square$

**Theorem 4.** *Given $\gamma \in (0,1)$ and $\eta \in (0,1)$, with probability at least $1 - \eta$, Algorithm 4 outputs a $\gamma$-optimal signaling scheme in a number of rounds $T$ such that:*

$$T \le \widetilde{\mathcal{O}}\left( \frac{n^3}{\gamma^2} \log^2\left(\frac{1}{\eta}\right) \left( d^8 B + d \binom{d+n}{d} \right) \right).$$

*Proof.* Thanks to Lemma 5, with probability at least $1 - \delta = 1 - \eta/2$ Algorithm 4 correctly completes Phase 1 in $T_1 = \mathcal{O}(1/\epsilon^2 \log(1/\eta) \log(d))$ rounds. Thus, with probability at least $1 - \eta/2$, both the event $\mathcal{E}_1$ and the inequalities $|\widehat{\mu}_\theta - \mu_\theta| \le \epsilon$ for each $\theta \in \Theta$ hold.

Consequently, under the event $\mathcal{E}_1$, with probability at least $1 - \zeta = 1 - \eta/2$, Algorithm 4 correctly partitions the search space $\mathcal{X}_\epsilon$ in at most:

$$\widetilde{\mathcal{O}}\left( \frac{n^2}{\epsilon} \log^2\left(\frac{1}{\eta}\right) \left( d^7 L + \binom{d+n}{d} \right) \right)$$

rounds, as stated by Lemma 3. Furthermore, we notice that $L = B + B_\epsilon + B_{\widehat{\mu}}$, with:

$$B_{\widehat{\mu}} = \mathcal{O}(\log(T_1)) = \mathcal{O}\left( \log(1/\epsilon) + \log(1/\eta) + \log(d) \right).$$

As a result, with probability at least $1 - \eta$, Algorithm 4 correctly terminates in a number of rounds $N$ which can be upper bounded as follows:

$$N \le \widetilde{\mathcal{O}}\left( \frac{1}{\epsilon^2} \log\left(\frac{1}{\eta}\right) \log(d) + \frac{n^2}{\epsilon} \log^2\left(\frac{1}{\eta}\right) \left( d^7 (B + B_\epsilon) + \binom{d+n}{d} \right) \right).$$

Furthermore, we observe that if $|\widehat{\mu}_\theta - \mu_\theta| \le \epsilon$ for each $\theta \in \Theta$, then the following holds:

$$\left| \sum_{\theta \in \Theta} \widehat{\mu}_\theta - \mu_\theta \right| \le \sum_{\theta \in \Theta} |\widehat{\mu}_\theta - \mu_\theta| \le \sum_{\theta \in \Theta} \epsilon = \epsilon d.$$

Consequently, thanks to the result provided by Lemma 4, with probability at least $1 - \eta$, Algorithm 4 computes a $12n\epsilon d$-optimal solution. Thus, by setting $\epsilon_1 := \gamma/12nd$ and $\epsilon \le \epsilon_1$, with probability at least $1 - \eta$ Algorithm 4 computes a $\gamma$-optimal solution in a number of rounds $N$ bounded by:

$$\begin{aligned} N &\le \widetilde{\mathcal{O}}\left( \frac{n^2 d^2}{\gamma^2} \log\left(\frac{1}{\eta}\right) + \frac{n^3}{\gamma} \log^2\left(\frac{1}{\eta}\right) \left( d^8 (B + B_\epsilon) + d \binom{d+n}{d} \right) \right) \\ &= \widetilde{\mathcal{O}}\left( \frac{n^3}{\gamma^2} \log^2\left(\frac{1}{\eta}\right) \left( d^8 (B + B_\epsilon) + d \binom{d+n}{d} \right) \right) \\ &= \widetilde{\mathcal{O}}\left( \frac{n^3}{\gamma^2} \log^2\left(\frac{1}{\eta}\right) \left( d^8 B + d \binom{d+n}{d} \right) \right), \end{aligned}$$

where the last equality holds because the bit-complexity of $\epsilon$ is $B_\epsilon = \mathcal{O}(\log(nd/\gamma))$, concluding the proof. $\qquad\square$

**Theorem 5.** *There exist two absolute constants $\kappa, \lambda > 0$ such that no algorithm is guaranteed to return a $\kappa$-optimal signaling scheme with probability of at least $1 - \lambda$ by employing less than $2^{\Omega(n)}$ and $2^{\Omega(d)}$ rounds, even when the prior distribution $\mu$ is known to the sender.*

*Proof.* In the proof of Theorem 2 we showed that, with probability $3/4$, in $N = \lfloor |\mathcal{P}|/4 \rfloor$ rounds any algorithm does not correctly identify the posterior inducing action $a_{d+1}$. This is because, any deterministic algorithm can identify the optimal posterior only in $\lfloor |\mathcal{P}|/4 \rfloor$ instances, as observed in the proof of Theorem 2.

As a result, in the remaining $|\mathcal{P}| - N$ instances, any deterministic algorithm will receive the same feedback and thus will always output the same posterior after $N$ rounds, which will result in the optimal one in only a single instance.

Thus, thanks to the Yao's minimax principle, there is no algorithm that is guaranteed to return an optimal solution with probability at least:

$$\frac{3}{4}\left(\frac{|\mathcal{P}| - N - 1}{|\mathcal{P}| - N}\right) \geq \frac{3}{8}\left(\frac{|\mathcal{P}| - N}{|\mathcal{P}| - N}\right) = \frac{3}{8}.$$

Finally, we observe that OPT $= 1/2$, while any algorithm that does not induce the posterior $\xi'$ provides an expected utility equal to zero. As a result, for each $\kappa < 1/2$ there is no algorithm that is guaranteed to return a solution which is $\kappa$-optimal in $\lfloor |\mathcal{P}|/4 \rfloor = 2^{\Omega(d)}$ rounds with probability at least $3/8$. $\square$

**Theorem 6.** *Given $\gamma \in (0, 1/8)$ and $\eta \in (0, 1)$, no algorithm is guaranteed to return a $\gamma$-optimal signaling scheme with probability at least $1 - \eta$ by employing less than $\Omega\left(\frac{1}{\gamma^2}\log(1/\eta)\right)$ rounds.*

*Proof.* To prove the theorem, we consider the same instance and the same definitions introduced in the proof of Theorem 3. In this case, we let $\mathbb{P}^1$ ($\mathbb{P}^2$) be the probability distribution generated by the execution of a given algorithm in the first (second) instance for $N = \lceil \log(1/4\eta)/22\epsilon^2 \rceil$ rounds. Furthermore, we introduce the event $\mathcal{E}$, under which the signaling scheme returned at the round $N$, according to the definition presented in Equation 10, is such that $\phi^N \leq 1/2$. We notice that, if such signaling scheme is such that $\phi^N < 1/2$, then the sender's expected utility in the first instance is smaller or equal to $1/2$, thus being $\epsilon/2$-optimal. At the same time, if $\phi^N \geq 1/2$ in the second instance, then the solution returned by the algorithm is not $\epsilon/2$-optimal.

Thus, by employing the Bretagnolle–Huber inequality we have:

$$\mathbb{P}^1(\mathcal{E}) + \mathbb{P}^2(\mathcal{E}^C) \geq \frac{1}{2}\exp\left(-\mathcal{KL}\left(\mathbb{P}^1, \mathbb{P}^2\right)\right) \geq \frac{1}{2}\exp\left(-22N\epsilon^2\right),$$

since $\mathcal{KL}\left(\mathbb{P}^1, \mathbb{P}^2\right) \leq 22N\epsilon^2$, as observed in the proof of Theorem 6. Finally, by employing the definition of $N$, we have:

$$\mathbb{P}^1(\mathcal{E}) \geq \eta \quad \vee \quad \mathbb{P}^2(\mathcal{E}^C) \geq \eta.$$

As a result, by setting $2\gamma = \epsilon$, the statement of the lemma holds. $\square$

## H    Sample complexity with known prior

In this section, we discuss the *Bayesian persuasion PAC-learning problem* when the prior distribution $\mu$ is known to the sender. To tackle the problem, we propose Algorithm 13. The main difference with respect to Algorithm 4 is that, in this case, we do not need to employ the `Build-Search-Space` procedure, as the prior is already known to the sender. This allows us to compute a $\gamma$-optimal signaling scheme in only $\mathcal{O}(1/\gamma)$ rounds, instead of $\mathcal{O}(1/\gamma^2)$ rounds as in the case with an unknown prior.

---

**Algorithm 13** PAC-Persuasion-w/o-Clue-Known

---

**Require:** $\eta \in (0,1), \gamma \in (0,1), \mu \in \Delta_\Theta$

1: $\epsilon_1 \leftarrow \gamma/10nd$
2: $\epsilon \leftarrow$ Compute-Epsilon$(\epsilon_1), \widehat{\mu} \leftarrow \mu$
3: $\widetilde{\Theta} \leftarrow \{\theta \in \Theta \mid \widehat{\mu}_\theta > 2\epsilon\}$
4: $\mathcal{X}_\epsilon \leftarrow \left\{x \in \mathcal{X} \mid \sum_{\theta \in \widetilde{\Theta}} \widehat{\mu}_\theta x_\theta \geq 2\epsilon\right\}$
5: $\mathcal{R}_\epsilon \leftarrow$ Find-Polytopes$(\mathcal{X}_\epsilon, \eta)$
6: $\phi \leftarrow$ Compute-Signaling$(\mathcal{R}_\epsilon, \mathcal{X}_\epsilon, \mu)$
7: **return** $\phi$

---

In this case, the following theorem holds.

**Theorem 7.** *With probability at least $1 - \eta$ and in Algorithm 2 computes a $\gamma$-optimal signaling scheme in $\widetilde{\mathcal{O}}\left(n^3/\gamma \log^2 (1/\eta)\left(d^8 B + d\binom{d+n}{d}\right)\right)$ rounds.*

*Proof.* Since $\widehat{\mu} = \mu$, the clean event $\mathcal{E}_1$ holds with probability one. Consequently, thanks to Lemma 3, with probability at least $1 - \eta$, Algorithm 13 correctly partitions the search space $\mathcal{X}_\epsilon$ in at most:

$$\widetilde{\mathcal{O}}\left(\frac{n^2}{\epsilon} \log^2 (1/\eta)\left(d^7 L + \binom{d+n}{d}\right)\right)$$

rounds, with $L := B + B_\epsilon + B_{\widehat{\mu}}$. According to Algorithm 13, we have $\epsilon \leq \epsilon_1 := \gamma/(10nd)$. As a result, $L = \mathcal{O}\left(B + \log(nd) + \log(1/\gamma)\right)$, since $B_{\widehat{\mu}} \leq B$ and $B_\epsilon = \mathcal{O}(\log(1/\epsilon_1)) = \mathcal{O}(\log(nd) + \log(1/\gamma))$.

Furthermore, under the event $\mathcal{E}_2$, Algorithm 13 computes a $10\epsilon nd$-optimal solution, as guaranteed by Lemma 4, with $\widehat{\mu} = \mu$.

Thus, with probability at least $1 - \eta$, Algorithm 13 computes a $\gamma$-optimal solution in:

$$\widetilde{\mathcal{O}}\left(\frac{n^3}{\gamma} \log^2 (1/\eta)\left(d^8 B + d\binom{d+n}{d}\right)\right)$$

rounds, concluding the proof. $\qquad\qquad\square$

We notice that, differently from the case with an unknown prior, it is possible to achieve a $\mathcal{O}\left(\log(1/\eta)/\gamma\right)$ upper bound with respect to the input parameters $\gamma, \eta > 0$. Finally, we show that such a dependence is tight, as shown in the following theorem.

**Theorem 8.** *Given $\gamma, \eta > 0$ no algorithm is guaranteed to return an $\gamma$-optimal signaling scheme with probability of at least $1 - \eta$ employing $\Omega\left(\log(1/\eta)/\gamma\right)$ rounds, even when the prior distribution is known to the sender.*

*Proof.* We consider two instances characterized by two states of nature and three receiver's actions. The two instances share the same prior distribution, defined as $\mu_{\theta_1} = 4\gamma$ and $\mu_{\theta_2} = 1 - 4\gamma$. In both the instances the sender's utility is given by $u_\theta^s(a_1) = 0$, $u_\theta^s(a_2) = 1/2$ and $u_\theta^s(a_3) = 1$ for all $\theta \in \Theta$. Furthermore, we assume that the receiver's utility in the two instances are given by:

$$① \begin{cases} u_{\theta_1}(a_1) = 1, \ u_{\theta_2}(a_1) = 1/2 \\ u_{\theta_1}(a_2) = 1/2, \ u_{\theta_2}(a_2) = 1 \\ u_{\theta_1}(a_3) = 1, \ u_{\theta_2}(a_3) = 0 \end{cases} \qquad ② \begin{cases} u_{\theta_1}(a_1) = 1, \ u_{\theta_2}(a_1) = 1/2 \\ u_{\theta_1}(a_2) = 1/2, \ u_{\theta_2}(a_2) = 1 \\ u_{\theta_1}(a_3) = 1/2, \ u_{\theta_2}(a_3) = 0 \end{cases}$$

We observe that the only case in which the sender receives different feedback in the two instances is when they induce the posterior distribution $\xi_1 := (1, 0)$. This is because, when the sender induces $\xi_1$ in the first instance, the receiver plays the action $a_3 \in \mathcal{A}$, breaking ties in favor of the sender, while in the second instance, the receiver plays the action $a_1 \in \mathcal{A}$. Such a posterior can be induced, in both the two instances, with a probability of at most $\gamma$ to be consistent with the prior.

We also observe that in the first instance the optimal sender's signaling scheme $\gamma$ is such that $\gamma(\xi_1) = 4\gamma$ and $\gamma(\xi_2) = 1 - 4\gamma$, where we let $\xi_2 := (0, 1)$. Furthermore, the sender's expected utility in $\gamma$ is equal to $(1 + 4\gamma)/2$. In the second instance, the optimal sender's signaling scheme $\gamma$

is such that $\gamma(\xi_2) = 1 - 8\gamma$ and $\gamma(\xi_3) = 8\gamma$, with $\xi_3 := (1/2, 1/2)$. It is easy to verify that such a signaling scheme achieves an expected utility of $1/2$.

In the following, we let $\mathbb{P}^1$ and $\mathbb{P}^2$ be the probability measures induced by the interconnection of a given algorithm executed in the first and in the second instances, respectively. Furthermore, we introduce the event $E_N$, under which, during the first $N$ rounds, the sender never observes the action $a_3 \in \mathcal{A}$. It is easy to verify that such an event holds with a probability of at least $\mathbb{P}^1(E_N) \geq (1-4\gamma)^N$ in the first instance. This is because, at each round, the action $a_3$ can be observed with a probability of at most $4\gamma$. In contrast, since in the second instance the receiver never plays the action $a_3$, it holds $\mathbb{P}^2(E_N) = 1 \geq (1-4\gamma)^N$.

Then, by letting $\gamma_1^N$ ($\gamma_2^N$) be the signaling scheme returned after $N$ rounds in the first (second) instance, we have:

$$
\begin{aligned}
\mathbb{P}^1\left(\gamma_1^N(\xi_1) \geq 2\gamma\right) + \mathbb{P}^2\left(\gamma_2^N(\xi_1) \leq 2\gamma\right) &\geq \mathbb{P}^1\left(\gamma_1^N(\xi_1) \geq 2\gamma, E_N\right) + \mathbb{P}^2\left(\gamma_2^N(\xi_1) \leq 2\gamma, E_N\right) \\
&\geq \mathbb{P}^1\left(\gamma_1^N(\xi_1) \geq 2\gamma \,|E_N\right)\mathbb{P}^1\left(E_N\right) + \mathbb{P}^2\left(\gamma_2^N(\xi_1) \leq 2\gamma \,|E_N\right)\mathbb{P}^2\left(E_N\right) \\
&\geq \left(\mathbb{P}^1\left(\gamma_1^N(\xi_1) \geq 2\gamma \,|E_N\right) + \mathbb{P}^2\left(\gamma_2^N(\xi_1) \leq 2\gamma \,|E_N\right)\right)\mathbb{P}^1\left(E_N\right) \\
&= \mathbb{P}^1\left(E_N\right) \\
&\geq (1-4\gamma)^N \geq 2\eta. \tag{13}
\end{aligned}
$$

The above result holds observing that, under the event $E_N$, the behaviour of any algorithm working in the first instance coincides with the behaviour of the same algorithm working in the second one, as they receive the same feedback. Furthermore, Inequality 13 holds when $N$ is such that:

$$
N \leq \frac{\log(1/2\eta)}{10\gamma} \leq \frac{\log(2\eta)}{\log(1-4\gamma)},
$$

if $\gamma \leq 1/5$.

Finally, we observe that if $\gamma_1^N(\xi_1) \leq 2\gamma$, then the sender's expected utility in the first instance is of most $1/2 + \gamma$. This is because, for any signaling scheme $\gamma_1^N$, we have:

$$
u^s(\gamma_1^N) \leq \gamma_1^N(\xi_1)u^s(\xi_1) + \frac{1}{2}\left(1 - \gamma_1^N(\xi_1)\right) = \frac{1}{2} + \gamma.
$$

Thus, if $\gamma_1^N(\xi_1) \leq 2\gamma$ the the signaling scheme $\gamma_1^N$ is at most $\gamma$-optimal.

Equivalently, if the sender's final signaling scheme $\gamma_2^N$ in the second instance is such that $\gamma_2^N(\xi_1) \geq 2\gamma$, then the sender's utility in the second instance is of at most $1/2 - \gamma$. Thus, if $\gamma_2^N(\xi_1) \geq 2\gamma$ the the signaling scheme $\gamma_2^N$ is at most $\gamma$-optimal. Then, we either have:

$$
\mathbb{P}^1\left(\gamma_1^N(\xi_1) \leq 2\gamma\right) \geq \eta \quad \text{or} \quad \mathbb{P}^2\left(\gamma_2^N(\xi_1) \geq 2\gamma\right) \geq \eta,
$$

showing that if the number of rounds $N \leq \log(1/2\eta)/(10\gamma)$, there exists an instance such that no algorithm is guaranteed to return a $\gamma$-optimal signaling scheme with probability greater than or equal to $1 - \eta$. $\qquad\square$

