# OpenReview forum: "Online Bayesian Persuasion Without a Clue"
_NeurIPS.cc/2024/Conference — NeurIPS 2024 spotlight_

### Official Review · Reviewer_yxNt · 2024-07-10

**Soundness:** 4
**Presentation:** 3
**Contribution:** 3
**Rating:** 6
**Confidence:** 3

**Summary:**

The paper studies online Bayesian persuasion with "no clue", i.e., the sender knows nothing about the prior distribution or the receiver's utility function a priori (it is however assumed that there exists a prior distribution and states are drawn iid from it).  The main results are (1) in the online model where the sender tries to learn a good signaling scheme on the fly while not losing too much utility in the process, there is an algorithm that achieves square-root regret, (2) lower bounds showing that the dependency of this regret bound on various parameters are in a sense unavoidable, and (3) in the PAC model, an adapted algorithm that achieves a nontrivial sample complexity bound.

**Strengths:**

The problem appears meaningful and technically interesting.  The idea of slices is natural, clean and powerful.  The overall plan is clearly plausible, and yet not obviously viable.  I'm happy to see that the authors were able to make it work, which seems to have involved nontrivial effort.  The bound is reasonably tight.

**Weaknesses:**

Ideally an instance-optimal bound would be more exciting, but perhaps that's too much to ask for.  I'd also like to see more discussion of the PAC model and the optimality of the bound in that model (if applicable).

**Questions:**

(also including detailed comments)

Line 22, "such as, e.g., ...": a bit repetitive?

Line 92: superscript missing in the definition of $u_\theta^s$?

Footnote 1: might as well quickly define int while you are defining $\Delta$

Line 110: missing ")"

Sec 3: the title was a bit confusing to me, but I think the section as a whole is very helpful for a non-expert reader to develop intuition.  It's also very clearly written.

Line 218: "sufficiency"

Overview part of sec 4: my first instinct is that explore-then-commit strategies normally give regret bounds like $O(T^{2/3})$.  This will probably become clear soon, but it might help to quickly discuss why you were able to get square-root regret here.  (My guess is somehow you managed to stay in the "realizable" regime where learning up to eps requires only 1/eps rounds.)

Sec 6: maybe quickly introduce the setup first (in particular, what is a "sample" here)?  Relatedly, it's also interesting that here you get $1/gamma^2$ instead of $1/\gamma$,

**Limitations:**

No concerns.

---

> ### Author Rebuttal · Authors · 2024-08-06
>
> We thank the Reviewer for the insightful feedbacks and for pointing out some typos. We will incorporate these suggestions and correct the typos in the final version of the paper.
>
> > Overview part of sec 4: my first instinct is that explore-then-commit strategies normally give regret bounds like $O(T^{2/3})$. This will probably become clear soon, but it might help to quickly discuss why you were able to get square-root regret here. (My guess is somehow you managed to stay in the "realizable" regime where learning up to eps requires only 1/eps rounds.)
>
> The Reviewer is right. The main reason we do not incur in a $O(T^{2/3})$ regret is that we can build a space of signaling schemes slices in $O(1 / \epsilon)$ rounds, such that, for each element, it is possible to associate a receiver's best response in $O(1 / \epsilon)$ rounds.
> Technically, this is made possible by employing the multiplicative Chernoff bound, which allows us, with $O(1 / \epsilon)$ rounds, to distinguish which states $\theta \in \Theta$ have a probability of being observed smaller than $\epsilon$, i.e., $\mu_\theta \le \epsilon$.
> In this way, the "exploration phase" of our explore-then-commit approach requires $O(1 / \epsilon)$ rounds, allowing us to achieve a $\sqrt{T}$ regret, by suitably choosing $\epsilon = O(1/\sqrt{T})$, instead of $O(T^{2/3})$ regret, as it is typically the case in explore-then-commit approaches.
>
> Finally, we observe that in our "commit" phase, the sender does not commit to a fixed signaling scheme. Instead, they select a different signaling scheme $\phi_t$ computed by means of Algorithm 3, which receives as input an estimate of the prior distribution $\widehat \mu_t$ that is updated at each round.
>
>
> > Sec 6: maybe quickly introduce the setup first (in particular, what is a "sample" here)?
>
> We thank the Reviewer for giving us the opportunity to better clarify the concept of ``sample'' within our framework.
> We use the term "sample" to refer to the feedbeack received by the sender when they commit to a signaling scheme.
> Intuitively, in a PAC-learning problem we do not have a finite time horizon, nor we are interested in the regret.
> Instead, we want to learn a $\gamma$-optimal solution by using the minimum possible number of samples (or equivalently, rounds), and we are not concerned about the regret accumulated while learning it.

---

> > ### Comment · Reviewer_yxNt · 2024-08-13
> >
> > Thank you for your helpful response.  I don't have further questions.

---

### Official Review · Reviewer_YTqh · 2024-07-12

**Soundness:** 3
**Presentation:** 3
**Contribution:** 3
**Rating:** 7
**Confidence:** 4

**Summary:**

This paper studies repeated Bayesian persuasion where the sender does not know the prior distribution of the state of the world and the receiver's utility (while the receiver knows the prior and their utility and myopically best responds in each period). The authors design online learning algorithm for the sender to achieve sublinear regret $O( binom(d+n, d) n^{3/2} d^3 \sqrt{BT})$. They also prove lower bounds of $2^{\Omega(d)}$ and $2^{\Omega(n)}$. The proposed algorithm, a sophisticated explore-than-commit algorithm, works by searching the space of signaling schemes under a non-standard representation.

**Strengths:**

(1) [Significance] This work is a significant improvement over previous works on online learning in Bayesian persuasion where the sender learns either the prior or the receiver's utility. Learning the prior and utility simultaneously seems to be challenging. The authors are able to solve this problem, which is a significant contribution.

(2) [Originality] A key technique in this work is performing high-dimensional binary search in the space of signaling schemes under a non-standard representation. In previous works, signaling schemes were usually represented by distributions over posterior distributions, and the search was performed on the space of posterior distributions. Such a technique does not work when the prior is unknown. This work instead represents signaling schemes by the vector $(\sigma_\theta(s))_{\theta \in \Theta}$, the probabilities of sending a signal $s$ under different states $\theta$. This new representation circumvents the unknown prior issue. So, I think this new representation is an interesting trick and a technical novelty.

(3) [Quality] Non-trivial lower bound results are provided, complementing the upper bounds results.

**Weaknesses:**

(1) [Quality] The proposed algorithm seems to have an exponential running time in the worst case. Specifically, the number of vertices $\mathcal V$ of the polytopes seem to be exponential in $d$ or $n$, which leads to an exponential running time. The authors didn't discuss the computational complexity of their algorithm, nor provided a computational-hardness result.

**Questions:**

Can the authors respond to weakness (1)?

**Limitations:**

**Suggestions:**

(1) Please define the B in algorithm 1 more clearly.

(2) In Appendix A Additional Related Works, in additional to saying what the related works did, please briefly compare those works with your work.

---

> ### Author Rebuttal · Authors · 2024-08-06
>
> We thank the Reviewer for the insightful feedbacks. We will also incorporate the two Reviewer's suggestions in the final version of the paper.
>
> > The proposed algorithm seems to have an exponential running time in the worst case. Specifically, the number of vertices $\mathcal{V}$ of the polytopes seem to be exponential in $d$ or $n$, which leads to an exponential running time. The authors didn't discuss the computational complexity of their algorithm, nor provided a computational-hardness result.
>
> We thank the Reviewer for giving us the opportunity to better clarify this aspect of our paper.
> The per-round running time of our algorithm is polynomial in the size of the input when either the number of receiver’s actions $n$ or that of states of nature $d$ is fixed.
> We agree with the Reviewer on the fact that, if both $n$ and $d$ are not fixed, then the per-round running time may no longer be polynomial in the input size in the worst case.
>
> We observe that Algorithm 6 can find the H-representation of the polytopes $\mathcal{X}\_\epsilon(a)$ with polynomial per-round running time.
> In particular, to enumerate the vertices of the upper bounds, it can compute and query a different vertex at each round, without ever storing them all.
> In the current version, given the H-representation of the polytopes, our algorithm also computes the set of all the vertices $\mathcal{V}$, and eventually employs this set to instantiate an LP, whose cardinality may be exponential in either $n$ or $d$.
> However, it is possible to avoid computing these vertices and solve an equivalent linear program whose size is polynomial in $n$ and $d$.
> This can be done by employing the H-representation of the polytopes $\mathcal{X}\_\epsilon(a)$ computed by Algorithm 6, and exploiting a slightly different LP.
> This approach does not require to compute $\mathcal{V}$, thus achieving a polynomial per-round running time in every phase of the algorithm.
>
> The theoretical analysis with the modified LP is almost straightforward given the results presented in the paper, as it only requires the introduction of some additional technical lemmas about polytopes.
> We will include this algorithmic approach in the final version of the paper and we are happy to provide additional technical details, if the Reviewer wants to.
>
>
> > (1) Please define the B in algorithm 1 more clearly. (2) In Appendix A Additional Related Works, in additional to saying what the related works did, please briefly compare those works with your work.
>
> We thank the Reviewer for the suggestions; we will incorporate them into the final version of the paper.

---

> > ### Comment · Reviewer_YTqh · 2024-08-10
> > **Happy with authors' response**
> >
> > I am happy with the authors' response and raise rating to 7.  Indeed, adding discussion on the computational complexity might further strengthen the paper.

---

### Official Review · Reviewer_cPaK · 2024-07-12

**Soundness:** 3
**Presentation:** 3
**Contribution:** 4
**Rating:** 7
**Confidence:** 4

**Summary:**

The paper studies Bayesian persuasion in a learning setting with minimum knowledge about the receiver: neither the receiver's prior nor their utility function is known. In the model, the sender can commit to different signaling strategies and acquire the receiver's optimal response to each signal they send. The paper presents a learning algorithm that achieves regret sublinear in the number of rounds but exponential in the number of states. The authors further show that the exponential dependency on the number of states is inevitable by proving a matching sample complexity.

**Strengths:**

The paper follows a line of work in the literature on Stackelberg games that study how to learn to commit optimally by querying the follower's optimal responses. I find it natural and reasonable to ask the same question in Bayesian persuasion. While the overall approach works similarly by building the follower's (receiver's) best response regions, the paper shows that there are some unique features of the persuasion setting, which require additional techniques.

The estimation of the prior, in particular, can only be done approximately. The paper nicely handles this aspect in their algorithm. The concept of slice is also novel and interesting (though the naming is somewhat less intuitive).

The results look very complete, with matching lower and upper regret bounds. The paper is also well-writen and overall clear.

**Weaknesses:**

- The part below Definition 1 until the end of that page is a little dense and could probably be improved. The notation in this part is also a bit hard to follow. However, this is overall minor.

- Line 296: It might be better to be a bit more specific about the distinguished features here.

- Is there any justification how the receiver gets to know the sender's strategy $\phi_t$ in each round? How would the results change if the receiver only observes the sender's signal in each round?

Typos:

- Line 51: of much (how much?)

- Line 314: clean event

**Questions:**

- In algorithm 2, the feedback a^t and u_t^s seem not useful? If so, better to remove this line?

- What would happen if the receiver is not truthful? And since the receiver knows their prior, does it make sense to design a mechanism to directly elicit this prior knowledge?

**Limitations:**

Nothing is mentioned, but no concern here.

---

> ### Author Rebuttal · Authors · 2024-08-06
>
> We thank the Reviewer for the insightful feedbacks and for pointing out some typos. Specifically, we will do our best to improve the part below Definition 1  and the comparison between previous models at Line 296 in the final version of the paper.
>
> > Is there any justification how the receiver gets to know the sender's strategy $\phi_t$ in each round? How would the results change if the receiver only observes the sender's signal in each round?
>
> The assumption that the receiver gets to know the sender's signaling scheme (a.k.a. the commitment assumption) is inherent in every Bayesian persuasion model. Moreover, the same assumption is standard in online Bayesian persuasion settings, where the receiver observes the sender's signaling scheme $\phi_t$ at each round $t \in [T]$. See, e.g., Castiglioni et al. (2020b) and Zu et al. (2021).
>
> Notice that, by dropping the commitment assumption, one falls in a completely different model, which is commonly referred to as "cheap talk" in the economic literature. Indeed, if the receiver only observes a signal $s \sim \phi_{t, \theta_t}$ sampled from the signaling scheme $\phi_t$  at round $t$, without knowing $\phi_t$, it is not immediate how to define a best response for the receiver. We believe that addressing online learning problems in such a different setting is a very interesting research direction that is worth addressing in the future.
>
>
> > In algorithm 2, the feedback $a^t$ and $u_t^s$ seem not useful? If so, better to remove this line?
>
> We agree with the Reviewer, we will omit it from Algorithm 2.
>
>
> > What would happen if the receiver is not truthful? And since the receiver knows their prior, does it make sense to design a mechanism to directly elicit this prior knowledge?
>
> Addressing settings with a non-truthful receiver is an interesting research direction that we intend to explore in the future. This begets considerable challenges that are out the scope of this paper.
>
> Designing a mechanism in which the sender elicits information from the receiver's knowledge of the prior is certainly an interesting idea. However, since the sender has never access to the receiver's payoffs, it is not clear how the sender could effectively benefit from the receiver's knowledge of the prior. Indeed, receiver's best responses depend on both the unknown prior and receiver's unknown payoffs, which makes it challenging to extract information about the prior only.

---

> > ### Comment · Reviewer_cPaK · 2024-08-12
> >
> > Thank you for your responses!

---

### Official Review · Reviewer_UDVm · 2024-07-13

**Soundness:** 4
**Presentation:** 4
**Contribution:** 4
**Rating:** 8
**Confidence:** 5

**Summary:**

The paper studies online Bayesian persuasion problems in which an informed sender
repeatedly faces a receiver with the goal of influencing their behavior through the provision of payoff-relevant information. The paper considers a setting where the sender does not know anything about the prior and the receiver’s utility function. At each round, the sender commits to a signaling scheme, and, then, they observe a state realization and send a signal to the receiver based on that. After each round, the sender gets partial feedback, namely, they only observe the best-response action played by the receiver in that round.

The main results of this paper are: (1) providing a learning algorithm that achieves $\tilde{O}(\sqrt{T})$ regret, and this regret bound has exponential dependence on the number of states $d$ and the number of receiver’s actions $n$; (2) a set of lower bounds showing that such $\sqrt{T}$-dependency is optimal, and such exponential dependency on $d, n$ is also optimal; (3) extending the no-regret learning algorithm to establish the sample complexity of the Bayesian persuasion PAC-learning problem.

**Strengths:**

I truly enjoy reading this work. I think this work significantly contributes to the recent line of research on using regret minimization to study the persuasion with uncertainty over the underlying environments. The results developed in this work are strong and interesting. The presentation of this work is crisp and the paper provides intuitions for the algorithmic challenges and its steps of algorithm design. Overall, I think this work makes a strong addition to the NeurIPS.

**Weaknesses:**

I don’t have particular concerns about the paper. Maybe one minor point is that the authors may want to include some discussions about the computational efficiency of the designed algorithm.

**Questions:**

It may be nice to present the $\tilde{O}(n^d \sqrt{T})$ regret and the $\tilde{O}(d^n \sqrt{T})$ regret as the two corollaries of the Theorem 1.

It is indeed a bit interesting to me that the unknown of both prior and receiver’s utilities make the learning problem significantly challenge: one has to incur $\sqrt{T}$-regret. Do you have intuitions on  this $\sqrt{T}$-regret is mainly due to unknown receiver's utilities or the unknown priors? \
-- It seems that lower bound $\sqrt{T}$ in Theorem 3 requires 4 receiver’s actions. Does $\sqrt{T}$ lower bound also hold if receiver only has binary actions? Do authors feel the regret may be improved (e.g., $\Theta(\log T)$ or even $\Theta(\log\log T)$) if one consider only BINARY receiver action? As in the paper "EC’21-Online bayesian recommendation with no regret", the authors show that when the sender only does not know the receiver’s utilities, a regret $\Theta(\log\log T)$ by some variant of binary search is attainable but focusing on binary actions. (So I kind of feel your $\sqrt{T}$-regret may be mainly due to unknown prior of the state realizations)

I would imagine that most of the results still hold even if the receiver has a different unknown prior to the sender (but the stats are still realized from sender's unknown prior), right?

---

> ### Author Rebuttal · Authors · 2024-08-06
>
> We thank the Reviewer for the insightful comments. We will adopt them in the final version of the paper and we will better discuss the points suggested by the Reviewer.
>
> > It is indeed a bit interesting to me that the unknown of both prior and receiver’s utilities make the learning problem significantly challenge: one has to incur $\sqrt{T}$-regret. Do you have intuitions on this $\sqrt{T}$-regret is mainly due to unknown receiver's utilities or the unknown priors?
>
> The $\Omega(\sqrt{T})$ regret lower bound is a consequence of the sender's lack of knowledge of **both** the receiver's utilities and the prior distribution.
> Indeed, in our setting, it is possible to construct two instances with similar prior distributions and define the receiver's utilities so that the sender gains no information to distinguish between the two instances by committing to any signaling scheme.
> This is not possible if the utilities are known.
> Consequently, to distinguish between the two instances, the sender can only leverage the information contained in the states of nature sampled at each round and this results in the regret being at least $\Omega (\sqrt T)$.
>
>
> > It seems that the lower bound $\sqrt{T}$ in Theorem 3 requires 4 receiver’s actions. Does the $\sqrt{T}$ lower bound also hold if the receiver only has binary actions? Do the authors feel the regret may be improved (e.g., $\Theta(\log T)$ or even $\Theta(\log \log T)$) if one considers only binary receiver actions? As in the paper "EC’21—Online Bayesian Recommendation with No Regret," the authors show that when the sender only does not know the receiver’s utilities, a regret of $\Theta(\log \log T)$ by some variant of binary search is attainable when focusing on binary actions. (So I kind of feel your $\sqrt{T}$-regret may be mainly due to the unknown prior of the state realizations).
>
> Developing a regret lower bound for instances with binary actions is an interesting research direction that we intend to explore in the future. We leave as an open problem whether it is possible to construct two instances similar to those presented in our lower bound, but with binary receiver actions, and still achieve a $\Omega(\sqrt{T})$ regret lower bound or a better upper bound can be achieved.
>
>
> > I would imagine that most of the results still hold even if the receiver has a different unknown prior to the sender (but the stats are still realized from sender's unknown prior), right?
>
> The Reviewer is right. Our algorithm can also be employed in settings where the receiver has a different prior but the states of nature are sampled from the sender's unknown prior. We will better outline this aspect in the final version of the paper.

---

### Decision · Program_Chairs · 2024-09-25

**Decision:**

Accept (spotlight)

**Comment:**

All reviewers are positive about the paper. They consider the paper significant in the online Bayesian persuasion literature. They like the paper's concept of slice. And they are happy with the paper's presentation and completeness.